# Stable water isotopes and tritium tracers tell the same tale: No evidence for underestimation of catchment transit times inferred by stable isotopes in SAS function models.

Siyuan Wang[1], Markus Hrachowitz[1], Gerrit Schoups[1], Christine Stumpp[2]

[1]Department of Water Management, Faculty of Civil Engineering and Geosciences, Delft University of Technology, Stevinweg 1, 2628CN Delft, Netherlands
[2]Institute of Soil Physics and Rural Water Management, University of Natural Resources and Life Sciences Vienna, Muthgasse 18, 1190 Vienna, Austria

*Correspondence to:* Siyuan Wang (S.Wang-9@tudelft.nl)

**Abstract.** Stable isotopes ($\delta^{18}O$) and tritium ($^3H$) are frequently used as tracers in environmental sciences to estimate age distributions of water. However, it has previously been argued that seasonally variable tracers, such as $\delta^{18}O$, generally and systematically fail to detect the tails of water age distributions and therefore substantially underestimate water ages as compared to radioactive tracers, such as $^3H$. In this study for the Neckar river basin in central Europe and based on a >20-year record of hydrological, $\delta^{18}O$ and $^3H$ data, we systematically scrutinized the above postulate together with the potential role of spatial aggregation effects to exacerbate the underestimation of water ages. This was done by comparing water age distributions inferred from $\delta^{18}O$ and $^3H$ with a total of 21 different model implementations, including time-invariant, lumped parameter sine-wave (SW) and convolution integral models (CO) as well as SAS-function models (P-SAS) and integrated hydrological models in combination with SAS-functions (IM-SAS).

We found that, indeed, water ages inferred from $\delta^{18}O$ with commonly used SW and CO models are with mean transit times (MTT) ~ 1 – 2 years substantially lower than those obtained from $^3H$ with the same models, reaching MTTs ~ 10 years. In contrast, several implementations of P-SAS and IM-SAS models did not only allow simultaneous representations of storage variations and stream flow as well as $\delta^{18}O$ and $^3H$ stream signals, but water ages inferred from $\delta^{18}O$ with these models were with MTTs ~ 11 – 17 years much higher and similar to those inferred from $^3H$, which suggested MTTs ~ 11 – 13 years. Characterized by similar parameter posterior distributions, in particular for parameters that control water age, P-SAS and IM-SAS model implementations individually constrained with $\delta^{18}O$ or $^3H$ observations, exhibited only limited differences in the magnitudes of water ages in different parts of the models as well as in the temporal variability of TTDs in response to changing wetness conditions. This suggests that both tracers lead to comparable descriptions of how water is routed through the system. These findings provide evidence that allowed us to reject the hypothesis that $\delta^{18}O$ as a tracer generally and systematically "cannot see water older than about 4 years" and that it truncates the corresponding tails in water age distributions, leading to underestimations of water ages. Instead, our results provide evidence for a broad equivalence of $\delta^{18}O$ and $^3H$ as age tracers for systems characterized by MTTs of at least 15 – 20 years. The question to which degree aggregation of spatial heterogeneity

can further adversely affect estimates of water ages remains unresolved as the lumped and distributed implementations of the IM-SAS model provided inconclusive results.

Overall, this study demonstrates that previously reported underestimations of water ages are most likely not a result of the use of $\delta^{18}O$ or other seasonally variable tracers *per se*. Rather, these underestimations can be largely attributed to choices of model approaches and complexity not considering transient hydrological conditions next to tracer aspects. Given the additional vulnerability of time-invariant, lumped SW and CO model approaches in combination with $\delta^{18}O$ to substantially underestimate water ages due to spatial aggregation and potentially other, still unknown effects, we therefore advocate to avoid the use of this model type in combination with seasonally variable tracers if possible, and to instead adopt SAS-based models or time-variant formulations of CO models.

## 1 Introduction

Age distributions of water fluxes ("transit time distributions", TTD) and water stored in catchments ("residence time distributions", RTD) are fundamental descriptors of hydrological functioning (Botter et al., 2011; Sprenger et al., 2019) and catchment storage (Birkel et al., 2015). They provide a way to quantitatively describe the physical link between the hydrological response of catchments and physical transport processes of conservative solutes. While the former is largely controlled by the celerities of pressure waves propagating through the system, the latter, in contrast, occur at velocities that can be up to several orders of magnitude lower (McDonnell and Beven, 2014; Hrachowitz et al., 2016).

Water age distributions cannot be directly observed. Instead, they can, in principle, be inferred from observed tracer breakthrough curves. While practically feasible at lysimeter (e.g. Asadollahi et al., 2020; Benettin et al., 2021) and small hillslope scales (e.g. Kim et al., 2022), lack of adequate observation technology together with logistical constraints make this problematic at scales larger than that. At the catchment-scale, estimates of water age distributions are therefore typically inferred from models that describe the relationships between time-series of observed tracer input and output signals.

Over the past decades a wide spectrum of such models has been developed. Early approaches often relied on simple lumped sine-wave (hereafter: SW) or lumped parameter convolution integral models (hereafter CO; Maloszewski and Zuber, 1982; Maloszewski et al.,1983; McGuire and McDonnell, 2006), originally developed for aquifers. In spite of their wide-spread application, these models feature multiple critical simplifying assumptions. Most importantly, the vast majority of these model implementations work under the assumption that water storage in catchments is at steady state and that, as a consequence, TTDs are time-invariant and can be *a priori* defined or calibrated. While the role of storage as first order control on water ages was described early in the general definition of mean turnover times (e.g. Eriksson, 1958; Bolin and Rodhe, 1973; Nir, 1973), the steady state assumption, i.e. constant storage, may have limited effect on TTDs in aquifers, as the fraction of transient water volumes in such systems is typically rather low. However, given the temporal variability in the hydro-meteorological system drivers (e.g. precipitation, atmospheric water demand) and the spatial heterogeneity in near-surface hydrological processes, this assumption is violated in most surface water systems world-wide and can lead to misinterpretations of the

model results. This triggered the development of a more coherent framework to estimate water age distributions without the need of an *a priori* definition of time-invariant TTDs. Instead, probability distributions, referred to as StorAge Selection (SAS) functions, are *a priori* defined or calibrated, and changes in water storage are explicitly accounted for. Thus, water fluxes within and released from the system are sampled from water volumes of different ages stored in the system according to these SAS functions (Botter et al., 2011; Rinaldo et al., 2015). The general concept is firmly rooted in the development of hydro-chemical routing schemes for the Birkenes, HBV or similar models going back to at least the 1970s (e.g. Lundquist, 1977; Christophersen and Wright, 1981; Christophersen et al., 1982; Seip et al., 1985; de Groisbois et al., 1988; Hooper et al., 1988; Barnes and Bonell, 1996), as illustrated by Figure 1 in Bergström et al. (1985). Although functionally very similar to CO model implementations that allow for transient, i.e. time-variant TTDs (Nir, 1973; Niemi, 1977), the sampling procedure based on SAS functions has the advantage to explicitly track the history of water (and tracer) input to and output from the system through the water age balance. As such it does explicitly account for non-steady state conditions, which in turn leads to the emergence of time-variable TTDs and RTDs (see review Benettin et al., 2022).

Irrespective of the modelling approach, two types of environmental tracers have in the past been frequently used to estimate water age distributions with the above models. The first type are tracers that are characterized by distinct differences in their seasonal signals. They include stable isotopes of water ($^2$H, $^{18}$O; e.g. Maloszewski et al., 1983; Vitvar and Balderer, 1997; Fenicia et al., 2010) or solutes, such as $Cl^-$ (e.g. Kirchner et al., 2001, 2010; Shaw et al., 2008; Hrachowitz et al., 2009a, 2015). With these tracers, water ages and (metrics of) their distributions can be estimated by the degree to which the seasonal amplitudes of the precipitation tracer concentrations are time-shifted and/or attenuated in the stream flow (McGuire and McDonnell, 2006; Kirchner, 2016). Broadly speaking, the stronger the attenuation of the seasonally variable tracer amplitude in stream flow ($A_s$) as compared to its amplitude in precipitation ($A_p$), i.e., the lower the amplitude ratio $A_s/A_p$, the older stream water is, on average. The second type of commonly used tracers are radioactive isotopes, such as tritium ($^3$H). Forming the basis for many water dating studies going back to the 1950s (e.g. Begemann and Libby, 1957; Eriksson, 1958; Dincer et al., 1970; Stewart et al., 2007; Morgenstern et al., 2010; Duvert et al., 2016; Gallart et al., 2016; Rank et al., 2018; Visser et al., 2019), water age can be estimated with radioactive tracers based on the level of radioactive decay experienced by precipitation input signals experience before they reach the stream.

The relationship between the tracer amplitude ratios $A_s/A_p$ and water age that is exploited by seasonally variable tracers is highly non-linear. With increasing attenuation of the tracer signal in the stream, i.e., a lower $A_s/A_p$, water therefore does not only become older but the age estimates become more sensitive to changes in the amplitude ratio (Kirchner, 2016). This implies that the older the water, uncertainties in the observed amplitude ratios lead to increased uncertainties in water age estimates. As a consequence, there is an upper limit to the age of water which can be practically and feasibly determined with seasonally variable tracers. A rare attempt to quantify this potential upper detectible age limit was reported by DeWalle et al. (1997). With an observed $\delta^{18}$O precipitation amplitude $A_p = 3.41$‰, an assumed lowest possible $\delta^{18}$O stream water amplitude that equaled the observational error $A_s = 0.1$‰, and the use of a lumped, time-invariant exponential TTD ("complete mixing") they determined a maximum detectable mean transit time (MTT) of around 5 years at their study site. Several authors

subsequently emphasized that estimates of MTT and in particular of maximum detectable MTT such as reported by DeWalle et al. (1997) are specific to $A_p$ at individual study sites (McGuire and McDonnell, 2006) and highly sensitive to choices in the modelling process (Stewart et al., 2010; Seeger and Weiler, 2014; Kirchner, 2016). For example, multiple previous studies demonstrated that the use of gamma distributions with a shape parameter $\alpha \sim 0.5$ as TTD produces model results that are more consistent with observed tracer data than the use of exponential distributions (i.e. $\alpha = 1$) in a wide range of contrasting environments world-wide (Kirchner et al., 2001; Godsey et al. 2010; Hrachowitz et al., 2010a, b). Merely replacing the exponential distribution by a gamma distribution with $\alpha = 0.5$ as TTD at the study site of DeWalle et al. (1997) leads, in a quick back-of-the-envelope calculation, to a substantial increase of the maximum MTT from the reported 5 years to $\sim 90$ years. This is exacerbated by the potential presence of spatial aggregation bias in the lumped implementation of that model, which may cause further considerable underestimation of MTT as demonstrated by Kirchner (2016).

The relevance of the above assumptions is often overlooked and in spite of little additional quantitative evidence, it remains widely assumed that water ages in systems characterized by MTTs > 4 – 5 years cannot be meaningfully quantified with seasonally variable tracers. Most notably, Stewart et al. (2010, 2012) argued that water older than that remains *hidden* to stable water isotopes and other seasonally variable tracers, which inevitably results in a misleading truncation of water age distributions. Such a pronounced and systematic underestimation of water ages would have far reaching consequences for estimates of water storage (e.g. Birkel et al., 2015; Pfister et al., 2017) and the associated turnover times of nutrients and contaminants in catchments (e.g. Harman, 2015; Hrachowitz et al., 2015). Stewart et al. (2012), further argue that the use of radioactive tracers, such as $^3H$, can largely avoid the truncation of the long tails of TTDs. This is mostly owed to the $^3H$ half-life of $T_{1/2} = 12.32$ years. Even with the current atmospheric $^3H$ concentrations that, after peaking in the early 1960s, have been converging back towards pre-nuclear bomb testing levels, precipitation $^3H$ signals can be detected in the system for several decades, making $^3H$ an effective tracer now and for the foreseeable future (Michel et al., 2015; Harms et al., 2016; Stewart and Morgenstern, 2016). Indeed, a range of studies, based on $^3H$ and often in conjunction with lumped parameter convolution integral approaches, suggest that many catchments and larger river basins world-wide are characterized by MTTs that are decadal or higher (e.g. Stewart et al., 2010 and references therein). It is further rather remarkable that such elevated water ages are largely absent in estimates derived from lumped parameter convolution integral studies based on seasonally variable tracers, which often indicate MTTs between 1 – 3 years (e.g. McGuire and McDonnell, 2006 and references therein; Hrachowitz et al., 2009b; Godsey et al., 2010), as correctly and importantly pointed out by Stewart et al. (2010). This in itself could be supporting evidence for the failure of seasonally variable tracers to detect long tails of TTDs, as postulated by Stewart et al. (2012). However, it could just as well be a mere artifact arising from a sample bias due to the different catchments analyzed or from choices in the modelling process. There are only a few studies that have directly and systematically compared estimates of water age derived from both, seasonally variable ($^2H$, $^{18}O$) and radioactive tracers ($^3H$) at the same study site and based on (at least partly) comparable model approaches (Maloszewski et al., 1983; Uhlenbrook et al., 2002; Stewart et al., 2007; Stewart and Thomas, 2008). The MTT estimates derived from seasonally variable tracers in these comparative studies are consistently, but to varying degrees lower than estimates based on $^3H$. However, these studies

are nevertheless subject to limitations that may weaken the generality of the conclusion that seasonally variable tracers underestimate catchment water ages. More specifically, tracer data were available for only rather short time periods of about 2 – 3 years, including, for some studies, only a handful of $^3$H data points. Many these studies relied on lumped parameter convolution integral approaches with time-invariant TTDs whose pre-defined functional form when applied with seasonally variable tracers was limited to shapes (e.g. exponential) that already *a priori* precluded the representation of heavy-tails and thus a meaningful representation of old ages. In addition, the models to estimate water ages in these studies were implemented in a spatially lumped way, which further exacerbates the potential for underestimating water ages due to spatial aggregation effects in environments that are likely subject to considerable heterogeneity in hydrological functioning (Kirchner, 2016).

Addressing some of the concerns above, a recent study by Rodriguez et al. (2021) compared catchment water ages inferred from two-year data records of a seasonally variable tracer ($^2$H; 1088 data points) and $^3$H (24 data points) using a spatially lumped implementation of a previously developed simple tracer circulation model based on the SAS approach, which generates time-variable TTDs (Rodriguez and Klaus., 2019). In spite of consistently higher age estimates obtained from $^3$H, the absolute differences to $^2$H inferred estimates were very minor. While the difference in mean transit times was estimated at $\Delta MTT \sim 0.22$ years for MTTs $\sim 3$ years, the difference in the estimate of the 90[th] percentile of water ages, as metric for the presence of old ages, was with $\Delta 90^{th} \sim 0.15$ years even lower. The authors concluded that these results cast some doubt on "[…] the perception that stable isotopes systematically truncate the tails of TTDs" (Rodriguez et al., 2021). However, their interpretation was questioned by Stewart et al. (2021), who pointed out that simply no older water may be present in their study catchment.

Building on the above work of Rodriguez et al. (2021), the objective of this study is therefore to further scrutinize the notion that the use of seasonally variable tracers leads to truncated estimates of water age distributions in a systematic comparative experiment. The novel aspects of this study for the $\sim$13.000 km$^2$ Neckar River basin in South-West Germany include that we here use (1) long-term records, i.e. > 20 years, of hydrological data as well as of seasonally variable ($^{18}$O) and radioactive tracers ($^3$H) together with (2) a suite of lumped and spatially semi-distributed implementations of (3) SW, CO and SAS-function based models, including a formulation of an integrated, process-based model to simultaneously reproduce hydrological and tracer response dynamics and to track temporally variable water age distributions in the system. The above points allow us to, at least partially, explore several unresolved questions how different factors may or may not contribute to the apparent underestimation of water ages by seasonally variable tracers, including potential effects of uncertainties arising from short data records, spatial aggregation and the use of oversimplified time-invariant, lumped models. More specifically, we here test the hypothesis that $^{18}$O as tracer generally and systematically cannot detect tails in water age distributions and that this truncation leads to systematically younger water age estimates than the use of $^3$H.

**2 Study site**

The Neckar River basin in South-West Germany has an area of $\sim$13,000 km$^2$. The elevation in the basin ranges from 122 m at

the outlet in the north to about 1019 m in the South (Fig. 1a; Table 1). Following the elevation gradient, the landscape is characterized by terrace-like elements and undulating hills with wide valleys used as grass- and croplands in lower regions, in particular in the northern parts of the Neckar Basin, and increasingly steep and narrow forested valleys towards the southern parts (Fig. 1c). Long-term mean annual precipitation (P) reaches ~909 mm yr$^{-1}$, with considerable spatial variability ranging from ~660 mm yr$^{-1}$ in the lower parts of the basin to over 1500 mm yr$^{-1}$ at high elevations in the southwest (Fig. 1b). With a long-term mean temperature of about 8.9 °C, potential evaporation (E$_P$) around ~870 mm yr$^{-1}$ and an aridity index (I$_A$) (i.e., I$_A$ = E$_P$/P) I$_A$ ~0.98 the basin is characterized by a temperate-humid climate, where snow cover can be present for several weeks in the winter months.

## 3 Data

### 3.1 Data

Daily hydro-meteorological data were available for the period 01/01/1970 – 31/12/2016. As the forcing data of the hydrological models, daily precipitation and daily mean air temperature were obtained from stations operated by the German Weather Service (DWD). Precipitation was recorded at 16 stations and temperature measurements were available at 12 stations (Fig. 1) in or close to the study basin. Daily mean discharge data for the period 01/01/1970 – 31/12/2016 at the outlet of the Neckar basin at Rockenau station were provided by the German Federal Institute of Hydrology (BfG). In addition, data of daily mean discharge for the same time period from three sub-catchments within the Neckar basin (Fig.1) at the gauges Kirchentellinsfurt (C1; 2324 km$^2$), Calw (C2; 584 km$^2$) and Untergriesheim (C3; 1827 km$^2$) were available from the Environmental Agency of the Baden-Württemberg region (LUBW).

Long-term volume-weighted monthly δ$^{18}$O data in precipitation was available for the period 01/01/1978 – 31/12/2016 at the Stuttgart station. At the sampling gauge, a monthly accumulation bottle was filled with the collected daily precipitation, and all collected water was mixed together. Therefore, the water samples of precipitation reflect the volume-weighted monthly isotopic composition. Then, a monthly isotope sample bottle for stable isotope (i.e., $^{18}$O) was filled with 50 ml precipitation water from the corresponding monthly accumulation bottle. All precipitation samples were tightly sealed and stored in a dark room at ~4°C before analysis. Monthly stream water samples were collected at Schwabenheim, close to the Rockenau discharge station, by the BfG for the period of 01/10/2001 – 31/12/2016 (Schmidt et al. 2020; Königer et al. 2022). Note that the available data do not represent instantaneous grab samples but bulk samples from mixed daily samples. River water was sampled automatically by samplers (SP III-XY-36, Maxx Meb- und Probenahmetechnik GmbH, Germany), which contained 36 bottles (each with a volume of 2.5 L). Every 30 minutes, 50 ml river water was pumped into one bottle (48 subsamples per day). A new bottle was filled every 24 h with the same procedure. All daily river water samples were stored in the sample compartment at ~4°C and were subsequently combined into monthly samples in the laboratory of BfG. This means the stream water samples reflect a non-flow-weighted monthly average isotopic composition. The stable isotopes ratios were analyzed with dual-inlet mass spectrometry and a laser-based cavity ring-down spectrometer (L2120-i/L2130-i, Picarro Inc.) at Helmholtz Zentrum

München, Germany. When changing from dual-inlet mass spectrometry to cavity ring-down spectrometry, the long-term precision of the analytical systems (±0.15 ‰ and ±0.1 ‰, respectively, for $\delta^{18}O$) was ensured (Stumpp et al. 2014; Reckerth et al., 2017).

Long-term monthly $^{3}H$ data in precipitation were obtained for the period 01/01/1978 – 31/12/2016 at Stuttgart station (same station as $^{18}O$ data in precipitation; Schmidt et al., 2020). For the purpose of establishing robust initial conditions for the model experiment (see section 4.2) the tritium record in precipitation was reconstructed for the preceding 1970-1977 period by bias correcting data from the sampling station Vienna, available from the Global Network of Isotopes in Precipitation which is a joint database of the International Atomic Energy Agency (IAEA) and the World Metrological Organization (WMO) (Supplementary Material Fig. S1). The precipitation for tritium data was sampled based on the same method as that for $^{18}O$ in precipitation which means that the precipitation samples for tritium also reflect the volume-weighted monthly isotopic composition. Stream water samples for tritium were collected based on the same method as that for as $^{18}O$ in stream. Therefore, tritium stream water samples also reflect non-volume-weighted monthly average isotopic compositions. The tritium stream water samples are not influenced by water release from nuclear power stations. All water samples were analyzed for tritium concentrations by the BfG Environmental Radioactivity Laboratory using liquid scintillation counters (Ultima Gold LLT) with a 2-sigma analytical uncertainty (Schmidt et al. 2020).

Land use types of the catchments are determined using the CORINE Land Cover data set of 2018 (https://land.copernicus.eu/pan-european/corine-land-cover). The 90 m × 90 m digital elevation model of the study region (Fig. 1a) was obtained from https://www.usgs.gov/ and used to derive the local topographic indices including height above nearest drainage (HAND) and slope.

## 3.2 Data pre-processing

For the subsequent model experiment (section 4.2), the study basin was stratified into four regions P1 – P4 that are characterized by distinct long-term precipitation pattern (hereafter: precipitation zones). In the following the procedure to infer these precipitation zones and to estimate the associated differences in $\delta^{18}O$ and $^{3}H$ input is described.

### 3.2.1 Spatial distribution of precipitation and identification of precipitation zones

To account, at least to some degree, for spatial heterogeneity in precipitation we stratified the Neckar River basin into precipitation zones that are each characterized by distinct average annual precipitation totals. Goovaerts (2000) and Lloyd (2005) showed that areal precipitation estimates informed by elevation data were often more accurate than those based on precipitation gauge observations alone. Thus, to interpolate and to estimate areal precipitation across the basin we used Co-Kriging, considering elevation, as a preliminary analysis suggested lower errors. Finally, the individual precipitation estimates for each grid cell were used with K-means clustering to establish four clusters, representing the four precipitation zones P1 – P4 (see Fig. 1b).

### 3.2.2 Spatial extrapolation of precipitation $\delta^{18}O$ to precipitation zones

Records of observed precipitation $\delta^{18}O$ are available at one location close to the center of the Neckar Basin (Fig. 1). However, it is well described (e.g. Kendall and Mcdonnell, 2012) that precipitation $\delta^{18}O$ input can be subject to considerable spatial heterogeneity, largely controlled by topographic and meteorological influences. Stumpp et al. (2014) specifically identified latitude, elevation and temperature as the key factors controlling $\delta^{18}O$ input heterogeneity in the greater study region. To at least partially account for these effects and to locally adjust $\delta^{18}O$ input signals throughout the study basin, we made use of the sinusoidal isoscapes method (Allen et al., 2018, 2019). Briefly, this method exploits the seasonal pattern in $\delta^{18}O$ precipitation signal by fitting sine functions to observed $\delta^{18}O$ input signals for a large sample of locations:

$$\delta^{18}O_P(t) = a_P \sin(2\pi t - \varphi_P) + b_P, \tag{1}$$

With $a_P$ [‰] the amplitude of the seasonal precipitation signal, $b_P$ [‰] a constant offset and $\varphi_P$ [rad] the phase of the signal. For each of the three fitting parameters, i.e., $a_P$, $b_P$ and $\varphi_P$, multiple regression relationships were previously developed (Allen et al., 2018). Depending on the fitting parameter, predictor variables included a selection of latitude, longitude, elevation, range of annual temperature range and mean annual precipitation (Allen et al., 2018). The relationships defined by these predictor variables then allow to estimate $a_P$, $b_P$ and $\varphi_P$, and thus the seasonal signal of $\delta^{18}O_P$ for locations where no precipitation $\delta^{18}O$ observations are available.

Here, we adopted the method as described in the following. In a first step, we estimated the sine wave parameters for the time series of precipitation $\delta^{18}O$ observed at the station Stuttgart, using the procedure described by Allen et al. (2018). Subsequently, we estimated the associated sine wave parameters $a_P$, $b_P$ and $\varphi_P$ in each of the four precipitation zones (P1 – P4; Supplementary Material Table S2) based on Eqs. (S1) - (S3) in the Supplement, using the above-described individual predictor variables, averaged for each precipitation zone (Supplementary Material Table S1). We then used the estimated sine wave parameters to construct an individual $\delta^{18}O_P$ sine wave for each precipitation zone (Eq.1). In a last step, we adjusted the observed $\delta^{18}O$ input for the four precipitation zones by rescaling and bias correcting the observed $\delta^{18}O$ signal according to the differences between the sine waves at the observation station and sine waves estimated for each precipitation zone, respectively (Supplementary Material Fig. S2).

### 3.2.3 Spatial extrapolation of precipitation $^3H$ to precipitation zones

As for $\delta^{18}O$, it is well documented that $^3H$ exhibits spatial heterogeneity that is to some extent controlled by geographical factors. It has been shown that the $^3H$ concentration in precipitation increases with latitude, with highest concentrations in polar regions (Rozanski et al., 1991). In addition, $^3H$ concentrations in precipitation increase with elevation due to the $^3H$-enriched upper troposphere and isotopic exchange between liquid water and atmospheric moisture, depleting $^3H$ in lower tropospheric layers (Tadros et al., 2014). Considering the above effects, we established a multiple linear regression relationship between $^3H$ concentrations in precipitation observed at 15 multiple locations across Germany (Supplementary Material Fig. S3) as available through the WISER database (IAEA and WMO, 2022; Schmidt et al., 2020), and their corresponding elevation

and latitude, respectively (Supplementary Material Fig. S4). We then used this relationship to adjust the $^3$H precipitation input for the four precipitation zones according to their corresponding average latitude and elevation estimate:

$$^3H_P(t) = -0.75(L_P - L_o) - 0.002(E_P - E_o) + \ ^3H_o, \tag{2}$$

where $^3H_P$ is the latitude- and elevation-adjusted tritium precipitation concentration for each precipitation zone (P1 – P4), $^3H_o$ is the tritium precipitation concentration observed at the Stuttgart station, $L_P$ and $E_P$ are the mean latitude and elevation, respectively, of each precipitation zone and $L_o$ and $E_o$ are the latitude and elevation, respectively, of the Stuttgart station.

## 4 Methods

The experiment to test the hypothesis that the use of $\delta^{18}$O data systematically leads to truncated water age distributions and associated underestimations of water ages is designed and executed in a step-wise approach. 21 different scenarios of model types and spatial implementations thereof are sequentially calibrated and tested to reproduce observed $\delta^{18}$O and $^3$H signals in stream flow. For each of these models, several metrics of water age distributions resulting from the 2 independent calibration procedures, i.e., for $\delta^{18}$O and $^3$H, respectively, are then estimated and compared. As a baseline and to ensure comparability with previous studies, water ages are quantified with spatially lumped, time-invariant implementations of twelve commonly used SW/CO model scenarios (Table 2): sine-wave models using exponential (SW-EM) and gamma distributions as TTDs (SW-GM; only $\delta^{18}$O), lumped parameter convolution integral models using exponential (CO-EM) and gamma distributions as TTDs (CO-GM), two parallel reservoirs (CO-2EM), three parallel reservoirs (CO-3EM) as well as an exponential piston flow (CO-EPM) implementation. The above baseline scenarios are complemented by nine additional models on the basis of SAS-functions (Table 3). In order of increasing complexity, these include three spatially integrated formulations of a "pure" SAS-function approach with one storage component and based on observed stream flow (P-SAS), three implementations of a spatially integrated hydrological model with tracer routing based on SAS-functions (IM-SAS-L) as well as three spatially distributed implementations of the same integrated hydrological model in combination with SAS-functions (IM-SAS-D).

### 4.1 Models

### 4.1.1 Sine-wave model (SW)

As demonstrated by Małoszewski et al. (1983), sine waves fitted to $\delta^{18}$O precipitation and stream flow signals can be used to indicatively determine water ages. More specifically, the ratio of the amplitudes of the fitted sine waves, i.e. $A_s/A_p$, can be used together with the assumption of a shape of the TTD to estimate the associated MTT of a system. In the case of a gamma distribution as TTD, this is done according to (Kirchner, 2016):

$$\bar{\tau} = \alpha\beta, \tag{3}$$

with

$$\beta = \frac{1}{2\pi f}\sqrt{\left(A_s/A_p\right)^{-2/\alpha} - 1},\tag{4}$$

where $\bar{\tau}$ is the MTT, $\alpha$ is a shape parameter, $\beta$ is a scale parameter and f here is the frequency for the seasonal $\delta^{18}O$ signal, i.e., $f = 1$ yr$^{-1}$. Here we analyze the two cases $\alpha = 1$ (SW-EM) and 0.5 (SW-GM). Note that with $\alpha = 1$, the gamma distribution is equivalent to an exponential distribution. The sine wave model is a simplification of a convolution integral model and can be directly derived from that. For a more detailed description of the method and underlying assumptions we refer to McGuire and McDonnell (2006) and Kirchner (2016).

### 4.1.2 Time-invariant, lumped parameter convolution integral model (CO)

While the sine wave approach requires regular cyclic signals of tracer composition, i.e., sine waves fitted to the observations, convolution integral models make direct use of the observed tracer data (e.g. Kreft and Zuber, 1978). Tracer composition in the system output can thus be estimated based on a convolution operation of the tracer composition in the system input together with an *a priori* assumption of a TTD (e.g. Maloszewski and Zuber, 1982; Kirchner et al., 2001):

$$C_o(t) = \int_0^\infty g(\tau)C_i(t-\tau)e^{-\lambda\tau}\,d\tau,\tag{5}$$

Where $C_o(t)$ is the tracer composition of the system output (here: stream flow) at time t, $C_i(t-\tau)$ is the tracer composition of the system input (here: precipitation) at any previous time $t - \tau$, $\lambda$ is the radioactive decay constant ($\lambda = 0.00015$ d$^{-1}$ for $^3H$ and $\lambda = 0$ d$^{-1}$ for stable isotopes) and $g(\tau)$ is the distribution of transit times $\tau$. Here, we used gamma distributions as basis for a flexible and general formulation of TTDs in the different CO scenarios tested in this study:

$$g(\tau) = \sum_{i=1}^{N} \eta f_i \frac{\tau^{\alpha-1}}{\beta_i{}^\alpha \Gamma(\alpha)} e^{\left(\frac{-\tau}{\eta\beta_i} + \frac{1}{\eta} - 1\right)} \qquad \text{for } \tau \geq \tau_m(1-\eta),\, g(\tau) = 0 \text{ otherwise}\tag{6}$$

With the $\alpha$ and $\beta_i$ being the shape and scale parameters, respectively, $f_i$ the fraction of the contribution of the $i^{th}$ reservoir, so that $\sum f_i = 1$ and $\eta$ the ratio of the exponential volume to the total volume. For a single exponential TTD (CO-EM) with $\alpha = 1$, $N = 1$, $\eta = 1$ and $f_1 = 1$, $\beta_1$ was the only calibration parameter. The two parallel exponential TTD model (CO-2EM) with $\alpha = 1$, $N = 2$, $\eta = 1$ and $f_2 = 1 - f_1$, required $\beta_1$, $\beta_2$ and $f_1$ as calibration parameters, while the three parallel exponential TTD model (CO-3EM) with $\alpha = 1$, $N = 3$, $\eta = 1$ and $f_3 = 1 - f_1 - f_2$, required $\beta_1$, $\beta_2$, $\beta_3$ as well as $f_1$ and $f_2$ as calibration parameters. The exponential piston flow model (CO-EPM) with $\alpha = 1$, $N = 1$ and $f_1 = 1$ was characterized by the two calibration parameters $\beta_1$ and $\eta$. In contrast, the Gamma distribution model (CO-GM), with $N = 1$, $\eta = 1$ and $f_1 = 1$, used both, $\alpha$ and $\beta_1$ as free calibration parameters.

The MTTs associated with the above parameters in the individual model implementations are then obtained with Eq. (7).

$$\bar{\tau} = \sum_{i=1}^{N} f_i \alpha \beta_i \qquad\qquad (7)$$

For more detailed description of the method and the individual shapes of TTDs considered here, refer to McGuire and McDonnell (2006).

### 4.1.3 SAS-function models (P-SAS, IM-SAS)

The storage-age selection function (SAS) concept as outlined by Rinaldo et al. (2015) requires the explicit tracking of water and tracer storage volumes. The age compositions of water fluxes are then sampled from the age composition in the associated storage volume. Two alternative and frequently used approaches to account for the evolution of water storage volumes were explored here: firstly, a "pure" SAS-function model in which the observed stream flow was used to account for changes in water storage volumes (P-SAS) and secondly, an integrated process-based hydrological model that generates stream flow and other fluxes in the system (IM-SAS). Water ages, their distributions, and the associated moments thereof were then estimated by tracking water and tracer fluxes through the models.

*Hydrological model*

The hydrological component of the "pure" SAS-function model (P-SAS) was implemented as described in Benettin et al. (2017). This model consists of one single storage volume, which receives observed precipitation P as input and releases observed stream flow as output. Evaporation $E_A$ from that storage is modelled following the simplifying assumption that there is negligible storage change over the entire 47-year study period (01/01/1970 – 31/12/2016), as expressed by:

$$E_A(t) = E_p(t)\left(\frac{\bar{P}-\bar{Q}}{\bar{E}_p}\right) \qquad\qquad (8)$$

With $\bar{P}$ and $\bar{Q}$ being long-term mean daily precipitation P (mm d$^{-1}$) and discharge Q (mm d$^{-1}$), respectively, and $\bar{E}_p$ the long-term mean daily potential evaporation $E_p$ (mm d$^{-1}$).

In contrast, the water storage fluctuations and fluxes in the IM-SAS approach were modelled based on a previously developed, process-based model, based on the DYNAMIT modular modelling scheme (Hrachowitz et al., 2013, 2021). Briefly, this hydrological model consists of a suite of storage components and associated water fluxes between them. The influence of functionally different landscape elements, i.e. forest, grass-/cropland and flat valley bottoms, for brevity hereafter referred to as wetland, is represented by parallel hydrological response units (HRU), linked by a common storage component representing the groundwater system (Fig. 2), as previously implemented and successfully tested in many contrasting environments (e.g. Gao et al., 2014; Gharari et al., 2014; Euser et al., 2015; Nijzink et al., 2016; Prenner et al., 2018; Hanus et al., 2021). Briefly, precipitation P (mm d$^{-1}$) falling on days with temperatures below threshold temperature $T_t$ (ºC), is accumulated as snow $P_{snow}$ (mm d$^{-1}$) in the snow storage $S_{snow}$ (mm). On days with temperatures higher than that, precipitation enters the system as rainfall $P_{rain}$ (mm d$^{-1}$) and, based on a simple degree-day approach, water is released from $S_{snow}$ as snow melt $M_{snow}$ (mm d$^{-1}$), controlled by melt factor $C_{melt}$ (mm d$^{-1}$ ºC$^{-1}$; e.g. Gao et al., 2017; Girons Lopez et al., 2020). Rain water is then routed through the interception storage $S_i$ (mm). With $E_i$ (mm d$^{-1}$) as interception evaporation at the potential evaporation rate, effective

precipitation $P_{re}$ (mm d$^{-1}$) generated by overflow once the maximum interception capacity ($S_{imax}$) is exceeded, together with $M_{snow}$, enters the unsaturated root-zone $S_u$ (mm). From $S_u$ water can then be released as vapor via a combined soil evaporation and transpiration flux $E_a$ (mm d$^{-1}$). Drainage of liquid water from $S_u$ can either recharge the groundwater $S_s$ (mm) over a percolation flux $R_{perc}$ (mm d$^{-1}$) and a faster preferential recharge $R_{pref}$ (mm d$^{-1}$). Alternatively, it can be routed via $R_{uf}$ (mm d$^{-1}$) to a faster responding component $S_f$ (mm) from where it is directly released to the stream as $Q_f$ (mm d$^{-1}$), representing lateral preferential flow. Rain and snow melt entering the wetland HRU directly reach $S_u$. Soil moisture levels in the wetland $S_u$ are further sustained by a fraction of groundwater $R_{cap}$ (mm d$^{-1}$) that is upwelling into $S_u$ from $S_s$ (e.g., Hulsman et al., 2021a). The detailed equations of the model are provided as Table S3 in the Supplementary Material.

*Tracer transport model*

$\delta^{18}O$ and $^3H$ were routed through the above-described storage components of both the P-SAS and the IM-SAS (Fig. 2) models by sampling the observed (i.e. Q in P-SAS) and modeled outflow volumes (i.e. $E_a$ in P-SAS; all outflows in IM-SAS) that leave the individual components at each time step t (d) (e.g. $M_{snow}$, $R_{perc}$, $E_a$, etc.) from the individual water volumes of different age T (d) that are stored in the associated storage component (e.g. $S_{snow}$, $S_u$, etc.) at each time step according to a SAS function. The distribution of water volumes of different ages in each storage component, i.e., the residence time distribution RTD, depends on the past sequence of inflows I (mm d$^{-1}$) and outflows O (mm d$^{-1}$) and therefore varies over time. As a consequence of being sampled from RTDs that evolve over time, both, inflows I and outflows O are correspondingly characterized by water age distributions (or transit time distributions TTD) that change over time. A straightforward implementation of this SAS concept is facilitated by the formulation of age-ranked storages $S_T(T,t)$ (mm). As emphasized by Benettin et al. (2017), $S_T(T,t)$ describes "at any time t the cumulative volumes of water in a storage component as ranked by their age T". Correspondingly, the total inflow (I) into as well as the total outflow volumes (O) from different storages can be expressed in terms of their cumulative, age-ranked volumes $I_T(T,t)$ and $O_T(T,t)$ (mm d$^{-1}$). At any time, closing the resulting water age balance for each storage component j (e.g. $S_{snow}$, $S_u$, etc.) also leads to an updated age-ranked storage $S_{T,j}(T,t)$ for that component, formulated as (Benettin et al., 2015a; Botter et al., 2011; Harman, 2015; Van Der Velde et al., 2012):

$$\frac{\partial S_{T,j}(T,t)}{\partial t} + \frac{\partial S_{T,j}(T,t)}{\partial T} = \sum_{n=1}^{N} I_{T,n,j}(T,t) - \sum_{m=1}^{M} O_{T,m,j}(T,t), \qquad (9)$$

Where $\partial S_T/\partial T$ is the aging process of water in storage. Here, the water age balance (Eq.7) was formulated individually for each storage reservoir j, also accounting for different numbers N of storage component inflows I (e.g. $P_{rain}$, $M_{snow}$, $R_{perc}$) and numbers M of outflows O (e.g., $R_{perc}$, $R_{pref}$, $E_a$) (Fig. 2), similar to previous studies (e.g. Hrachowitz et al., 2021). For a daily modelling time step, it can in the water age balance be assumed that precipitation P(t) that is falling on day t is characterized by an age T = 0. This implies for the age ranked inflow $I_{T,P,j}(0,t) = P_T(0,t) = P(t)$. Note, that all other age ranked inflows $I_{T,n,j}(T,t)$ that enter a storage component are equivalent to the corresponding age ranked outflows $O_{T,m,j}(T,t)$ that leave a "higher" storage component.

Depending on the total volume of outflow $O_{m,j}(t)$ and the cumulative distribution of ages $P_{o,m,j}(T,t)$ of that flow, an age-ranked

outflow $O_{T,m,j}(T,t)$ for each flux m released from each storage component j can be defined as:

$$O_{T,m,j}(T,t) = O_{m,j}(t)P_{o,m,j}(T,t), \tag{10}$$

While the outflow $O_{m,j}(t)$ from any storage component j is computed for each time step t by the hydrological model described above, the associated $P_{o,m,j}(T,t)$ cannot be assumed to be known as it is controlled by the temporally evolving distribution of water ages present in that storage component $S_{T,j}(T,t)$ at t. However, the temporally variable $P_{o,m,j}(T,t)$ can be inferred for each time step t by defining for each storage j and for each outflow m released from j a SAS function $\omega_{o,m,j}$ together with its cumulative form $\Omega_{o,m,j}$. These functions then describe how the water volumes of different ages, stored in component j at time 390 t, i.e. $S_{T,j}(T,t)$, are sampled and combined into the corresponding total outflow volume $O_{m,j}(t)$:

$$P_{o,m,j}(T,t) = \Omega_{o,m,j}\big(S_{T,j}(T,t),t\big), \tag{11}$$

The probability density function $p_{o,m,j}(T,t)$ associated with the cumulative distribution of ages $P_{o,m,j}(T,t)$, then represents the transit time distribution TTD of that outflow and can be written as:

$$p_{o,m,j}(T,t) = \varpi_{o,m,j}\big(S_{T,j}(T,t),t\big)\frac{\partial S_{T,j}}{\partial T}, \tag{12}$$

Conservation of mass dictates that

$$\Omega_{o,m,j}\big(S_{T,j}(T,t) \to S_j(t),t\big) = 1, \tag{13}$$

Where $S_j$ (mm) is the total volume of water stored in component j at time t. The resulting need to rescale $\omega_{o,m,j}$ for each time step was here avoided by instead normalizing and therefore bounding the age ranked storage to the interval [0,1] according to

400 $$S_{T,norm,j}(T,t) = \frac{S_{T,j}(T,t)}{S_j(t)}, \tag{14}$$

Note that $S_{T,norm,j}$ also represents the RTD of storage component j at time t.

For the P-SAS model implementation in this study, we used power law distributions with one parameter to sample streamflow ($k_Q$) and evaporation ($k_E$), respectively, as described by Benettin et al. (2017). In contrast, we used uniform distributions in the form of $\omega$ = const. as SAS function in each storage component in the IM-SAS model implementations as previously shown to 405 be effective in many studies (e.g. Birkel et al., 2011; van der Velde et al., 2015; Benettin et al., 2015b, 2017; Ala-Aho et al., 2017; Kuppel et al., 2018; Rodriguez et al., 2018). The latter implies random sampling and the assumption that each storage component is fully mixed and that there is no preference for sampling younger or older water. However, note that due to distinct storage capacities and time-scales of the individual storage components, the "combined" SAS functions of all storage components will *not* lead to an overall fully mixed system response. Uniform SAS functions were here chosen over other 410 shapes, such as beta-distributions (e.g. van der Velde et al., 2012; Hrachowitz et al., 2021), as they do not need additional model parameters and avoid the need for explicit calculation of TTDs at each model time step to route tracers through the

model (Benettin et al., 2015b), thereby drastically reducing computer memory requirements and computational time (Benettin et al., 2022).

To adequately damp tracer input signals, suitable system storage volumes have to be defined as calibration parameters. In the P-SAS implementation the parameter $S_{tot}$ is used, reflecting the initial total system storage (e.g. Benettin et al., 2017). In contrast, the IM-SAS implementations made use of additional and hydrologically passive storage volumes (e.g. Christophersen and Wright, 1981; Birkel et al., 2010; Hrachowitz et al., 2015, 2016), which physically represents groundwater volumes below the river bed, as illustrated by Zuber (1986; Fig.1 therein). Such a passive water storage volume $S_{s,p}$ (mm), characterized by $dS_{s,p}/dt = 0$, was thus added as calibration parameter to the active groundwater storage $S_s$ (Fig. 2). While the outflow $Q_s$ from the groundwater storage is exclusively regulated by the temporally varying storage volume in $S_s$ (Supplementary Material Eq. S9), the tracer and age composition of that outflow is also randomly sampled from the total groundwater storage volume $S_{s,tot} = S_s + S_{s,p}$.

The $\delta^{18}O$ and $^3H$ concentrations were then routed through each individual storage component according to (e.g. Harman, 2015; Benettin et al., 2017):

$$C_{o,m,j}(t) = \int_0^{S_j} C_{s,j}(S_{T,j}(T,t),t)\omega_{o,m,j}(S_{T,j}(T,t),t)e^{-\lambda T}\, dS_T, \tag{15}$$

Where $C_{o,m,j}$ is the tracer concentration in outflow m from storage component j at time t, $C_{s,j}$ is the tracer concentration of water in storage at time t and $\lambda$ is the radioactive decay constant ($\lambda = 0$ d$^{-1}$ for $\delta^{18}O$ and $\lambda = 0.00015$ d$^{-1}$ for $^3H$).

## 4.2 Model implementation

### 4.2.1 Spatially lumped model implementation

The original argument that the use of seasonally variable tracers' underestimates water ages was exclusively based on lumped, time-invariant implementations of sine-wave and convolution integral models (Stewart et al., 2010). For a baseline comparison and to check whether the above conclusion would also have been reached for our study basin using the same methods, we here similarly implemented the sine-wave (SW-EM, SW-GM) and convolution integral (CO-EM, CO-GM, CO-2EM, CO-3EM, CO-EPM) in a spatially lumped way. For this baseline case the catchment average tracer input was estimated as the spatially weighted mean from the four precipitation zones P1 – P4 as described in section 3.2. The calibration parameters of the CO implementations are shown in Table 2.

The "pure" SAS-model (P-SAS; Table 3) and the spatially lumped implementation of the integrated model (IM-SAS-L) were also forced with the same spatially averaged input. In addition, the spatial fractions of the grassland and wetland HRUs for IM-SAS-L, respectively, were set to 0 and the entire study basin therefore represented by one HRU which is equivalent to the forest HRU described in distributed model, similar to many traditional lumped formulations of process-based conceptual models (Bouaziz et al., 2021; Clark et al., 2008; Fenicia et al., 2006; Fovet et al., 2015; Seibert et al., 2010). This implementation has 11 calibration parameters (Table 3).

### 4.2.2 Spatially distributed model implementation

To balance the need for spatial detail to some extent with the adverse effects of increased parameter uncertainty (e.g. Beven, 2006) and computational capacity (in particular for the calculation of TTDs), we here implemented the integrated model in parallel (IM-SAS-D) in the four precipitation zones P1 – P4 and forced it with the corresponding input (e.g. P, $\delta^{18}O$ and $^3H$) for each precipitation zone as described in section 3.2. Each precipitation zone was further discretized (1) into 100 m elevation zones for a stratified representation of the snow storage $S_{snow}$ (e.g. Mostbauer et al., 2018) and (2) into three HRUs, i.e., forest, grassland, wetland (Fig.2; e.g. Gharari et al., 2014; Hanus et al., 2021). Rain $P_{rain}$ and melt water $M_{snow}$ from the different elevation zones was aggregated according to their associated spatial weights in each elevation zone. This total liquid water input was then routed through the three parallel HRUs. The classification into the three HRUs was based on the metric Height-above-nearest-drainage (HAND; Gharari et al., 2011) and land cover. While landscape elements with HAND < 5 m were classified as wetland, all other parts of the landscape were classified as forest or grassland according to land-use data. In total, there are therefore 12 individual, parallel model components, i.e., three HRUs in each of the four precipitation zones, not counting the elevation zones for the snow module. All flux and storage variables of the 12 components are weighted according to their areal fractions. While each of the three HRUs was characterized by individual parameters (e.g. Gao et al., 2016; Prenner et al., 2018), the same parameter values were used in all four precipitation zones in distributed moisture accounting approach (e.g. Ajami et al., 2004; Euser et al., 2015; Hulsman et al., 2021b; Roodari et al., 2021). Overall, the spatially distributed implementation has 19 model parameters, including five global parameters ($T_t$, $C_{melt}$, $C_a$, $K_s$ and $S_{s,p}$) that are identical for each HRU and 14 HRU-specific parameters (Table 3; Fig.2).

### 4.3 Model calibration and post-calibration evaluation

The models were run at a daily time step, whereby the observed volume-weighted monthly tracer concentration in precipitation was used as model input for each day of that month together with the daily data of precipitation. Model performance was evaluated based on the Mean Square Error (MSE) as error metric. The time-invariant, lumped convolution integral models, using uniform prior parameter distributions as shown in Table 2, were individually calibrated to the observed $\delta^{18}O$ (calibration strategy $C_{\delta^{18}O}$; Table 2) and $^3H$ stream water concentrations ($C^{3H}$), respectively. In contrast, a multi-objective calibration approach was applied for the integrated IM-SAS models to simultaneously reproduce stream flow volumes and tracer concentrations thereof (e.g. $^3H$ and/or $\delta^{18}O$). Briefly, the model parameters were calibrated by using Borg_MOEA algorithm (Borg Multi-objective evolutionary algorithm; Hadka and Reed, 2013) and based on uniform prior distributions (Table 3). The model performances were evaluated based on the models' ability to simultaneously reproduce multiple signatures of stream flow as well as signatures of tracer dynamics as shown in Table 3. The sets of pareto optimal solutions obtained from the calibration procedures were then retained as acceptable solutions for the subsequent analysis. To compare the water age distributions (i.e., TTDs and RTDs) and thus to test the research hypothesis, different calibration strategies – $C_{\delta^{18}O,Q}$, $C^{3H,Q}$ and $C_{\delta^{18}O,^3H,Q}$ – were adopted (Table 3). While in strategy $C_{\delta^{18}O,Q}$ the models were calibrated to simultaneously reproduce signatures

of stream flow and $\delta^{18}O$, $C^{3}_{H,Q}$ combined the stream flow signatures with $^{3}H$. In strategy $C_{\delta^{18}O,^{3}H,Q}$ the model was finally calibrated to simultaneously reproduce the six stream flow signatures, $\delta^{18}O$, *and* $^{3}H$ dynamics. For each strategy, all performance metrics were also combined into an overall performance metric based on the Euclidian distance ($D_E$), where $D_E$ = 0 indicates a perfect fit. To find a somewhat balanced solution in absence of more detailed information all individual performance metrics were here equally weighted (e.g., Hrachowitz et al., 2021; Hulsman et al., 2021b):

$$D_E = \sqrt{\frac{1}{2}\left(\frac{\sum_{n=1}^{N}\left(E_{MSE,Q,n}\right)^2}{N} + \frac{\sum_{m=1}^{M}\left(E_{MSE,tracer,m}\right)^2}{M}\right)},$$  (16)

Where $N = 6$ is the number of performance metrics with respect to stream flow ($E_{MSE,Q,n}$) and $M$ is the number of performance metrics for tracers ($E_{MSE,tracer,m}$) in each combination (e.g. $M=1$ for $C_{\delta^{18}O,Q}$, and $C^{3}_{H,Q}$, $M=2$ for $C_{\delta^{18}O,^{3}H,Q}$). Note

that the different units and thus different magnitudes of residuals introduce some subjectivity in finding the most balanced overall solution according to $D_E$ (Eq. 16). However, a preliminary sensitivity analysis with varying weights for the individual performance metrics in $D_E$ suggested limited influence on the overall results and is thus not further reported here.

After a warm-up period 01/01/1978 – 30/09/2001 the models were calibrated for the 01/10/2001 – 31/12/2009 period. The calibration period was chosen so that observations of all three calibration variables, i.e., Q, $^{3}H$ and $\delta^{18}O$, are available for the

490 entire calibration period to allow a consistent comparison. The long model warm-up period was deemed necessary to meaningfully approximate the model initial conditions due to the potential and *a priori* unknown relevance of old water in the study basin, and thus to avoid underestimation of water ages inferred from $^{3}H$ data. The pareto optimal solutions (parameter sets) of the Neckar basin model were then used to test the model in the post-calibration evaluation period 01/01/2010 – 31/12/2016. In addition, the model was tested for its ability to represent spatial differences in the hydrological response by

495 evaluating it against streamflow observations in three sub-catchments (C1 – C3) of the Neckar without further re-calibration whereby each one of them largely represents the hydrological response from one of the precipitation zones (Fig. 1). The water age distributions, i.e., TTDs and RTDs, extracted from the individual models and calibration strategies were then estimated based on the corresponding sets of pareto optimal solutions obtained for each calibration strategy.

**5 Results**

**5.1 Model performance**

The stream tracer responses of the lumped baseline models were found to be broadly consistent with the available observations (Table 4). For the SW models (scenarios 1, 2) in particular the sine wave fitted to the stream water $\delta^{18}O$ observations provides a robust characterization of the observed signal with $MSE_{\delta^{18}O}$ = 0.121 and 0.144 ‰ for calibration and model evaluation periods, respectively (Supplementary Material Fig. S5). Similarly, the CO models (scenarios 3, 5, 7, 9, 11) reproduced the

overall pattern of seasonal fluctuations and the degree of dampening of the $\delta^{18}O$ response (Supplementary Material Fig. S6). The best performing model, the CO-3EM model, was characterized by $MSE_{\delta^{18}O} = 0.171$ and $0.191$ ‰ for the calibration and model evaluation periods, respectively while, in comparison, the CO-EM implementation with exhibited considerably higher errors with $MSE_{\delta^{18}O} = 0.327$ and $0.432$ ‰ (Table 4). When used with $^3H$ data (scenarios 4, 6, 8, 10, 12), the CO models do capture the general decrease in the magnitude of stream water $^3H$ concentrations although fluctuations at shorter timescales are not well reproduced (Supplementary Material Fig. S7). The CO-2EM model gives the best performance with $MSE_{^3H} = 5.171$ and $3.964$ $TU^2$ for the calibration and evaluation periods, respectively, while the CO-EPM model resulted in $MSE_{^3H} = 5.926$ and $5.115$ $TU^2$ (Table 4). It is also noted that the models already mimic the $^3H$ response well in the 1978 – 2001 pre-calibration model warm-up period.

The P-SAS implementations (scenarios 13 – 15; Table 5; Fig. 3a – d and Fig. 4a – d) show a somewhat higher skill to reproduce the dampening of $\delta^{18}O$ response with $MSE_{\delta^{18}O} = 0.069 – 0.078$ ‰ for the calibration and $0.215 – 0.231$‰ for the evaluation periods, respectively, as well as the general decrease in the magnitude of stream water $^3H$ with $MSE_{^3H} < 3$ $TU^2$. In contrast to the above, the implementations of the integrated model IM-SAS (Table 5) aim to not only to reproduce the $\delta^{18}O$ or $^3H$ stream signals, but to additionally and simultaneously describe the hydrological response (Table 5). Both, the lumped IM-SAS-L (scenario 16; Supplementary Material Fig. S8a, b) and the distributed IM-SAS-D (scenario 19; Fig. 3e, f) reproduce the seasonal fluctuations as well as the degree of dampening of the $\delta^{18}O$ signals with $MSE_{\delta^{18}O} = 0.079 – 0.083$ ‰ for the calibration and $0.273 – 0.332$ ‰ for the evaluation periods similar to or better than the baseline SW/CO models. The IM-SAS models do also describe the evolution of the $^3H$ stream signals rather well (scenarios 17 and 20). With $MSE_{^3H} < 3$ $TU^2$, IM-SAS-L (Supplementary Material Fig. S9) and IM-SAS-D (Fig. 4e – h) do not only outperform the baseline models with respect to the overall magnitude of $^3H$, but do, in spite of somewhat underestimating the magnitude of seasonal amplitudes, also provide a better representation of these intra-annual fluctuations. Similar to the SW/CO baseline models, both the P-SAS and IM-SAS implementations also very well capture the overall decline of the stream water $^3H$ levels in the 1978 – 2001 pre-calibration model warm-up period. The simultaneous calibration to the hydrological response and the $\delta^{18}O$ *and* $^3H$ stream signals (scenarios 18 and 21) led to a comparable model skill to reproduce the tracer signals. In addition to the tracer concentrations, all IM-SAS implementations do also reproduce the main features of the hydrological response (Table 5). More specifically, the modelled hydrographs in particular describe well the timing of peaks as well as the shape of recessions, although in some cases peak flows were underestimated and low flows overestimated as shown for scenario 21 in Figure 5 (for scenarios 16 – 20 see Supplementary Material Figs. S10 – S14). The resulting in $MSE_Q$ remains $\leq 0.336$ $mm^2$ $d^{-2}$ across all IM-SAS implementations (scenarios 16 – 21). Crucially, the models also reproduce well the other observed stream flow signatures such as the flow duration curves ($MSE_{FDCQ} \leq 0.047$ $mm^2$ $d^{-2}$; Fig. 5d), the seasonal runoff coefficients ($MSE_{RC} \leq 0.008$; Fig. 5e) and the autocorrelation functions ($MSE_{ACQ} \leq 0.007$; Fig. 5f). The model, calibrated on the overall response of the Neckar basin, also exhibited considerable skill to represent spatial differences in the hydrological response by reproducing observed stream flow in the three sub-catchments (C1 – C3) similarly well (Fig.6) without any further re-calibration.

## 5.2 Model parameters

Parameters of the SW/CO baseline models (scenarios 1 – 12) directly define the shapes of parametric TTDs and thus the associated metrics of water age, such as MTT following Eqs. (3 – 7). The CO models representing $^3$H signals (scenarios 4, 6, 8, 10, 12) are characterized by values of parameters $\beta_1$, $\beta_2$ and $\beta_3$ that are by a factor of up to ~ 10 higher than the same parameters of models calibrated to $\delta^{18}$O signals (Table 2). For example, $\beta_1 = 513$ d for the CO-EM in scenario 3 and 3795 d in scenario 4.

The individual parameters of the P-SAS and IM-SAS model implementations (scenarios 13 – 21), in contrast, do not directly define parametric TTDs nor can they be readily and directly be linked to water ages. However, it has been previously shown that the sizes of water storage volumes is an important control on water ages (e.g. Harman, 2015) and that in particular total storage volumes, represented by parameter $S_{tot}$ in P-SAS, and the hydrologically passive storage volumes, represented by parameter $S_{S,p}$ in IM-SAS models, are key to regulate in particular older water ages in many systems (e.g. Hrachowitz et al., 2016). Calibration of P-SAS to $\delta^{18}$O in scenario 13 suggested $S_{tot}$ ~ 15595 mm while calibration of the lumped IM-SAS-L to $\delta^{18}$O and stream flow ($C_{\delta^{18}O,Q}$) in scenario 16 led to a moderately well identifiable range of this parameter $S_{S,p}$ ~ 4107 – 10029 mm across all pareto optimal solutions and in the same order of magnitude as P-SAS (Fig. 7a, Table 3). Reflecting the water storage capacity in the unsaturated root zone, which is an important control on younger water ages (Hrachowitz et al., 2021), the parameter $S_{umaxF}$ was found to range between ~ 314 – 415mm (Fig. 7b, Table 3) for the same IM-SAS-L scenario. The calibration of the same models to $^3$H (scenarios 14, 17) resulted in a similar parameter ranges for $S_{tot}$ ~ 16638mm, $S_{S,p}$ ~ 3924 – 9339 mm (Fig. 7a) as well as, albeit slightly lower, $S_{umaxF}$ ~236 – 355 mm (Fig. 7b). The similarities between these two scenarios are also reflected in the parameter ranges obtained from the simultaneous calibration to $\delta^{18}$O and $^3$H ($C_{\delta^{18}O,^3H,Q}$) in scenarios 15 and 18. The calibration of the distributed IM-SAS-D model following all the three calibration strategies in scenarios 19 – 21, resulted in values for $S_{S,p}$ ~ 3270 – 9011 mm (Fig. 7c) that are broadly in the similar ranges as for IM-SAS-L ($S_{S,p}$ ~ 3924 – 13676 mm). In contrast, the distinction into the individual HRUs led to clear differences between $S_{umaxF}$, $S_{umaxG}$ and $S_{umaxW}$ (Figs. 7d-f), reflective of the different hydrological functioning of these HRUs. Nevertheless, the area-weighted average of these parameters comes close to the equivalent parameter from the lumped model implementation ($S_{umaxF}$). The general consistency of these parameters obtained from the different calibration strategies is exacerbated by the limited differences in the most balanced solutions (smallest $D_E$) between the different scenarios. For example the most balanced solutions of $S_{S,p}$ fall between ~ 4000 – 5000 mm for all IM-SAS scenarios 16 – 21 (Fig. 7a, c). All other parameters, which are less clearly related to water ages, exhibit different levels of variation across the individual scenarios yet not following any clear and systematic pattern (Table 3).

## 5.3 Water age distributions

Based on a $\delta^{18}$O amplitude ratio $A_s/A_p = 0.21$ (Table 2), the results of the SW models (scenarios 1, 2) suggest a system that is characterized by rather young stream water with MTT ~ 0.7 – 1.8 yr, depending on the choice of TTD (Table 6; Fig. 8). The

TTDs obtained from the CO models calibrated to $\delta^{18}O$ (scenarios 3, 5, 7, 9, 11) are broadly consistent with that, suggesting MTT ~ 1.4 – 2.4 yr. These TTDs suggest mean water ages that are up to ~ 9 yr lower than estimates from CO models calibrated to $^3H$ (scenarios 4, 6, 8, 10, 12) with MTT ~ 9.4 – 10.4 yr (Table 6; Fig. 8). For higher percentiles the differences in water ages can even reach more than 20 years (Table 6). Correspondingly, the fractions of water younger than 3 months, F(T < 3 m), exhibit considerable differences of -2 – 22% points between $\delta^{18}O$ and $^3H$ inferred estimates, which further increase to

differences of 30 – 64% for F(T < 3 yr).

In contrast, from the implementations of the P-SAS and IM-SAS models in scenarios 13 – 21, it can be clearly seen that the stream water ages inferred from $\delta^{18}O$ are across most percentiles by a factor of around 10 higher than those from SW and CO models, resulting in volume-weighted average MTT ~ 11 – 17 yr over the modelling period (Table 7; Fig. 9). Similarly, all water fractions below 20 years are substantially lower for the P-SAS and IM-SAS models than for SW and CO models. The

most pronounced difference is observed at F(T < 5 yr) that reaches 38 – 57% for SAS-functions models and 91 – 100% for SW and CO, which equals to a difference of more than 50%. As such, these water age estimates from $\delta^{18}O$ in SAS-function models (scenarios 13, 16, 19) are not only very similar to the estimates from $^3H$ in these models (scenarios 14, 17, 20) but $\delta^{18}O$ suggests, against the expectations, even slightly *older* water than $^3H$ does. More specifically, while $\delta^{18}O$ results in stream water MTT 11 -17 yr (scenarios 13, 16, 19), the $^3H$-based estimates reach MTT ~ 11 – 13 yr (scenarios 14, 17, 20) and thus up to

five years younger (Table 7; Fig. 9). The differences between $\delta^{18}O$ and $^3H$ water ages from individual P-SAS and IM-SAS model implementations (scenarios 13 – 21) are similar over all percentiles with $\Delta TT_{\delta^{18}O-^3H}$, on average, ~ 1.4 yr and not exceeding ~ 5.5 yr. Accordingly, the fractions of water of any given age up to T < 20 years is ~ 1 – 8 % higher for $^3H$ than for $\delta^{18}O$, suggesting higher fractions of old water modelled with $\delta^{18}O$ (Table 7). Equivalent pattern and comparable magnitudes are found for the combined use of $\delta^{18}O$ and $^3H$ in scenarios 15, 18 and 21.

An explicit comparison between the lumped IM-SAS-L (scenarios 16 – 18) and the distributed IM-SAS-D (scenarios 19 – 21) also suggests a good correspondence between the respective inferred water ages for both tracers. While IM-SAS-L generates MTT ~ 11.2– 17.4 years, the MTT obtained from IM-SAS-D reach ~ 12.8 – 15.6 years (Table 7, Fig. 9). Besides the MTT, also the differences in water ages across all percentiles is minor and reaches a maximum of 4.6 years at the 75th percentile. Accordingly, the fractions of water with ages T < 20 yr exhibit only marginal differences between the lumped (IM-SAS-L) and

distributed model (IM-SAS-D) implementations. It is noted that these overall water ages from IM-SAS-D for the entire Neckar basin emerge from the aggregation of TTDs of the four individual precipitation zones P1 – P4 (Supplementary Material Figure S29-31 and Table S6), which are characterized by pronounced differences with MTT ranging from ~ 8 – 10 years in P4 and ~ 18 – 22 years in P2, depending on the scenario.

The consistency between water ages inferred from $\delta^{18}O$ and $^3H$, respectively, in all SAS-function model scenarios is further

illustrated by the direction and magnitude of change in water age distributions as a consequence of changing wetness conditions. In particular during wet-up and wet periods, a marked variability of daily TTDs can be observed, with young water fractions F(T < 3 m) ranging between ~ 20 – 65% for $\delta^{18}O$-based estimates and ~ 25 – 70% for $^3H$ (Fig. 10a, b, e, f). Less variability in daily TTDs is found under drying and dry conditions with generally F(T < 3 m) in the range of ~ 1 – 20%, with only very few

outliers > 30%. Overall, the volume-weighted average TTDs for wet conditions suggest slightly older water inferred from $\delta^{18}O$ with a median water age of ~ 3 year and F(T < 3 m) ~ 30%, for wet conditions than from $^3H$, for which a median age of ~ 1 year and F(T < 3 m) ~ 40 % were found (Fig. 10d, h). This is in opposite to dry conditions for which the differences between $\delta^{18}O$ and $^3H$-derived water age estimates become mostly negligible (Fig. 10d, h).

With P-SAS and IM-SAS models, not only MTT/TTD in streams can be derived but also in any fluxes/storages (i.e., transpiration flux $E_a$, ground water storage). An even more pronounced young water variability in daily TTDs was found for the transpiration flux $E_a$ leaving the unsaturated root zone storage $S_u$ in the IM-SAS models (scenarios 16 – 21). As shown in Figure 11a, the transpiration TTDs inferred from $\delta^{18}O$ suggest a median transpiration age during wet conditions of ~ 2 – 40 days and F(T < 3 m) ~ 60 – 100%. This variability shifts to median ages between ~ 30 – 100 days and F(T < 3 m) ~ 30 – 95% for dry conditions. This pattern of variability in daily TTDs in wet and dry periods is very closely matched by the estimates based on $^3H$ (Fig. 11b). Overall, the volume-weighted average TTDs of transpiration suggest median ages of around 14 days for wet conditions and between 35 days ($^3H$) and 70 days ($\delta^{18}O$) for dry conditions (Fig. 11d).

The modelled groundwater, in comparison, was found to be characterized by substantially less temporal variability in TTDs and older water ages (Fig. 12). The TTDs inferred from both, $\delta^{18}O$ and $^3H$, are similar and characterized by a median age of ~ 10 years under both, wet and dry conditions. While F(T < 3 m) of the groundwater largely remains < 1%, around 20 – 25 % of the groundwater is older than 20 years.

## 6 Implications, limitations and unresolved questions

What can we learn from the above? We believe the results obtained in this study have several implications for the utility of different tracer and model types, as described in detail below.

### 6.1 The individual roles of the choices of tracers and models for underestimation of water ages

The overall magnitude of water ages here estimated from time-invariant, lumped SW and CO models in combination with $\delta^{18}O$ reach MTTs of ~ 2 years (Table 6, Fig. 8). These values fall within the age ranges reported for comparable model experiments with seasonally variable tracers in many other catchments world-wide (see McGuire and McDonnell, 2006; Godsey et al., 2009; Hrachowitz et al., 2009; Stewart et al., 2010 and references therein). Similarly, the water ages estimated with the same CO models in combination with $^3H$ are with MTTs ~ 10 yrs by a factor of ~ 5 higher (Table 6, Fig. 8), and also well reflect the findings of previous studies, many of which suggest $^3H$-inferred catchment MTTs of ~ 10 – 15yr (Stewart et al., 2010 and references therein). This suggests that the Neckar basin does not exhibit unusual or unexpected water age characteristics. By themselves, these results would therefore lend further supporting evidence for the interpretation provided by Stewart et al. (2010) and, crucially, lead us to the same conclusion, that the use of $\delta^{18}O$ and comparable seasonally variable tracers truncate stream water ages.

However, and in stark contrast, the estimates of water age obtained from all P-SAS and IM-SAS model implementations in

this study, i.e., scenarios 13 – 21, are similar to each other irrespective of the tracer used. Water ages estimated from $\delta^{18}O$ are, with MTT > 11.4 yr, not only substantially older than those inferred from the SW and CO models (scenarios 1 – 3, 5, 7, 9, 11), but, most importantly, similar to those inferred from $^3H$ in P-SAS and IM-SAS models, which reach MTT ~ 11 – 13 yr (Table 7, Fig. 9). These water ages highlight the importance of old water in the Neckar basin, similar to what is suggested by the use of $^3H$ in CO models (scenarios 4, 6, 8, 10, 12).

It is important to note that the IM-SAS and, to a lesser degree, P-SAS models can simultaneously reproduce several signatures of the hydrological response together with the $\delta^{18}O$ and $^3H$ stream water signals. They therefore provide a more holistic description of physical transport processes in the system (Table 7, Fig. 3 – 5) than the SW and CO models, which mimic one single tracer signal and thus one isolated variable at a time. In addition, the P-SAS and IM-SAS model parameters that are most linked to tracer circulation, e.g. $S_{tot}$, $S_{s,p}$ and $S_{umax}$ (Fig. 7), exhibit little difference when obtained from calibration to $\delta^{18}O$ or $^3H$, respectively. This implies that both, $\delta^{18}O$ and $^3H$, provide similar information about how tracers are routed through the model and how water is stored in and released from the system. As a consequence, also the *simultaneous* representation of all three types of variables under consideration, i.e., the hydrological response as well as the $\delta^{18}O$ and $^3H$ stream signals, in these models is consistent with the observed data (scenarios 18, 21).

The above is further corroborated by how water ages in the Neckar basin respond to changing wetness conditions. Although not identical, $\delta^{18}O$ and $^3H$-inferred daily TTDs exhibit nevertheless broad agreement in the directions and magnitudes of change in response to changing wetness conditions (Fig. 10). Changes in stream flow TTDs in IM-SAS are not primarily caused by changes of water ages within individual storage components. In particular, the modelled water age distributions in the groundwater $S_s$ show limited sensitivity to changing wetness conditions, with MTT varying between ~ 18 years in dry periods and ~ 17 years in wet periods (Fig. 12). The TTDs in the transpiration flux $E_a$, which are reflective of the water ages in the unsaturated root zone $S_u$, exhibit with MTTs between ~ 0.20 and 0.12 years in dry and wet periods (Fig. 11), respectively, magnitudes and fluctuations over time that are similar to what has been previously reported in other studies (e.g., Hrachowitz et al., 2015; Soulsby et al., 2016; Visser et al., 2019; Birkel et al., 2020; Kuppel et al., 2020). However, the level of these age fluctuations alone is insufficient to explain the magnitude of change in stream flow TTDs, which can vary by several years. Instead, the temporal variability of stream flow TTDs is largely controlled by switches in the relative contributions of individual storage components to stream flow under different wetness conditions. Under increasingly wet conditions, considerably increasing proportions of comparably young water from $S_U$ contribute over shallow preferential flow pathways ($S_F$) to stream flow, while the relative proportion of groundwater contributing to stream flow under such conditions is reduced (Hrachowitz et al., 2013). Both tracers, $\delta^{18}O$ and $^3H$, generate these patterns in a corresponding way.

Altogether, this suggests that the P-SAS and IM-SAS models and the resulting estimates of water ages inferred from both, $\delta^{18}O$ and $^3H$, provide plausible descriptions of transport processes and thus water ages in the Neckar basin. Clearly, with current observation technology, it is impossible to know the real water age distribution at river basin scale. However, the water ages and their temporal variability inferred from both, $\delta^{18}O$ and $^3H$ using P-SAS and IM-SAS models are widely consistent. This is suggestive that it is not the use of $\delta^{18}O$ *per se* that leads to truncation of TTDs, but rather that time-invariant, lumped

convolution integral models are incapable of extracting sufficient information from $\delta^{18}O$ signals. These results mirror anecdotal evidence from several previous studies (e.g., Hrachowitz et al., 2015, 2021; Ala-aho et al., 2017; Buzacott et al., 2020; Yang et al, 2021). Although no direct comparison with $^3H$ data is provided in these studies, they demonstrated the potential of $\delta^{18}O$ in SAS-based model approaches to estimate water age distributions with considerable fractions of water older than 5 – 10 years and Birkel et al. (2020) explicitly estimated MTTs of up to 18 years. Our results also strongly support the findings and general conclusions of Rodriguez et al. (2021), who undertook a direct comparison of water ages inferred from $\delta^{18}O$ and $^3H$. In their study for a small catchment and based on shorter tracer time series, i.e., 2 years, and a system that is characterized by rather low MTT of ~ 3 years, they found that although $^3H$ led to higher MTTs than $\delta^{18}O$, the absolute difference between these ages estimates was with 0.2 years limited and even decreasing for higher percentiles of the water age distributions.

We therefore argue that the evidence emerging from our results and the above considerations is strong enough to reject the hypothesis that $\delta^{18}O$ as a tracer generally and systematically "cannot see water older than about 4 years" (Stewart et al., 2010, 2012) and the corresponding tails in water age distributions, leading to underestimations of water ages. We further argue that previous underestimations of water ages are rather a consequence of the use of time-invariant, lumped parameter convolution integral model techniques that cannot resolve the information contained $\delta^{18}O$ signals in a meaningful way for catchments with transient flow conditions. In contrast, the combined information using hydrological and tracer data and thus the consideration of transient flow conditions results in similar MTTs, independent of the used tracer. Note, that for this reason, time-variant implementations of convolution integral models that can describe transient conditions may hold the potential to similarly generate water age estimates from $\delta^{18}O$ signals that reflect the results of the P-SAS and IM-SAS models tested here.

However, and notwithstanding the rejection of the above hypothesis it is important to note that overall and in spite of the similarity between $\delta^{18}O$ and $^3H$ inferred water ages in the study basin on the basis of P-SAS and IM-SAS models, there may be no general equivalence between $\delta^{18}O$ and $^3H$ tracers. Instead, it is plausible to assume that differences will gradually increase with higher water ages. In systems characterized by water older than the water in the Neckar study basin, and where the amplitudes of the $\delta^{18}O$ stream signal are attenuated to below the analytical precision, the water age estimates from $\delta^{18}O$ will indeed become subject to increasing uncertainty up to the point where further attenuation and thus older water ages cannot be discerned anymore independent of modelling approaches. The specific magnitude of such a water age threshold remains difficult to quantify with the available data. However, given the results in the Neckar study basin, the question raised by Stewart et al. (2021), if $\delta^{18}O$ allows to see "the full range of travel times", can to some extent be answered: it can be assumed that, when used with a suitable model, $\delta^{18}O$ contains sufficient information for a meaningful characterization of water ages in systems characterized by MTTs of at least ~15 – 20 years, which encompasses the vast majority of river basins so far analyzed in literature (see Stewart et al., 2010 and references therein). As a step forward, the original hypothesis above can, for future research, be reformulated into: $\delta^{18}O$-inferred water age estimates are subject to increasing uncertainty and bias when compared to $^3H$-inferred estimates when stream water MTTs of ~ 15 – 20 years are exceeded in systems characterized by increasingly old water.

## 6.2 The role of spatial aggregation on underestimation of water ages

In addition to the above, Kirchner (2016) demonstrated that the use of seasonally variable tracers with time-invariant, lumped parameter model approaches, i.e., SW and CO, has considerable potential to underestimate water ages due to spatial aggregation of heterogeneous MTTs in systems characterized by large spatial contrasts in MTTs. We could here not reproduce that exact experiment, as stream observations were available only at one location for each tracer. However, in the distributed implementation of the IM-SAS-D model (scenarios 19 – 21), we nevertheless explicitly accounted, albeit to a limited degree, for heterogeneity in the system input variables as well as for potential differences in landscape types, as expressed by the three model HRUs. This resulted in different TTDs for the individual precipitation zones (Supplementary Material Figures S29-S31 and Table S6) and elevation zones and HRUs therein (not shown). The comparison between the lumped IM-SAS-L (scenarios 16 – 18) and the distributed IM-SAS-D models does not show major differences in their ability to reproduce the various hydrological signatures nor the $\delta^{18}$O and $^3$H stream signals (Table 5). Against evidence from various previous studies (e.g., Euser et al., 2015; Gao et al., 2016; Nijzink et al., 2016; Nguyen et al., 2022), this reflects to some degree the conclusion by Loritz et al. (2021), who found in a comparative analysis that distributed model implementations do not necessarily improve a model's ability to reproduce the hydrological response as compared to spatially lumped formulations. In addition, the contrasts in water ages between the discretized model components, with MTTs reaching from ~ 8 to ~ 22 yrs in the individual precipitation zones, may not be sufficient to significantly affect basin overall MTTs. As a consequence, the results of IM-SAS-L and IM-SAS-D also do not show major differences in the associated water age estimates, with MTTs ~ 11 – 17 yrs and 12 – 16 yrs, respectively (Table 7, Fig. 9).

How can this be interpreted? If significantly older ages were inferred from the distributed IM-SAS-D implementation, this would have provided strong supporting evidence for the role and effect of spatial heterogeneity on water ages as demonstrated by Kirchner (2016). However, the similar water ages inferred from the spatially lumped and distributed implementations, respectively, allow two possible but mutually contradicting interpretations. Either, it could indicate that the aggregation of spatial heterogeneity does not have any discernible effect on water ages inferred from the IM-SAS model in the study basin or, on the contrary, the spatial contrasts in water ages, limited by the spatial resolution of the model and the available data, were not sufficient to detect any significant differences. The evidence found here therefore remains inconclusive and further research is required to describe the role of the aggregation of spatial heterogeneity for estimates of water ages using IM-SAS type of models.

For any estimates of water ages in this study – as in any other study – it is important to bear in mind that they are conditional on the available data and models used. Uncertainties can and do arise from both, data and from decisions taken in the modelling process (e.g., Beven, 2006; Kirchner, 2006). One challenge in this study was that precipitation $\delta^{18}$O and $^3$H compositions were only available at rather coarse spatial and temporal resolutions. We have used the best available information to spatially extrapolate the tracer precipitation data from the individual sampling stations to estimate their spatial variation across the Neckar basin including stations outside the study basin. The monthly $\delta^{18}$O and $^3$H distribution in precipitation within South-

Germany is generally similar (Stumpp et al. 2014; Schmidt et al. 2020); still, regional correction for $\delta^{18}O$ might not be sufficient to explain local differences in $\delta^{18}O$ precipitation data. A similar limitation applies to the temporal resolution of tracer composition in precipitation as only monthly information was available. However, as the available data nevertheless reflect the seasonal variation in $\delta^{18}O$ and $^3H$ precipitation input, the uncertainties arising can be assumed to largely affect the short-term dynamics of tracers in the stream and thus rather young water ages, whereas the objective of our analysis was focused on the right tail of the water age distributions and thus on old ages. Notwithstanding these uncertainties, the overall model performances with respect to the hydrological and tracer responses, suggest that the choice of input data and the model formulations led to model results that are largely consistent with the observed responses in the stream. The observation that there is little ambiguity in the results further suggests that the remaining uncertainties are unlikely to affect the overall interpretation and conclusions of this study.

## 7 Conclusions

$\delta^{18}O$ and $^3H$ are frequently used as tracers in environmental sciences to estimate age distributions of water. However, it has previously been argued that seasonally variable tracers, such as $\delta^{18}O$, fail to detect the tails of water age distributions and therefore substantially underestimate water ages as compared to radioactive tracers, such as $^3H$. In this study for the Neckar River basin in central Europe and based on a >20-year record of hydrological, $\delta^{18}O$ and $^3H$ data we systematically scrutinized the above postulate by comparing water age distributions inferred from $\delta^{18}O$ and $^3H$ with a total of 21 different model implementations. The main findings of our analysis are the following:

(1) Water ages inferred from $\delta^{18}O$ with commonly used time-invariant, sine wave (SW) and lumped parameter convolution integral models (CO) are with MTTs ~ 1 – 2 years substantially lower that those obtained from $^3H$ with the same models, reaching MTTs ~ 10 years.

(2) In contrast, the concept of SAS-functions in combination with hydrological models (P-SAS, IM-SAS) did not only allow *simultaneous* representations of water storage fluctuations together with $\delta^{18}O$ and $^3H$ stream signals, but water ages inferred from $\delta^{18}O$ were with MTTs ~ 11 – 17 years much higher and even *higher* than inferred from $^3H$, which suggested MTTs ~ 11 – 13 years.

(3) Constraining P-SAS and IM-SAS model implementations individually with $\delta^{18}O$ and $^3H$ observations resulted in similar values for parameters that control water ages, such as the total storage $S_{tot}$ (P-SAS) or passive groundwater volumes $S_{s,p}$ (IM-SAS) In addition, $\delta^{18}O$ and $^3H$-constrained models both exhibited limited differences in the magnitudes of water ages in different parts of the models as well as in the temporal variability of TTDs in response to changing wetness conditions. This suggests that both tracers lead to comparable descriptions of how water is routed through the system.

(4) Based on the points above, we reject the hypothesis that $\delta^{18}O$ as a tracer generally and systematically "cannot see water older than about 4 years" (Stewart et al., 2010, 2012) and that it truncates the corresponding tails in water age distributions, leading to underestimations of water ages.

(5) Instead, our results provide evidence of broad equivalence of $\delta^{18}O$ and $^3H$ as age tracers for systems characterized by MTTs of at least 15 – 20 years.

(6) The question to which degree aggregation of spatial heterogeneity can further adversely affect estimates of water ages remains unresolved as the lumped and distributed implementations of the IM-SAS model provided similar and thus inconclusive results.

Overall, this study demonstrates that previously reported underestimations of water ages are most likely not a result of the use of $\delta^{18}O$ or other seasonally variable tracers *per se*. Rather, these underestimations can be largely attributed to the choices of model approaches which rely on assumptions not frequently met in catchment hydrology. Given the vulnerability of lumped, time-invariant parameter convolution integral approaches in combination with $\delta^{18}O$ to substantially underestimate water ages due to transient flow conditions, spatial aggregation and potentially other, still unknown effects, we therefore strongly advocate to avoid the use of this model type in combination with seasonally variable tracers and to instead adopt SAS-based or other model formulations that allow for the representation of transient conditions.

*Code availability.* The model codes underlying this paper will be available online in the 4TU data repository (DOI: 10.4121/b75c9108-c5b8-4266-9b82-1ad08c76adcc). The equations used in the model are described in supplement.

*Data availability.* The meteorological and hydrological data used in this study can be obtained from German Weather Service (DWD) and the German Federal Institute of Hydrology (BfG). Both $\delta^{18}O$ and $^3H$ input data used in this study can be obtained from the WISER database portal of the International Atomic Energy Agency (http://www-naweb.iaea.org/napc/ih/IHS_resources_gnip.html). Both $\delta^{18}O$ and $^3H$ output data used in this study can be made available by Christine Stumpp upon request.

*Author contributions.* SW, MH and GS designed the study, SW executed the experiments, all authors contributed to general idea, the discussion and writing of the manuscript.

*Competing interests.* Some authors are members of the editorial board of the HESS journal. The peer-review process was guided by an independent editor, and the authors have also no other competing interests to declare.

*Acknowledgements.* We thank Dr. Axel Schmidt of BAFG for valuable assistance with and information on tritium data. We gratefully acknowledge financial support from China Scholarship Council (CSC). We would like to thank the editor and two reviewers for providing a list of critical and very valuable comments that helped to considerably improve the manuscript.

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

**Table1.** Characteristics of the Neckar catchment in Germany

| Characteristics | |
|---|---|
| latitude (N) | 48°02′00″-49°33′45″ |
| longitude (E) | 8°18′45″-10°18′45″ |
| Area (km$^2$) | 13,041 |
| Average annual precipitation (mm yr$^{-1}$) | 909 |
| Average annual temperature (°C) | 8.9 |
| Elevation range (m) | 122-1019 |
| Mean elevation (m) | 569 |
| Slope range (°) | 0-53 |
| Mean slope (°) | 5.1 |
| Forest dominated land (%) | 38.1 |
| Grass dominated land (%) | 51.2 |
| Wetland (%) | 10.7 |

1110

**Table 2.** The 12 time-invariant, lumped SW/CO model scenarios here implemented for the Neckar study basin together with the associated calibration strategies, the individual calibration performance metric, the type of models as well as the prior parameter ranges and the optimal parameter value from calibration. SW indicates sine-wave models, CO indicates time-invariant, lumped parameter convolution integral models. EM represents an exponential TTD and GM indicates a gamma distribution TTD. 2EM indicates a two parallel linear reservoir model, 3EM indicates a three parallel linear reservoir model and EPM indicates an exponential piston flow model. The calibration strategies show which variable a model was calibrated to using the Mean Square Error (MSE) with $C_\delta{}^{18}O$ calibration to the observed stream water $\delta^{18}O$ signal and $C^3{}_H$ calibration to observed stream water $^3H$. *) Note, that for SW models calibration involves least-square fits of sine waves to both, the precipitation and stream flow signals available. †) fixed to a value of 1.

| Scenario | | 1 | 2 | 3 | 4 | 5 | 6 | 7 | 8 | 9 | 10 | 11 | 12 |
|---|---|---|---|---|---|---|---|---|---|---|---|---|---|
| Model | | SW-EM | SW-GM | CO-EM | | CO-GM | | CO-2EM | | CO-3EM | | CO-EPM | |
| Signature | Calibration strategy → / Performance metric ↓ | $C_x{}^{*)}$ | $C_x{}^{*)}$ | $C_\delta{}^{18}O$ | $C^3{}_H$ | $C_\delta{}^{18}O$ | $C^3{}_H$ | $C_\delta{}^{18}O$ | $C^3{}_H$ | $C_\delta{}^{18}O$ | $C^3{}_H$ | $C_\delta{}^{18}O$ | $C^3{}_H$ |
| Times series $\delta^{18}O$ | $MSE_{\delta^{18}O}$ | • | • | • | - | • | - | • | - | • | - | • | - |
| Time series $^3H$ | $MSE_{^3H}$ | - | - | - | • | - | • | - | • | - | • | - | • |
| Parameter | Prior range | Optimal parameter value | | | | | | | | | | | |
| $A_p$ (‰) | -*) | 2.69 | | | | | | | | | | | |
| $A_s$ (‰) | -*) | 0.57 | | | | | | | | | | | |
| α (-) | 0.1 − 2 | - | - | 1[†] | 1[†] | 0.44 | 0.58 | 1[†] | 1[†] | 1[†] | 1[†] | 1[†] | 1[†] |
| $\beta_1$ (d) | 1 − 15000 | - | - | 513 | 3795 | 2048 | 6086 | 16 | 84 | 11 | 66 | 662 | 3665 |
| $\beta_2$ (d) | 1 − 15000 | - | - | - | - | - | - | 832 | 5388 | 12 | 112 | - | - |
| $\beta_3$ (d) | 1 − 15000 | - | - | - | - | - | - | - | - | 963 | 5299 | - | - |
| $f_1$ (-) | 0 − 1 | - | - | 1[†] | 1[†] | 1[†] | 1[†] | 0.18 | 0.36 | 0.06 | 0.02 | 1[†] | 1[†] |
| $f_2$ (-) | 0 − 1 | - | - | - | - | - | - | - | - | 0.12 | 0.34 | - | - |
| η (-) | 1 − 3 | - | - | 1[†] | 1[†] | 1[†] | 1[†-)] | 1[†] | 1[†] | 1[†] | 1[†] | 1.91 | 1.01 |

**Table 3.** The 9 P-SAS and IM-SAS model scenarios here implemented for the Neckar study basin together with the associated calibration strategies, the individual calibration performance metrics and the type of spatial implementation (lumped or distributed) as well as the associated prior parameter ranges and the ranges of the pareto optimal solutions from calibration. P-SAS indicates the model with one compartment as described in Benettin et al. (2017), and IM-SAS indicates the integrated hydrological model based on SAS-functions. The symbols L and D indicate lumped and distributed model implementations, respectively. The calibration strategies show which variables/signatures a model was simultaneously calibrated to using the Mean Square Error (MSE) with $C_{\delta^{18}O,Q}$ simultaneous calibration to $\delta^{18}O$ and six signatures of stream flow Q; $C_{^3H,Q}$ simultaneous calibration to $^3H$ and the signatures of Q; $C_{\delta^{18}O,^3H,Q}$ the simultaneous calibration to $\delta^{18}O$, $^3H$ and the signatures of Q. †) fixed to a value of 1.

| Scenario | | 13 | 14 | 15 | 16 | 17 | 18 | 19 | 20 | 21 |
|---|---|---|---|---|---|---|---|---|---|---|
| Model | | | P-SAS | | | IM-SAS-L | | | IM-SAS-D | |
| Implementation | | | | | Lumped | | | | Distributed | |
| Signature | Calibration strategy → / Performance metric ↓ | $C_{\delta^{18}O}$ | $C_{^3H}$ | $C_{\delta^{18}O,^3H}$ | $C_{\delta^{18}O,Q}$ | $C_{^3H,Q}$ | $C_{\delta^{18}O,^3H,Q}$ | $C_{\delta^{18}O,Q}$ | $C_{^3H,Q}$ | $C_{\delta^{18}O,^3H,Q}$ |
| Times series $\delta^{18}O$ | $MSE_{\delta^{18}O}$ | • | - | • | • | - | • | • | - | • |
| Time series $^3H$ | $MSE_{^3H}$ | - | • | • | - | • | • | - | • | • |
| Time series of stream flow (Q) | $MSE_Q$ | - | - | - | • | • | • | • | • | • |
| Time series of log(Q) | $MSE_{log(Q)}$ | - | - | - | • | • | • | • | • | • |
| Flow duration curve of Q (FDC$_Q$) | $MSE_{FDC_Q}$ | - | - | - | • | • | • | • | • | • |
| Flow duration curve log(Q) (FDC$_{log(Q)}$) | $MSE_{FDC_{log(Q)}}$ | - | - | - | • | • | • | • | • | • |
| Seasonal runoff coefficient (RC) | $MSE_{RC}$ | - | - | - | • | • | • | • | • | • |
| Autocorrelation function of Q (AC$_Q$) | $MSE_{AC_Q}$ | - | - | - | • | • | • | • | • | • |
| **Parameter** | **Prior range** | | | | **Optimal parameter value** | | | | | |
| $k_E$ | 0.1-1.0 | 1$^†$ | 1$^†$ | 1$^†$ | - | - | - | - | - | - |
| $k_Q$ | 0.1-1.0 | 0.34 | 0.28 | 0.29-0.33 | - | - | - | - | - | - |
| $S_{tot}$ (mm) | 100-20000 | 15595 | 16638 | 7414-18245 | - | - | - | - | - | - |
| $T_t$ (°C) | -2.5-2.5 | - | - | - | -0.94-2.08 | -0.88-1.75 | -2.15-1.57 | -1.84-1.81 | -1.74-0.16 | -1.92-1.54 |
| $C_{melt}$ (mm°C$^{-1}$d$^{-1}$) | 1-5 | - | - | - | 2.32-4.42 | 1.67-3.96 | 1.79-3.77 | 2.30-4.89 | 1.56-3.25 | 1.23-4.10 |
| $S_{imaxF}$ (mm) | 0.1-5 | - | - | - | 1.53-3.73 | 1.35-4.39 | 0.55-4.10 | 3.18-4.03 | 2.94-4.98 | 2.04-4.39 |
| $S_{imaxG}$ (mm) | 0.1-5 | - | - | - | - | - | - | 0.30-0.60 | 0.46-0.70 | 0.38-1.39 |
| $C_a$ (-) | 0.1-0.7 | - | - | - | 0.24-0.43 | 0.35-0.55 | 0.33-0.62 | 0.30-0.66 | 0.38-0.52 | 0.30-0.56 |
| $S_{umaxF}$ (mm) | 50-500 | - | - | - | 314-415 | 236-355 | 233-464 | 355-438 | 301-441 | 352-485 |
| $S_{umaxG}$ (mm) | 50-500 | - | - | - | - | - | - | 161-199 | 152-287 | 173-297 |
| $S_{umaxW}$ (mm) | 50-500 | - | - | - | - | - | - | 56-149 | 89-149 | 85-148 |
| $\gamma_F$ (-) | 0.1-5 | - | - | - | 0.93-1.68 | 0.61-1.01 | 0.57-2.03 | 0.99-4.59 | 2.04-3.98 | 0.76-4.94 |
| $\gamma_G$ (-) | 0.1-5 | - | - | - | - | - | - | 0.15-0.26 | 0.23-0.53 | 0.11-0.52 |
| $\gamma_W$ (-) | 0.1-5 | - | - | - | - | - | - | 0.14-3.64 | 0.12-0.32 | 0.10-2.88 |
| $D$ (-) | 0-1 | - | - | - | 0.30-0.77 | 0.41-0.81 | 0.30-0.69 | 0.03-0.35 | 0.06-0.33 | 0.03-0.33 |
| $C_{pmaxF}$ (mm d$^{-1}$) | 0.1-4 | - | - | - | 1.04-2.03 | 0.98-1.83 | 1.05-2.62 | 0.91-3.19 | 0.94-3.66 | 1.37-3.72 |
| $C_{pmaxG}$ (mm d$^{-1}$) | 0.1-4 | - | - | - | - | - | - | 0.74-1.80 | 0.22-1.17 | 0.93-2.13 |
| $C_{max}$ (mm d$^{-1}$) | 0-4 | - | - | - | - | - | - | 0.00-0.31 | 0.02-1.06 | 0.01-0.98 |
| $K_{fF}$ (d$^{-1}$) | 0.2-5 | - | - | - | 0.27-2.99 | 0.24-1.52 | 0.31-3.79 | 0.21-3.03 | 0.21-0.70 | 0.50-4.21 |
| $K_{fG}$ (d$^{-1}$) | 0.2-5 | - | - | - | - | - | - | 0.21-4.04 | 0.25-0.41 | 0.25-3.66 |
| $K_s$ (d$^{-1}$) | 0.002-0.2 | - | - | - | 0.04-0.19 | 0.05-0.18 | 0.05-0.18 | 0.05-0.17 | 0.03-0.14 | 0.05-0.17 |
| $S_{s,p}$ (mm) | 100-20000 | - | - | - | 4107-10029 | 3924-9339 | 4078-13676 | 4278-9011 | 3270-4622 | 4150-8568 |

**Table 4**. Performance metrics of the 12 time-invariant, lumped SW/CO model implementations for the 2001 – 2009 calibration period (cal.) and the 2010 – 2016 model evaluation period (val.). For brevity only the values for the most balanced solution are shown here. *) The MSE values provided for $C_x$ describe the sine wave fits of both, the precipitation and stream flow $\delta^{18}O$ signals, respectively.

| Scenario | | 1 | 2 | 3 | 4 | 5 | 6 | 7 | 8 | 9 | 10 | 11 | 12 |
|---|---|---|---|---|---|---|---|---|---|---|---|---|---|
| Model | | SW-EM | SW-GM | CO-EM | | CO-GM | | CO-2EM | | CO-3EM | | CO-EPM | |
| Calibration strategy → Performance metric ↓ | | $C_x$ | $C_x$ | $C_{\delta^{18}O}$ | $C_{^3H}$ | $C_{\delta^{18}O}$ | $C_{^3H}$ | $C_{\delta^{18}O}$ | $C_{^3H}$ | $C_{\delta^{18}O}$ | $C_{^3H}$ | $C_{\delta^{18}O}$ | $C_{^3H}$ |
| $MSE_{\delta^{18}O}$ | cal. | 3.850/0.121[*)] | | 0.327 | - | 0.204 | - | 0.171 | - | 0.171 | - | 0.254 | - |
| | val. | 5.208/0.144[*)] | | 0.432 | - | 0.192 | - | 0.192 | - | 0.191 | - | 0.683 | - |
| $MSE_{^3H}$ | cal. | - | - | - | 5.903 | - | 5.791 | - | 5.171 | - | 5.170 | - | 5.926 |
| | val. | - | - | - | 5.155 | - | 4.597 | - | 3.964 | - | 4.000 | - | 5.115 |

**Table 5**. Performance metrics of the 9 P-SAS and IM-SAS model scenarios for the 2001 – 2009 calibration period (cal.) and the 2010 – 2016 model evaluation period (val.). For brevity only the values for the most balanced solution, i.e., lowest $D_E$ (Eq. 16) are shown here. The ranges of all performance metrics for the full set of pareto optimal solutions for the multi-objective calibration cases (Scenarios 15 – 21) are provided in the Table S5 in supplement.

| Scenario | | 13 | 14 | 15 | 16 | 17 | 18 | 19 | 20 | 21 |
|---|---|---|---|---|---|---|---|---|---|---|
| Model | | P-SAS | | | IM-SAS-L | | | IM-SAS-D | | |
| Implementation | | Lumped | | | | | | Distributed | | |
| Calibration strategy → Performance metric ↓ | | $C_{\delta^{18}O}$ | $C_{^3H}$ | $C_{\delta^{18}O,^3H}$ | $C_{\delta^{18}O,Q}$ | $C_{^3H,Q}$ | $C_{\delta^{18}O,^3H,Q}$ | $C_{\delta^{18}O,Q}$ | $C_{^3H,Q}$ | $C_{\delta^{18}O,^3H,Q}$ |
| $MSE_{\delta^{18}O}$ | cal. | 0.069 | - | 0.078 | 0.083 | - | 0.118 | 0.079 | - | 0.114 |
| | val. | 0.231 | - | 0.215 | 0.332 | - | 0.273 | 0.273 | - | 0.475 |
| $MSE_{^3H}$ | cal. | - | 2.828 | 2.847 | - | 2.972 | 2.823 | - | 2.920 | 2.981 |
| | val. | - | 1.717 | 1.710 | - | 2.389 | 2.285 | - | 2.357 | 2.450 |
| $MSE_Q$ | cal. | - | - | - | 0.202 | 0.299 | 0.308 | 0.228 | 0.263 | 0.317 |
| | val. | - | - | - | 0.224 | 0.297 | 0.329 | 0.251 | 0.283 | 0.336 |
| $MSE_{log(Q)}$ | cal. | - | - | - | 0.120 | 0.158 | 0.174 | 0.130 | 0.171 | 0.161 |
| | val. | - | - | - | 0.120 | 0.148 | 0.150 | 0.127 | 0.201 | 0.165 |
| $MSE_{FDC_Q}$ | cal. | - | - | - | 0.058 | 0.024 | 0.073 | 0.022 | 0.017 | 0.025 |
| | val. | - | - | - | 0.103 | 0.022 | 0.142 | 0.043 | 0.065 | 0.059 |
| $MSE_{FDC_{log(Q)}}$ | cal. | - | - | - | 0.011 | 0.011 | 0.047 | 0.006 | 0.019 | 0.009 |
| | val. | - | - | - | 0.015 | 0.009 | 0.047 | 0.009 | 0.050 | 0.018 |
| $MSE_{RC}$ | cal. | - | - | - | 0.004 | 0.005 | 0.007 | 0.003 | 0.006 | 0.003 |
| | val. | - | - | - | 0.004 | 0.004 | 0.005 | 0.003 | 0.008 | 0.003 |
| $MSE_{AC_Q}$ | cal. | - | - | - | 0.003 | 0.002 | 0.003 | 0.002 | 0.001 | 0.001 |
| | val. | - | - | - | 0.008 | 0.002 | 0.001 | 0.005 | 0.002 | 0.007 |

**Table 6.** Metrics of stream flow TTDs derived from the 12 SW/CO model scenarios with the different associated calibration strategies based on different, where $C_{\delta^{18}O}$ indicates calibration to $\delta^{18}O$, $C^{3}H$ calibration to $^{3}H$. The TTD metrics represent the best fits of the respective time-invariant TTD. The water fractions are shown as the fractions of below a specific age T, i.e. F(T<age). The columns with absolute difference $\Delta$ summarize the differences in TTDs from the same models calibrated to $\delta^{18}O$ and $^{3}H$, respectively. The subscripts indicate the scenarios that are compared (e.g., $\Delta_{3,4}$ compares scenarios 3 and 4). *Note that the fraction of water younger than 3 months F(T<3m) is comparable to the fraction of young water as suggested by Kirchner (2016)

| Scenario | | 1 | 2 | 3 | 4 | 5 | 6 | 7 | 8 | 9 | 10 | 11 | 12 | $\Delta_{3,4}$ | $\Delta_{5,6}$ | $\Delta_{7,8}$ | $\Delta_{9,10}$ | $\Delta_{11,12}$ |
|---|---|---|---|---|---|---|---|---|---|---|---|---|---|---|---|---|---|---|
| Model | | SW-EM | SW-GM | CO-EM | | CO-GM | | CO-2EM | | CO-3EM | | CO-EPM | | Absolute difference | | | | |
| Calibration strategy → TTD metrics ↓ | | $C_x$ | $C_x$ | $C_{\delta^{18}O}$ | $C^{3}H$ | $C_{\delta^{18}O}$ | $C^{3}H$ | $C_{\delta^{18}O}$ | $C^{3}H$ | $C_{\delta^{18}O}$ | $C^{3}H$ | $C_{\delta^{18}O}$ | $C^{3}H$ | $\Delta TT_{\delta^{18}O\text{-}^{3}H}$ / $\Delta F(T<x)_{\delta^{18}O\text{-}^{3}H}$ | | | | |
| Percentiles (yr) | Mean (yr) | 0.7 | 1.8 | 1.4 | 10.4 | 2.4 | 9.7 | 1.9 | 9.5 | 2.1 | 9.4 | 1.8 | 10 | -9.0 | -7.3 | -7.6 | -7.3 | -8.2 |
| | 10th | 0.1 | <0.1 | 0.1 | 1.1 | <0.1 | 0.3 | <0.1 | <0.1 | <0.1 | 0.9 | 1.0 | 1.1 | -1.0 | -0.2 | 0.0 | -0.8 | -0.1 |
| | 25th | 0.2 | 0.2 | 0.4 | 3.0 | 0.2 | 1.3 | 0.2 | 0.3 | 0.2 | 2.8 | 1.1 | 2.9 | -2.6 | -1.1 | -0.1 | -2.6 | -1.8 |
| | 50th (median) | 0.5 | 0.8 | 1.0 | 7.2 | 1.0 | 5.0 | 1.1 | 3.6 | 1.3 | 7.3 | 1.5 | 7 | -6.2 | -4.0 | -2.5 | -6.0 | -5.5 |
| | 75th | 1.0 | 2.3 | 1.9 | 14.4 | 3.2 | 13.1 | 2.7 | 13.8 | 3.1 | 15.0 | 2.2 | 13.9 | -12.5 | -9.9 | -11.1 | -11.9 | -11.7 |
| | 90th | 1.7 | 4.8 | 3.2 | 26.3 | 6.8 | 25.4 | 4.8 | 27.3 | 5.6 | 25.6 | 3.0 | 23.1 | -23.1 | -18.6 | -22.5 | -20.0 | -20.1 |
| Water fractions (%) | F(T<3 m)* | 29 | 29 | 16 | 2 | 28 | 10 | 26 | 25 | 25 | 3 | 0 | 2 | 14 | 18 | 1 | 22 | -2 |
| | F(T<6 m) | 49 | 41 | 30 | 5 | 38 | 14 | 34 | 34 | 32 | 6 | 0 | 5 | 25 | 24 | 0 | 26 | -5 |
| | F(T<1 yr) | 74 | 55 | 51 | 9 | 50 | 21 | 47 | 40 | 44 | 10 | 13 | 9 | 42 | 29 | 7 | 34 | 4 |
| | F(T<3 yr) | 98 | 81 | 88 | 25 | 74 | 39 | 78 | 48 | 74 | 26 | 90 | 26 | 63 | 35 | 30 | 48 | 64 |
| | F(T<5 yr) | 100 | 91 | 97 | 38 | 85 | 50 | 91 | 55 | 88 | 38 | 99 | 39 | 59 | 35 | 36 | 50 | 60 |
| | F(T<10 yr) | 100 | 98 | 100 | 62 | 95 | 68 | 99 | 68 | 98 | 60 | 100 | 63 | 38 | 27 | 31 | 38 | 37 |
| | F(T<20 yr) | 100 | 100 | 100 | 85 | 100 | 85 | 100 | 84 | 100 | 84 | 100 | 86 | 15 | 15 | 16 | 16 | 14 |

**Table 7.** Metrics of stream flow TTDs derived from the 9 P-SAS and IM-SAS model scenarios with the different associated calibration strategies, where $C_{\delta^{18}O}$ indicates calibration to $\delta^{18}O$, $C^{3}H$ calibration to $^{3}H$, while $C_{\delta^{18}O,Q}$, $C^{3}H,Q$ and $C_{\delta^{18}O,^{3}H,Q}$ indicate multi-objective, i.e. simultaneous calibration to combinations of $\delta^{18}O$, $^{3}H$ and stream flow. The TTD metrics represent the mean of all volume-weighted daily streamflow TTDs for the modelling period 01/10/2001 – 31/12/2016 from the most balanced solutions (i.e. lowest $D_E$). The values in brackets indicate the 5th/95th percentiles of TTDs representing the pareto optimal solutions. The mean TT was estimated by fitting Gamma distributions to the volume-weighted mean TTDs of each scenario. The water fractions are shown as the fractions of below a specific age T, i.e. F(T<age). The columns with absolute difference $\Delta$ summarize the differences in TTDs from the most balanced solutions of the same models calibrated to $\delta^{18}O$ and $^{3}H$, respectively. The subscripts indicate the scenarios that are compared (e.g., $\Delta_{13,14}$ compares scenarios 13 and 14). *Note that the fraction of water younger than 3 months F(T<3m) is comparable to the fraction of young water suggested by Kirchner (2016).

| Scenario | | 13 | 14 | 15 | 16 | 17 | 18 | 19 | 20 | 21 | $\Delta_{13,14}$ | $\Delta_{16,17}$ | $\Delta_{19,20}$ |
|---|---|---|---|---|---|---|---|---|---|---|---|---|---|
| Model | | P-SAS | | | IM-SAS-L | | | IM-SAS-D | | | Absolute difference | | |
| Calibration strategy → TTD metrics ↓ | | $C_{\delta^{18}O}$ | $C^{3}H$ | $C_{\delta^{18}O,^{3}H}$ | $C_{\delta^{18}O,Q}$ | $C^{3}H,Q$ | $C_{\delta^{18}O,^{3}H,Q}$ | $C_{\delta^{18}O,Q}$ | $C^{3}H,Q$ | $C_{\delta^{18}O,^{3}H,Q}$ | $\Delta TT_{\delta^{18}O\text{-}^{3}H}$ / $\Delta F(T<x)_{\delta^{18}O\text{-}^{3}H}$ | | |
| Percentiles (yr) | Mean (yr) | 11.4 | 11.0 | 11.0 | 17.4 (16.9/21.1) | 11.9 (11.5/21.3) | 11.2 (9.9/16.8) | 15.6 (12.0/19.9) | 13.2 (13.2/21.1) | 12.8 (11.1/18.6) | 0.4 | 5.5 | 2.4 |
| | 10th | 0.0 | 0.0 | 0.0 | 0.5 (0.0/0.1) | 0.5 (0.0/0.1) | 0.4 (0.0/0.1) | 0.3 (0.0/0.0) | 0.3 (0.0/0.0) | 0.3 (0.0/0.1) | 0.0 | 0.0 | 0.0 |
| | 25th | 0.4 | 0.2 | 0.2 | 2.1 (0.1/0.4) | 1.9 (0.1/1.2) | 1.5 (0.1/1.7) | 2.1 (0.1/0.2) | 1.5 (0.1/0.2) | 1.4 (0.2/0.4) | 0.2 | 0.2 | 0.6 |
| | 50th (median) | 3.2 | 2.4 | 2.5 | 9.0 (9.8/15.9) | 6.5 (3.6/11.7) | 5.7 (4.8/11.6) | 8.6 (4.7/10.9) | 6.7 (1.6/5.8) | 6.6 (5.4/12.3) | 0.7 | 2.5 | 1.9 |
| | 75th | 13.7 | 12.5 | 12.5 | 22.2 (25.1/28.3) | 17.6 (17.1/27.7) | 16.3 (14.7/25.0) | 20.8 (18.0/26.9) | 18.8 (14.3/18.0) | 17.8 (16.4/26.7) | 1.2 | 4.6 | 2.0 |
| | 90th | 33.4 | 33.4 | 32.7 | 31.3 (32.0/34.0) | 29.2 (27.3/33.8) | 28.6 (25.2/31.8) | 31.1 (28.2/33.1) | 30.4 (26.3/28.9) | 29.9 (27.1/32.9) | 0.0 | 2.1 | 0.7 |
| Water fractions (%) | F(T<3 m)* | 22 | 26 | 26 | 18 (23/29) | 23 (19/38) | 21 (15/33) | 16 (28/36) | 22 (26/43) | 23 (20/29) | -5 | -5 | -6 |
| | F(T<6 m) | 27 | 32 | 32 | 21 (25/31) | 29 (22/43) | 30 (18/36) | 20 (30/38) | 27 (30/47) | 27 (23/32) | -5 | -8 | -7 |
| | F(T<1 yr) | 34 | 39 | 39 | 24 (26/33) | 32 (24/44) | 35 (19/37) | 22 (31/39) | 30 (33/49) | 29 (25/35) | -5 | -8 | -8 |
| | F(T<3 yr) | 49 | 53 | 52 | 31 (31/37) | 39 (31/49) | 42 (22/43) | 30 (34/45) | 37 (40/53) | 37 (31/42) | -4 | -8 | -7 |
| | F(T<5 yr) | 57 | 60 | 60 | 38 (33/41) | 46 (35/53) | 49 (24/51) | 38 (38/51) | 44 (47/58) | 44 (36/48) | -3 | -8 | -6 |
| | F(T<10 yr) | 69 | 71 | 71 | 52 (41/50) | 59 (41/62) | 62 (46/64) | 53 (46/62) | 58 (60/68) | 58 (46/62) | -2 | -7 | -5 |
| | F(T<20 yr) | 82 | 83 | 83 | 71 (55/65) | 77 (52/78) | 79 (65/78) | 74 (59/78) | 76 (75/81) | 77 (61/79) | -1 | -6 | -2 |

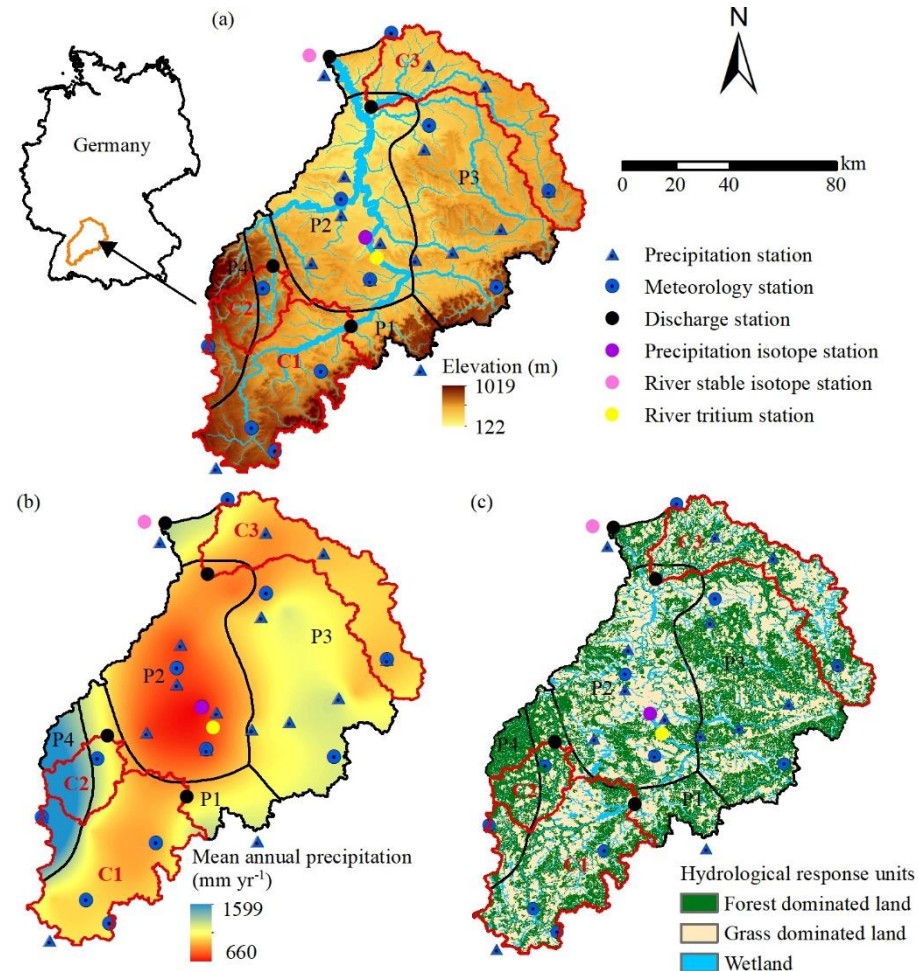

**Figure 1.** (a) Elevation of the Neckar catchment with discharge and hydro-meteorological stations as well as the water sampling locations used in this study, (b) the spatial distribution of long-term mean annual precipitation in the Neckar catchment and the stratification into four distinct precipitation zones P1 – P4 (black outline), and the red outlines indicate three sub-catchments (C1: Kirchentellinsfurt, C2: Calw, and C3: Untergriesheim) within the Neckar basin, (c) hydrological response units classified according to their land-cover and topographic characteristics.

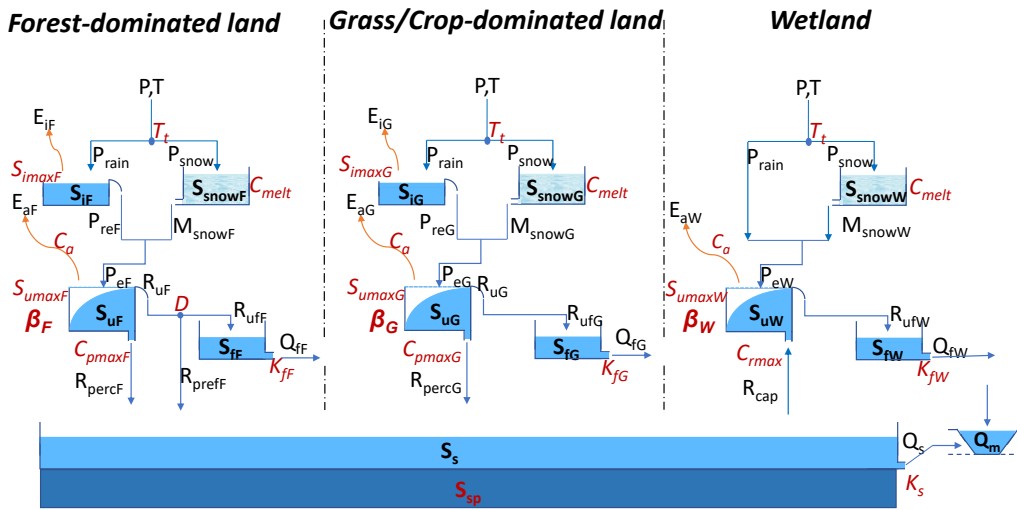

**Figure 2.** Model structure of the integrated model, discretized into three parallel hydrological response units HRU, i.e. forest, grassland and wetland in each precipitation zone P1 – P4. The light blue boxes indicate the hydrologically active individual storage volumes. The dark blue box indicates the hydrologically passive storage volume $S_{s,p}$. The arrow lines indicate water fluxes and model parameters are shown in red. All symbols are described in Table S4 in the Supplementary Material.

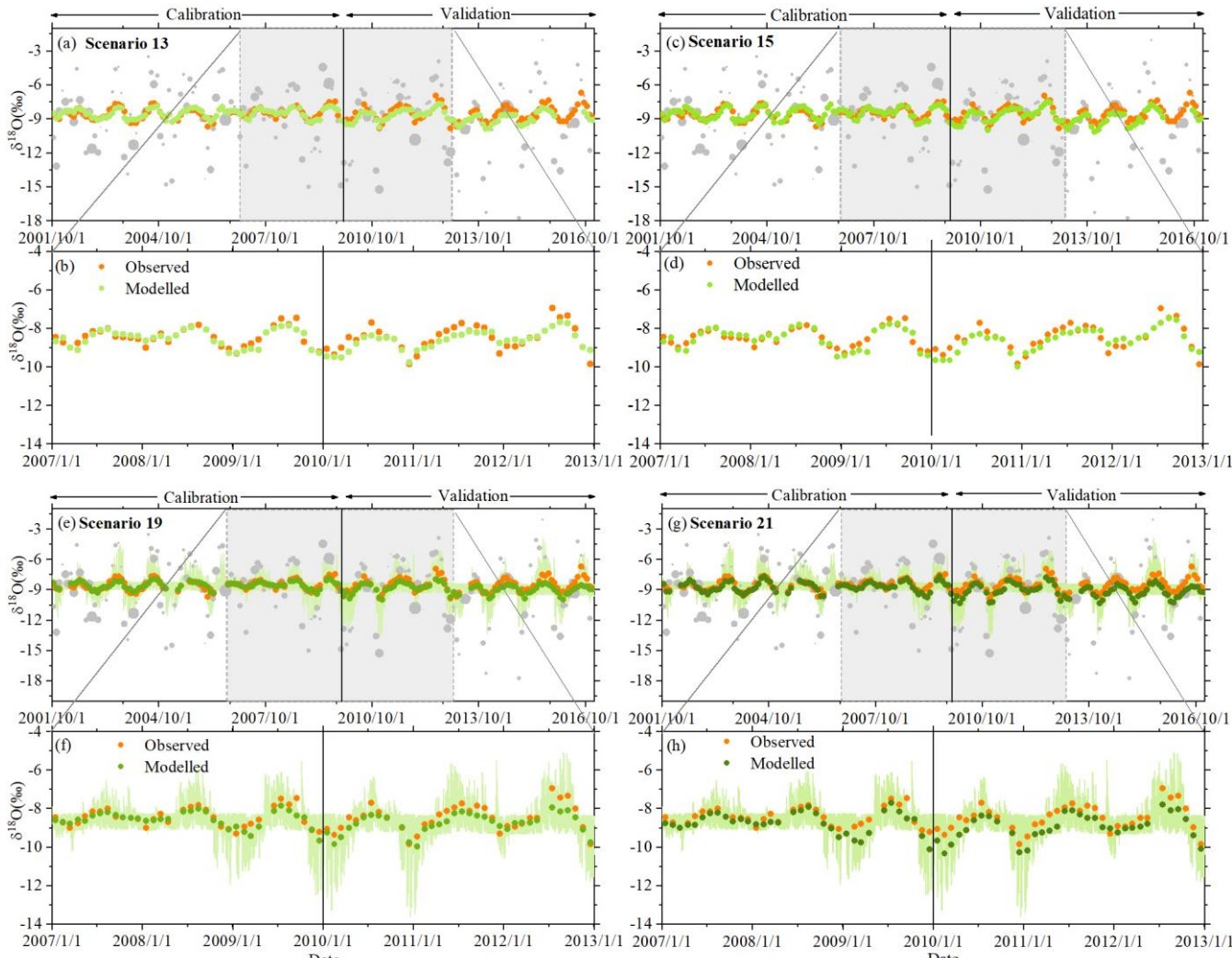

**Figure 3**. The time series of stream δ¹⁸O reproduced by models P-SAS (scenarios 13 and 15) and IM-SAS-D (scenarios 19 and 21) based on different calibration strategies. IM-SAS-D model based on simultaneous calibration to δ¹⁸O and the streamflow signatures, i.e. calibration strategy $C_{\delta^{18}O,Q}$ (scenario 19) and $C_{\delta^{18}O,^3H,Q}$ (scenario 21), for the model calibration and evaluation periods. (a) Observed δ¹⁸O signals in precipitation (light grey dots; size of dots indicates the precipitation volume) and observed stream δ¹⁸O signals (orange dots) as well as the most balanced modelled δ¹⁸O signal in the stream (light green dots) for scenario 13 from calibration strategy $C_{\delta^{18}O}$, (b) zoom-in of observed and modelled δ¹⁸O signals in the stream for the 01/01/2007 – 31/12/2012 period for scenario 13, (c) Observed δ¹⁸Osignals in precipitation and in stream same as (a), and the modelled stream δ¹⁸Osignals (relatively darker green dots) for scenario 15 from calibration strategy $C_{\delta^{18}O,^3H}$, (d) zoom-in of observed and modelled δ¹⁸O signals in the stream for the 01/01/2007 – 31/12/2012 period for scenario 15. (e) Observed δ¹⁸O signals in precipitation and in stream same as (a), and the modelled stream δ¹⁸O signals (relatively darker green dots) for scenario 19 and the 5th/95th percentile of all retained pareto optimal solutions obtained from calibration strategy $C_{\delta^{18}O,Q}$ (light green shaded area), (f) zoom-in of observed and modelled δ¹⁸O signals in the stream for the 01/01/2007 – 31/12/2012 period for scenario 19, (g) Observed δ¹⁸O signals in precipitation and in stream same as (a), and the modelled stream δ¹⁸O signals (relatively darker green dots) for scenario 21 and the 5th/95th percentile of all retained pareto optimal solutions obtained from calibration strategy $C_{\delta^{18}O,^3H,Q}$ (light green shaded area), (h) zoom-in of observed and modelled δ¹⁸O signals in the stream for the 01/01/2007 – 31/12/2012 period for scenario 21.

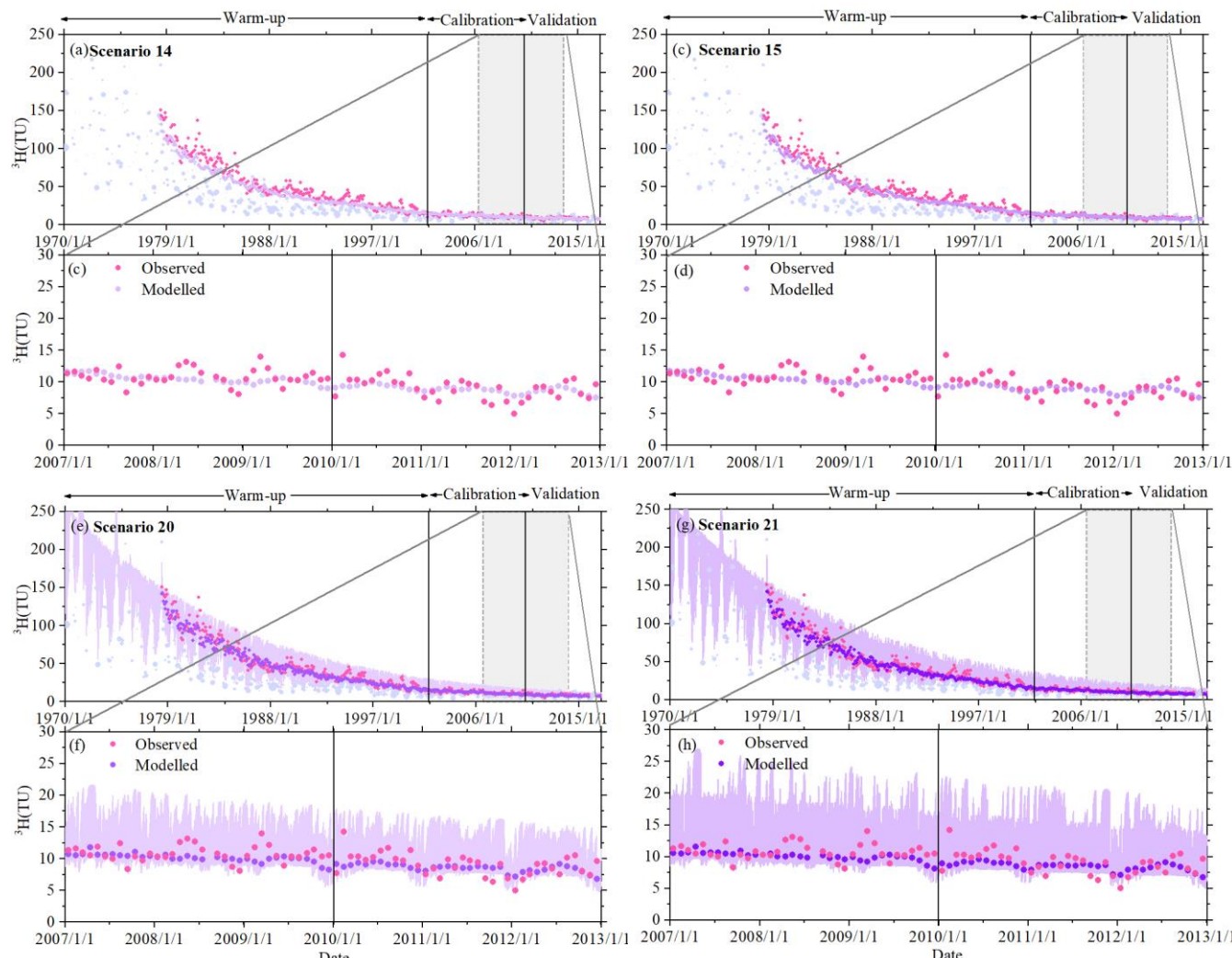

**Figure 4.** The time series of stream $^3$H reproduced by models P-SAS (scenarios 14 and 15) and IM-SAS-D (scenarios 20 and 21) based on different calibration strategies. IM-SAS-D model based on simultaneous calibration to $^3$H and the streamflow signatures, i.e. calibration strategy $C_{^3H,Q}$ (scenario 20 and $C_{\delta^{18}O,^3H,Q}$ (scenario 21), for the model calibration and evaluation periods. (a) Observed $^3$H signals in precipitation (light blue-purple dots; size of dots indicates the precipitation volume) and observed stream $^3$H signals (pink dots) as well as the most balanced modelled $^3$H signal in the stream (light purple dots) for scenario 14 from calibration strategy $C_{^3H}$, (b) zoom-in of observed and modelled $^3$H signals in the stream for the 01/01/2007 – 31/12/2012 period for scenario 14, (c) Observed $^3$H signals in precipitation and in stream same as (a), and the modelled stream $^3$H signals (relatively darker purple dots) for scenario 15 from calibration strategy $C_{\delta^{18}O,^3H}$, (d) zoom-in of observed and modelled $^3$H signals in the stream for the 01/01/2007 – 31/12/2012 period for scenario 15. (e) Observed $^3$H signals in precipitation and in stream same as (a), and the modelled stream $^3$H signals (relatively darker purple dots) for scenario 20 and the 5th/95th percentile of all retained pareto optimal solutions obtained from calibration strategy $C_{^3H,Q}$ (light purple shaded area), (f) zoom-in of observed and modelled $^3$H signals in the stream for the 01/01/2007 – 31/12/2012 period for scenario 20, (g) Observed $^3$H signals in precipitation and in stream same as (a), and the modelled stream $^3$H signals (relatively darker green dots) for scenario 21 and the 5th/95th percentile of all retained pareto optimal solutions obtained from calibration strategy $C_{\delta^{18}O,^3H,Q}$ (light green shaded area), (h) zoom-in of observed and modelled $^3$H signals in the stream for the 01/01/2007 – 31/12/2012 period for scenario 21.

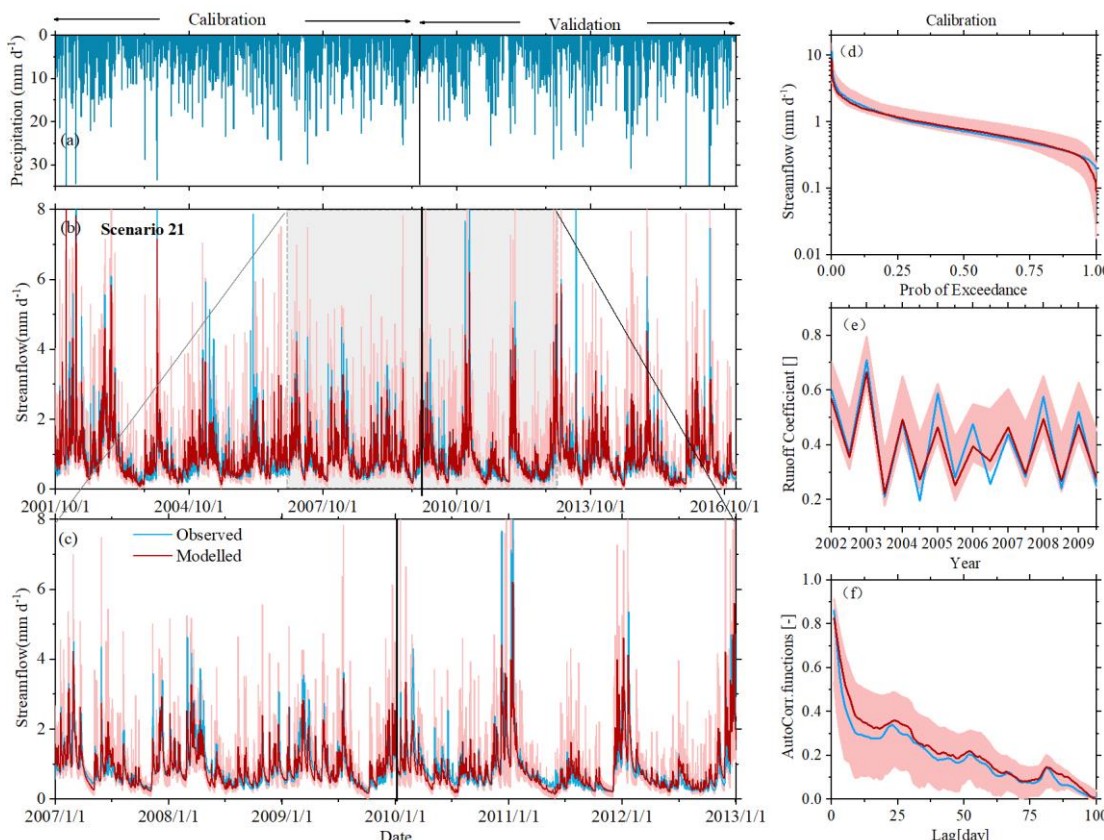

**Figure 5**. Hydrograph and selected hydrological signatures reproduced by IM-SAS-D, following a simultaneous calibration to the hydrological response, $\delta^{18}O$ and $^3H$ ($C_{\delta^{18}O,^3H,Q}$; scenario 21). (a) Time series of observed daily precipitation; observed and modelled (b) daily stream flow (Q), where the red line indicates the most balanced solution, i.e., lowest $D_E$, and the red shaded area the $5^{th}/95^{th}$ inter-quantile range obtained from all pareto optimal solutions; (c) stream flow zoomed-in to the 01/01/2007 – 31/12/2012 period; (d) flow duration curves ($FDC_Q$), (e) seasonal runoff coefficients ($RC_Q$) and (f) autocorrelation functions of stream flow ($AC_Q$) for the calibration period. Blue lines indicate values based on observed streamflow ($Q_o$), red lines are values based on modelled stream flow ($Q_m$) representing the most balanced solutions, i.e., lowest $D_E$ and the red shaded areas show the $5^{th}/95^{th}$ inter-quantile ranges obtained from all pareto optimal solutions.

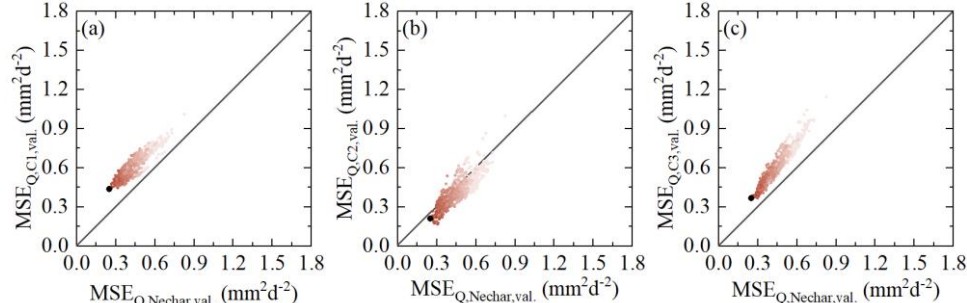

**Figure 6.** Selected model performances in the 01/01/2010 – 31/12/2016 validation period of the overall Neckar basins against the model performance in uncalibrated sub-catchment (a) Kirchentellinsfurt (C1), (b) Calw (C2) and (c) Untergriesheim (C3) based on Scenario 19. The dots indicate all Pareto-optimal solutions in the multi-objective model performance space. The shades from dark to light indicate the overall model performance based on the Euclidean Distance $D_E$, with the black solutions representing the overall better solutions (i.e. smaller $D_E$)

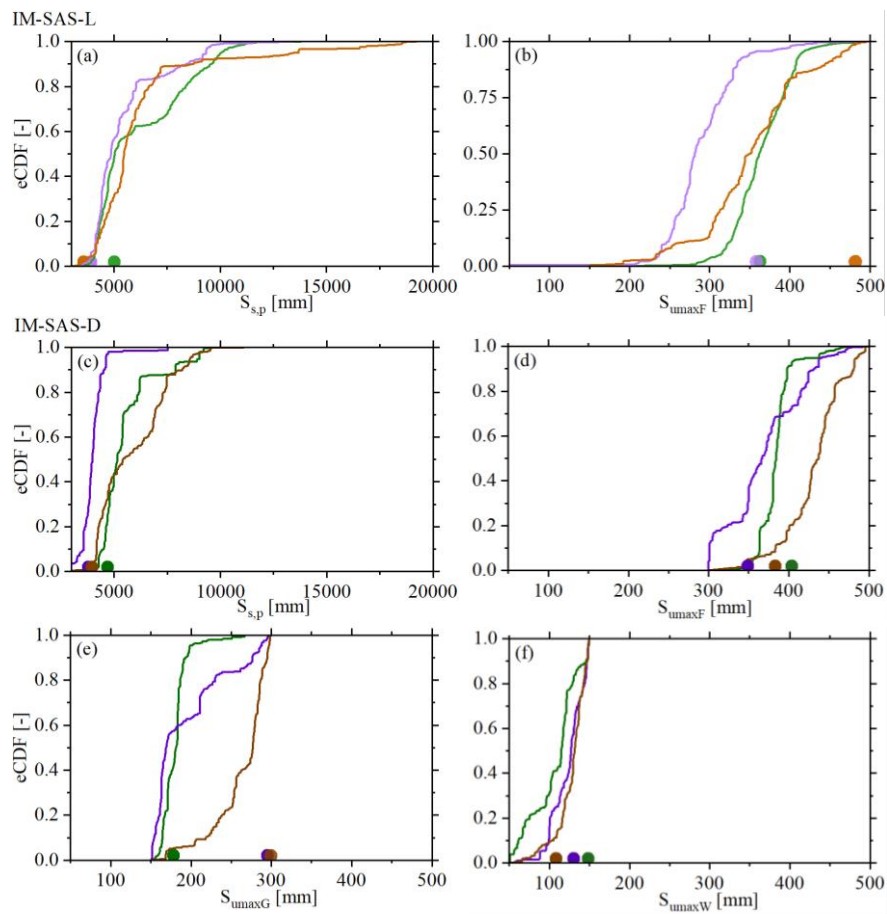

**Figure 7** Pareto-optimal distributions of selected parameters of the IM-SAS models (i.e., IM-SAS-L, IM-SAS-D) shown as the associated empirical cumulative distribution functions (lines). Light green shades indicate scenario 16, light purple shades indicate scenario 17 and light brown shades indicate scenario 18 in (a) and (b); relatively darker green shades indicate scenario 19, relatively darker purple shades indicate scenario 20 and relatively darker brown shades indicate scenario 21 in (c) - (f). The dots indicate the parameter values associated with the most balanced solution, i.e. lowest $D_E$.

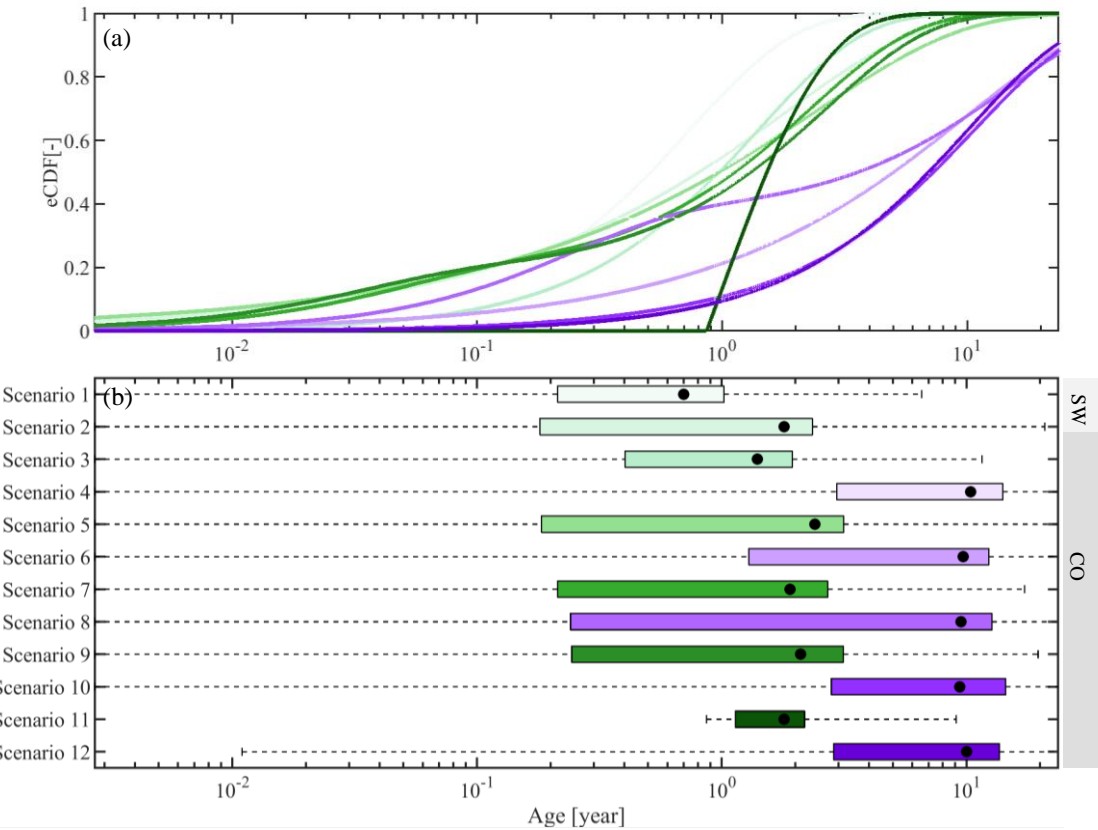

**Figure 8.** Stream flow TTDs derived from the 12 SW/CO model scenarios with the different associated calibration strategies based on different lumped, time-
invariant models. The TTDs represent the best fits of the respective time-invariant TTD. Green shades represent the TTDs inferred from $\delta^{18}O$ (from lighter to
darker for scenarios 1, 2, 3, 5, 7, 9, 11) in (a) and (b); the purple shades represent TTDs inferred from $^3H$ (from lighter to darker for scenario 4, 6, 8, 10 and
12); the black dots in (b) indicate the mean transit time for each model scenario.

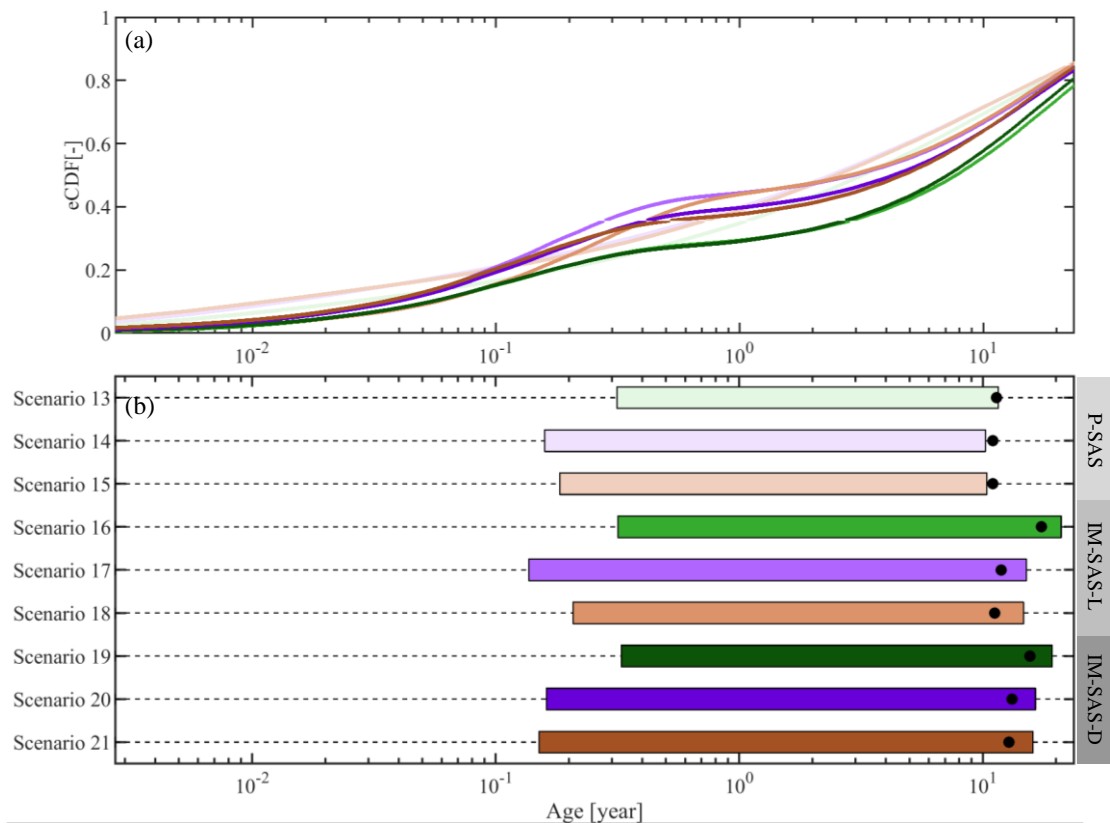

**Figure 9.** Stream flow TTDs derived from the 9 model scenarios with the different associated calibration strategies of P-SAS (scenarios 13 – 15), IM-SAS-L scenarios 16 – 18) and IM-SAS-D model implementations (scenarios 19 – 21). The TTDs represent the volume weighted average daily TTDs for the modelling period 01/10/2001 – 31/12/2016. Green shades represent the TTDs inferred from $\delta^{18}O$ (from lighter to darker for scenario 13, 16, 19), the purple shades represent TTDs inferred from $^3H$ (from lighter to darker for scenario 14, 17, 20), the brown lines represent TTDs inferred from combined $\delta^{18}O$ and $^3H$ (brown shades from lighter to darker for scenario 15, 18, 21); the black dots in (b) indicate the mean transit time for each model scenario. Note that the mean transit time was estimated by fitting Gamma distributions to the volume-weighted mean TTDs of each individual scenario.

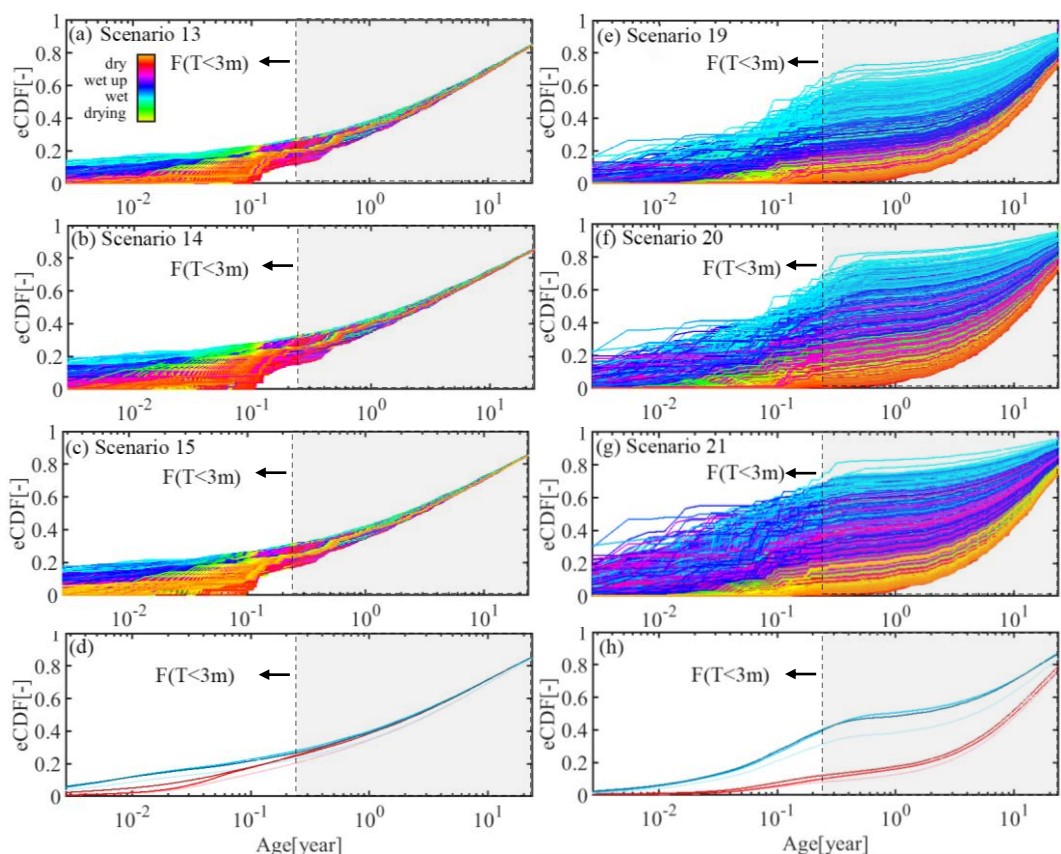

**Figure 10.** Daily streamflow (Q) TTDs extracted from the most balanced model solutions of P-SAS (scenarios 13 – 15) based on (a) calibration strategy $C_{\delta^{18}O}$ (scenario 13), (b) calibration strategy $C^3_H$ (scenario 14) and (c) calibration strategy $C_{\delta^{18}O,^3H}$ (scenario 15), and IM-SAS-D implementations (scenarios 19 – 21), based on (e) calibration strategy $C_{\delta^{18}O,Q}$ (scenario 19), (f) calibration strategy $C^3_{H,Q}$ (scenario 20) and (g) calibration strategy $C_{\delta^{18}O,^3H,Q}$ (scenario 21). The line colors represent the transition between dry and wet periods. Panel (d) shows the volume weighted average TTDs for the wet and dry periods respectively for P-SAS model, the light shades represent calibration strategy $C_{\delta^{18}O}$ (scenario 13), the intermediate shades indicate calibration strategy $C^3_H$ (scenario 14) and the dark shades are calibration strategy $C_{\delta^{18}O,^3H}$ (scenario 15). Panel (h) shows the volume weighted average TTDs for the wet and dry periods respectively for IM-SAS-D model, the light shades represent calibration strategy $C_{\delta^{18}O,Q}$ (scenario 19), the intermediate shades indicate calibration strategy $C^3_{H,Q}$ (scenario 20) and the dark shades are calibration strategy $C_{\delta^{18}O,^3H,Q}$ (scenario 21). For illustrative purposes, also the fraction of water younger than 3 months F (T < 3 m) is indicated.

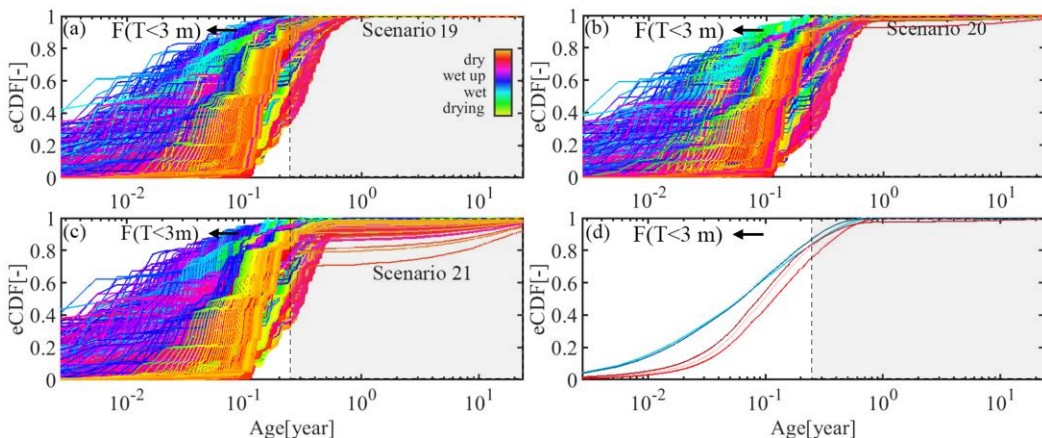

**Figure 11**. Daily transpiration ($E_a$) TTDs extracted from the most balanced model solutions of the IM-SAS-D implementations (scenarios 19 – 21), based on (a) calibration strategy $C_{\delta^{18}O,Q}$ (scenario 19), (b) calibration strategy $C_{^3H,Q}$ (scenario 20) and (c) calibration strategy $C_{\delta^{18}O,^3H,Q}$ (scenario 21). The line colors represent the transition between dry and wet periods. Panel (d) shows the volume weighted average TTDs for the wet and dry periods, respectively. The light shades represent calibration strategy $C_{\delta^{18}O,Q}$ (scenario 19), the intermediate shades indicate calibration strategy $C_{^3H,Q}$ (scenario 20) and the dark shades are calibration strategy $C_{\delta^{18}O,^3H,Q}$ (scenario 21). For illustrative purposes, also the fraction of water younger than 3 months F (T < 3 m) is indicated.

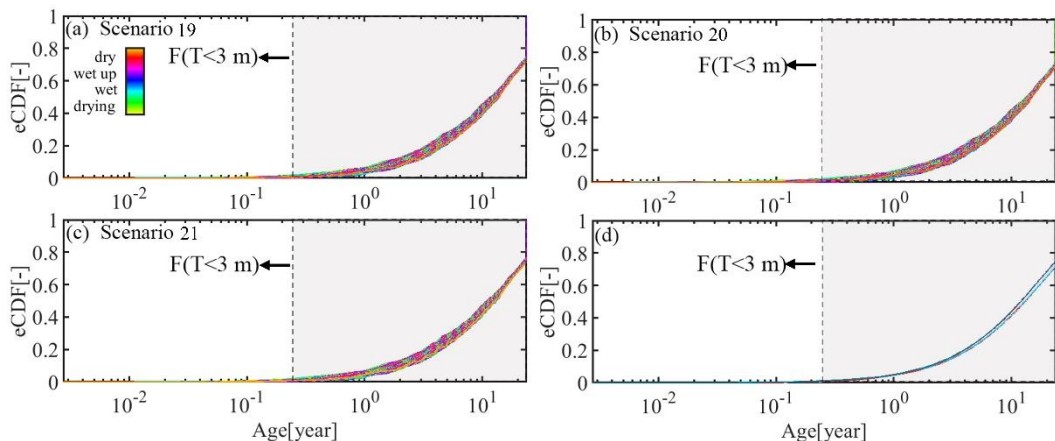

**Figure 12.** Daily groundwater ($S_s$) RTDs extracted from the most balanced model solutions of the IM-SAS-D implementations (scenarios 19 – 21), based on (a) calibration strategy $C_{\delta^{18}O,Q}$ (scenario 19), (b) calibration strategy $C_{^3H,Q}$ (scenario 20) and (c) calibration strategy $C_{\delta^{18}O,^3H,Q}$ (scenario 21). The line colors represent the transition between dry and wet periods. Panel (d) shows the volume weighted average RTDs for the wet and dry periods, respectively. The light shades represent calibration strategy $C_{\delta^{18}O,Q}$ (scenario 19), the intermediate shades indicate calibration strategy $C_{^3H,Q}$ (scenario 20) and the dark shades are calibration strategy $C_{\delta^{18}O,^3H,Q}$ (scenario 21). For illustrative purposes, also the fraction of water younger than 3 months F (T < 3 m) is indicated.