# Peer review of "Stable water isotopes and tritium tracers tell the same tale: No evidence for underestimation of catchment transit times inferred by stable isotopes in SAS function models."

_Hydrology and Earth System Sciences, 2022_

## Referee Comment (RC1)

This manuscript presents a series of catchment scale numerical experiments intended to explore how the choice of tracer data and model type controls the mean transit times and TTDs. The authors applied the O18 and H3 tracers with relatively long time series. They mainly found that the use of O18 tracer and H3 tracer could somehow lead to similar MTT once the right model type was chosen (IM-SAS model). The science questions and approach would appeal to the HESS journal audience and make a nice contribution to understanding the effect of data, model types and in-stream concentrations.

However, I have a major concern that need to be addressed prior to publication:

1)  I suggest to provide more context / justification / details about the calibration procedure – for example, how do you make sure your calibrated best-fits were not local best-fits but globe ones. The best-fit results of different implementations (such as IM-sas-L and IM-sas-D) were similar, but that does not mean the modeled results such as MTT was true. This generally requires an analysis of the potential uncertainty. While I understand a full uncertainty analysis may be unfeasible, the impact of operational choices done in the calibration exercise need to be better discussed.

2)  I do agree with the authors that the H3 and δ18O tracers both are informative for the flow systems, what is needed is just a model good enough to resolve such information in a meaningful way. Especially for the catchments with strong seasonality. However, I am not sure if the model has to use combined date sets of hydrological and tracer as the author argued that "only the combined information using hydrological and tracer data and the consideration of transient flow conditions gives similar MTT, independent of the used tracer". I think the important thing is that the flow model can represent the reality in a good way, such that the tracer transport can be well reproduced. Using hydrological data in calibration may not a key control for that.

Other comments with line number:

Line 160: What are Ep and P?

Line 368: perhaps say that the storage component is just locally full-mixed and those local full mixtures do not lead to an overall fully mixed system…

Line 373: I don't think that to reduce computational time and computer memory requirements is good reason for using uniform sas functions rather than other shapes of sas function. I think the right way should be describing the model of reduced complexity (parameters) was already enough for your modelling targets.

Line 378: could you explain in more detail how was the tracer sampled from the passive and active volumes? Also random sampling from $S_{s,tot}$ ?

Line 393-395: maybe simply say the lumped implementation used a single HUR to represent the entire basin. Is that what you mean? In this case the precipitation zones were not used any more, right? Maybe clarify this.

Equation 14: what are $E_{mse,Q,n}$ and $E_{mse,tracer,m}$?

Line 473: it looks like that when using all the data, the lumped model (scenario 9) was even better than the distributed model (scenario 12) that has more parameters, does that mean the high model complexity is not essential for a better model performance in your case, could you clarify that

Line 508: Table 3?

---

## Referee Comment (RC2)

**Review of hess-2022-400 – *Stable water isotopes and tritium tracers tell the same tale: No evidence for underestimation of catchment transit times inferred by stable isotopes in SAS function models.**

The authors present an analysis of water ages obtained by different transit time (TT) models and two different tracers (i.e., stable water isotopes and tritium) in the Neckar basin (Germany). They test the common assertion that water ages above around five years cannot be well identified in TT models using stable water isotopes as model tracer, as opposed to tritium. They compare mean transit times (MTTs) as TT metric calculated via the sine-wave (SW) approach, convolution integral models (CO) and a process-based hydrological model using Storage Selection (SAS) functions. As MTT estimates substantially differ between the two tracers for SW and CO models, but are similar for the SAS model, the authors conclude that stable water isotopes are just as well fit for identifying old water ages as tritium and that differences in estimated water ages for the two tracers are an artifact of the SW and CO models.

**General comments**

The study fits the scope of HESS and makes a valuable contribution to the field of transit time modelling and tracer hydrology. Illustrating the capacity of stable water isotopes to quantify older water will open up new opportunities for TT modelling in catchments that are assumed to show comparably large MTTs. Hence, I support the general motivation and objectives of the study. However, I have some concerns related to the modelling approach and study basin used:

First, I am not sure whether a catchment (river basin) of 13,000 km² with at the same time limited availability of tracer data is the best choice for the study objectives. While individual controls on TTs remains largely elusive, it has been shown that TTs (or their metrics) vary widely depending on catchment characteristics such as elevation, topography or climate (e.g., Jasecko et al., 2016, Kumar et al., 2020). Modelling TTs in a river basin that shows a gradient of more than 800 mm yr$^{-1}$ in annual precipitation, an elevation gradient of around 900 meters and varying land use types adds a lot of complexity that could have been avoided when using a much smaller and more homogeneous catchment. At the same time, the study relies on only one precipitation station for both stable water isotopes and tritium (within the basin) providing monthly composite samples. Hence, the tracer data are rather sparse both temporally and spatially, which adds another layer of uncertainty to the modelling. An alternative might be to compile data from previous TT modelling approaches that have been conducted in smaller catchments with more highly-resolved (space and/or time) stable water isotope and tritium data (e.g., Rodriguez et al., 2021 – reference already in manuscript).

Secondly, there is a remarkably great difference in model complexities between the individual TT modelling approaches. On the one hand, simple CO models with only one compartment, no temporal/seasonal variation and two pre-defined shape parameters for the TTs have been used, while on the other hand, the SAS model consists of three hydrological response units with multiple storage volumes each, has 11 calibration parameters and is also tested in a spatially distributed implementation. As the authors are clearly aware of, time-variant concepts of CO models (see Hrachowitz et al., 2010; and references cited therein) as well as multi-compartment models representing fast and slow flow routes have been used; using especially the latter is a common approach in CO modelling. Moreover, the SAS model with its comparably large number of parameters is calibrated simultaneously to discharge and at least one of the two tracers, while the CO models are calibrated to only one tracer. I am thus wondering

to what extent results from these TT models can be compared at all. I understand that the objective of this paper is not to dismiss a specific model type, but rather to analyse the flexibility of stable water isotopes as TT model tracers. However, this requires to use model setups and data similar to those used in the papers that have demonstrated the truncation of TT distributions by calibration to stable water isotopes. To address this concern, one could think of (i) focussing on a smaller (or even headwater) catchment with preferably daily tracer data, (ii) using established CO models such as the more complex ones in Stewart et al. (2010), and (iii) using measured and modelled P, ET, storage and Q data as input for SAS modelling (potentially with non-random sampling) with one or a maximum of two SAS function compartments, as commonly done in more recent SAS modelling studies (e.g., Benettin et al., 2017; Harman, 2015; Nguyen et al., 2021).

Thirdly, the fact that spatial aggregation introduces bias in CO model-based MTTs, as stated also by the authors, raises the question to what extent comparison of MTT estimates is meaningful. I understand that the authors would like to test the validity of stable water isotopes in TT modelling particularly of *older* water ages, and that MTT has been a metric commonly reported for CO models. Nonetheless, according to Kirchner (2016 – reference already in manuscript), sine-wave fitting to seasonal isotope data does give robust estimates of the young water fraction $F_{yw}$. Hence, it might be more meaningful to compare $F_{yw}$ estimates by the different TT model approaches, or, even better, to add this as further TT metric in the comparison.

Finally, I would highly appreciate if the authors could increase traceability of their results and provide the underlying tracer data as well as model codes. Traceability is one of the main criteria for HESS nowadays and given that the authors address such a fundamental claim in tracer hydrology and TT modelling, I find it necessary for the entire TT community to benefit from this study not only via the paper, but also in terms of data and code accessibility.

Overall, I would be glad to see how the authors address the above mentioned methodological issues. Please see below for some specific comments and suggestions.

**Specific comments**

- Lines 35—37: if this refers to the findings by Kirchner (2016), one could be more precise by specifying that the *MTT* (as commonly reported metric) derived from CO models is affected by spatial aggregation errors.
- Line 59: in what sense is there more coherence?
- Line 70: does Cl⁻ have a clear seasonal cycle? I assume both weathering and anthropogenic effects (e.g., application of road salt) govern its concentrations. Another possible distinction would be radioactive vs. conservative tracers.
- Lines 80—98: the focus on the amplitude ratio for the "traditional" TT approaches is fine for simple one-compartment gamma (and thus also exponential) models, but is this also relevant for multiple-compartment CO models and other pre-defined TT shapes such as the dispersion model? This suggests that CO models are exclusively based on the amplitude ratio and shift in seasonal isotope ratios.
- Lines 84—85: "practically" and "feasibly" twice?
- Lines 97: to what extent could a spatial aggregation bias also affect spatially lumped (one-compartment) SAS models?

- Lines 197: you used the CORINE dataset from 2018. To what extent has land use remained stable since 2001?
- Line 374: we do not necessarily see passive storage volumes in the most recent SAS model studies.
- Lines 398—414: I am wondering to what extent we can trust the spatially distributed implementation, given that there is only one calibration gauge at the outlet of the entire catchment. This also relates to my general comment about the considerable size and few data for the study basin.
- Line 411: could you specify what the distributed moisture accounting approach is?
- Lines 420—421: why have the authors not applied a multi-objective calibration to the CO models?
- Line 424: this is interesting but I think, as stated in my general comments, that TTs should be obtained from a SAS model with storage, input and output fluxes defined a priori (as if they were "real" data), rather than computing TTs from simultaneous calibration against flow and tracers. I think that this would be a more straightforward methodology given the scope of TT modelling and tracers. As presented here, we do not know to what extent simulated TTs are affected by equifinality in the hydrological model parameters.
- Lines 438—439: would normalisation of the errors help?
- Lines 553—555: not a complete sentence
- Line 571: not only, but also…?
- Lines 577—578: I think you could easily implement the multi-objective calibration for the CO models as well.
- Lines 619—620: so here one could at least test how time-variant/seasonal CO models perform
- Lines 642—644: could this not be an indication of the fact that there are too many degrees of freedom and the model always succeeds to fit the tracer data, regardless of whether it is spatially lumped or semi-distributed?
- Lines 656—657: see, e.g., Nguyen et al. (2022) who found substantial differences in SAS-based transport models between spatially lumped and semi-distributed setup.

**References**

Benettin, P., Soulsby, C., Birkel, C., Tetzlaff, D., Botter, G., and Rinaldo, A. (2017), Using SAS functions and high-resolution isotope data to unravel travel time distributions in headwater catchments, Water Resour. Res., 53, 1864– 1878, doi:10.1002/2016WR020117.

Harman, C. J. (2015), Time-variable transit time distributions and transport: Theory and application to storage-dependent transport of chloride in a watershed, Water Resour. Res., 51, 1– 30, doi:10.1002/2014WR015707.

Hrachowitz, M., Soulsby, C., Tetzlaff, D., Malcolm, I. A., and Schoups, G. (2010), Gamma distribution models for transit time estimation in catchments: Physical interpretation of parameters and implications for time-variant transit time assessment, Water Resour. Res., 46, W10536, doi:10.1029/2010WR009148.

Jasechko, S., Kirchner, J. W., Welker, J. M., & McDonnell, J. J. (2016). Substantial proportion of global streamflow less than three months old. Nature Geoscience, 9(2), 126–129. https://doi.org/10.1038/ngeo2636

Kumar, R., Heße, F., Rao, P.S.C. et al. Strong hydroclimatic controls on vulnerability to subsurface nitrate contamination across Europe. Nat Commun 11, 6302 (2020). https://doi.org/10.1038/s41467-020-19955-8

Nguyen, T. V., Kumar, R., Musolff, A., Lutz, S. R., Sarrazin, F., Attinger, S., & Fleckenstein, J. H. (2022). Disparate seasonal nitrate export from nested heterogeneous subcatchments revealed with StorAge Selection functions. Water Resources Research, 58, e2021WR030797. https://doi.org/10.1029/2021WR030797

Nguyen, T. V., Kumar, R., Lutz, S. R., Musolff, A., Yang, J., & Fleckenstein, J. H. (2021). Modeling nitrate export from a mesoscale catchment using storage selection functions. Water Resources Research, 57, e2020WR028490. https://doi.org/10.1029/2020WR028490

---

## Community Comment (CC1)

[supplement omitted: unrelated document]

---

## Author Comment (AC1)

We highly appreciate the time and effort that the Reviewer has dedicated to providing feedback on our manuscript and are grateful for the insightful comments on our manuscript. Please find below our detailed replies to the individual comments.

**(1) Reviewer Comment:**

*I suggest to provide more context / justification / details about the calibration procedure – for example, how do you make sure your calibrated best-fits were not local best-fits but globe ones. The best-fit results of different implementations (such as IM-SAS-L and IM-SAS-D) were similar, but that does not mean the modeled results such as MTT was true. This generally requires an analysis of the potential uncertainty. While I understand a full uncertainty analysis may be unfeasible, the impact of operational choices done in the calibration exercise need to be better discussed.*

**Reply:**

We completely agree with this point. We have therefore done an uncertainty analysis to quantify the effects of parameter uncertainty on the modelled TTDs by randomly sampling from the posterior parameter distributions for both, IM-SAS-L and IM-SAS-D models. While parameter uncertainty can cause some variability in TTDs and thus in the actual magnitudes of water ages, this variability is consistently within similar age ranges for $^{18}$O and $^{3}$H, respectively. It does therefore not affect the overall interpretation of the results and the rejection of the hypothesis that $^{18}$O underestimates water ages, as shown for scenarios 10-12 in Figure FR1 here below. We will add these results in the revised manuscript.

[Figure]

**Figure FR1.** Stream flow TTDs derived from the 6 model scenarios based on IM-SAS models with the different associated calibration strategies (scenarios 10-12). Each line represents the volume weighted average daily TTDs during the modelling period 01/10/2001 – 31/12/2016, generated from parameters randomly sampled from the posterior distribution (light shades) and the most balanced solution of each scenario (dark shades). (a) TTDs inferred from δ$^{18}$O in scenario 10; (b) TTDs inferred from $^{3}$H in scenario 11; (c) The TTDs inferred from combined δ$^{18}$O and $^{3}$H in scenario 12.

**(2) Reviewer Comment:**

*I do agree with the authors that the $^3H$ and $\delta^{18}O$ tracers both are informative for the flow systems, what is needed is just a model good enough to resolve such information in a meaningful way. Especially for the catchments with strong seasonality. However, I am not sure if the model has to use combined date sets of hydrological and tracer as the author argued that "only the combined information using hydrological and tracer data and the consideration of transient flow conditions gives similar MTT, independent of the used tracer". I think the important thing is that the flow model can represent the reality in a good way, such that the tracer transport can be well reproduced. Using hydrological data in calibration may not a key control for that.*

**Reply:**

We agree with this point. We will therefore reformulate that sentence on P.20, l.620, "only the combined information using hydrological and tracer data and the consideration of transient flow conditions gives similar MTT, independent of the used tracer" in the revised manuscript so that it better reflects that point.

**(3) Reviewer Comment:**

*Line 160: What are Ep and P?*

**Reply:**

Thank you for pointing this out. While Ep represents potential evaporation, P represents precipitation. We will add the definitions in the revised manuscript.

**(4) Reviewer Comment:**

*Line 368: perhaps say that the storage component is just locally full-mixed and those local full mixtures do not lead to an overall fully mixed system*

**Reply:**

We completely agree with this suggestion. It was mentioned on P.12, L.368ff, but we will make it clearer in the revised manuscript.

**(5) Reviewer Comment:**

*I don't think that to reduce computational time and computer memory requirements is good reason for using uniform SAS functions rather than other shapes of SAS function. I think the right way should be describing the model of reduced complexity (parameters) was already enough for your modelling targets.*

**Reply:**

We agree with the argument that reduced complexity here already allows to draw robust conclusions. We will reformulate the statement and add this aspect. However, we would also like to explicitly re-iterate

here that computational capacity imposes major practical obstacles to testing other SAS function shapes: in contrast to uniform distributions, the sampling process then requires an explicit generation of RTDs and TTDs for each time step and to "carry" all RTDs and TTDs of all model components through the entire model period, including the warm-up period (here: 46 years). This entails for a daily modelling time-step the simultaneous handling of multiple matrices > 16.800x16.800 elements in floating number format (i.e. 8B each), which corresponds to >2 GB/matrix. With a working memory of common but good computers (i.e. 16-32 GB) this means that the generation of RTDs and TTDs alone will use (if not exceed) the memory of these computers, not to speak of other processes required.

**(6) Reviewer Comment:**

*Line 378: could you explain in more detail how was the tracer sampled from the passive and active volumes? Also random sampling from Ss,tot ?*

**Reply:**

The tracer and age composition of that outflow is indeed randomly sampled from the total groundwater storage volume $S_{S,tot}$. We will clarify this in the revised manuscript.

**(7) Reviewer Comment:**

*Line 393-395: maybe simply say the lumped implementation used a single HUR to represent the entire basin. Is that what you mean? In this case the precipitation zones were not used any more, right? Maybe clarify this.*

**Reply:**

Indeed, the lumped implementation used a single HRU (equivalent to the forest HRU described in distributed model, Fig.2) to represent the entire catchment and the precipitation zones were not used any more in this lumped case. We have will clarify this in the revised manuscript.

**(8) Reviewer Comment:**

*Equation 14: what are $E_{mse,Q,n}$ and $E_{mse,tracer,m}$?*

**Reply:**

Thank you for pointing this out. We will add the missing definitions in the revised manuscript.

**(9) Reviewer Comment:**

*Line 473: it looks like that when using all the data, the lumped model (scenario 9) was even better than the distributed model (scenario 12) that has more parameters, does that mean the high model complexity is not essential for a better model performance in your case, could you clarify that.*

**Reply:**

This is an interesting aspect. However, while the distributed implementation IM-SAS-D can indeed not be considered to outperform the lumped IM-SAS-L implementation, the opposite cannot be concluded either: as can be seen in Table 4, considering the most balanced solution, some signatures were indeed captured better by IM-SAS-L than by IM-SAS-D. Yet, others were much better reproduced by IM-SAS-D. In addition, it can be seen that the full set of pareto front solutions of IM-SAS-L includes a considerable number with poorer performance metrics (i.e. upper limit of performance ranges shown in Table S5 in the Supplementary Material).

*(10) Reviewer Comment:*

*Line 508: Table 3?*

**Reply:**

Indeed. We will correct that.

---

## Author Comment (AC2)

**_(1) Reviewer Comment:_**

_The study fits the scope of HESS and makes a valuable contribution to the field of transit time modelling and tracer hydrology. Illustrating the capacity of stable water isotopes to quantify older water will open up new opportunities for TT modelling in catchments that are assumed to show comparably large MTTs. Hence, I support the general motivation and objectives of the study._

**Reply:**

We highly appreciate this positive overall assessment of our work and we thank the reviewer for her interest in our work as well as for the thoughtful and detailed comments that helped to strengthen our analysis. Below, we provide clarifications and our perspectives to respond in detail to the individual reviewer comments.

**_(2) Reviewer Comment:_**

_First, I am not sure whether a catchment (river basin) of 13,000 km² with at the same time limited availability of tracer data is the best choice for the study objectives. While individual controls on TTs remains largely elusive, it has been shown that TTs (or their metrics) vary widely depending on catchment characteristics such as elevation, topography or climate (e.g., Jasecko et al., 2016, Kumar et al., 2020). Modelling TTs in a river basin that shows a gradient of more than 800 mm yr-1 in annual precipitation, an elevation gradient of around 900 meters and varying land use types adds a lot of complexity that could have been avoided when using a much smaller and more homogeneous catchment. At the same time, the study relies on only one precipitation station for both stable water isotopes and tritium (within the basin) providing monthly composite samples. Hence, the tracer data are rather sparse both temporally and spatially, which adds another layer of uncertainty to the modelling. An alternative might be to compile data from previous TT modelling approaches that have been conducted in smaller catchments with more highly-resolved (space and/or time) stable water isotope and tritium data (e.g., Rodriguez et al., 2021 – reference already in manuscript)._

**Reply:**

_Choice of study region_

We agree that it remains a defining challenge in hydrology to fully account for heterogeneities in larger systems. Unfortunately, there is no "silver bullet" to solve that problem. This is also explicitly discussed in the Discussion section of our manuscript (p.21, l.658ff). While we share the reviewer's view that studies at smaller scales are very important, these types of studies typically suffer from other limitations. Specifically for the case of stable isotope and tritium comparisons and apart from the fact that there are hardly any catchments world-wide in which data for both tracers are available, the study cited by the Reviewer (Rodriguez et al., 2021) is indeed conducted in a smaller catchment with higher tracer sampling frequency. _However_, and as explicitly mentioned in the manuscript (p.4, l.132-150), it relies on much shorter time series, i.e. 2 years, and only a handful of tritium samples, i.e. 24. In addition, conclusions from that study on the ability of stable isotopes to see older water may be hampered by the potential \*absence\* of older water. In other words, if there is no older water present in a catchment, stable isotopes

can also not see it, as recently pointed out by Stewart et al. (2021). We therefore believe, that in spite of potential uncertainties arising from the size of the system, our study allows us to explore aspects of the research question that could not (or not fully) be addressed by Rodriguez et al. (2021).

*Role of heterogeneity for older water ages – catchment as low-pass filter*

It is also important to note that in our study we are mostly interested in older water ages. As catchments act as low-pass filters, they already smooth out much of short time-scale and small spatial-scale hydro-climatic variability. The remaining higher-frequency components in the response, e.g. responses to individual rain events, then mostly affect water ages at the younger side of the spectrum. These can indeed be sensitive to spatial-temporal heterogeneities. In contrast, older water ages are mostly controlled by low frequency components of the system and thus variabilities at much larger spatial and longer temporal scales, e.g. seasonal or inter-annual changes in groundwater tables, and are thus much less sensitive to small-scale heterogeneities. This can for example be seen in the significant differences between the power spectra of stream tracer concentrations of fast responding parts of the system (i.e. short time-scales, high-frequency components and thus younger water ages) and groundwater tracer concentrations (i.e. much longer time-scales, low-frequency components of the system and older water ages), as for example demonstrated by Hrachowitz et al. (2015; Figure 8 therein) and which define the recurrently described, very characteristic 1/f scaling of stream tracer responses across many system in contrasting environmental settings across the world (e.g. Kirchner et al., 2001; Godsey et al., 2009; Hrachowitz et al., 2009; Aubert et al., 2014; Kirchner and Neal, 2013). Another piece of evidence for the lower sensitivity of older water to heterogeneity is the higher sensitivity of high-frequency components and younger water ages to hydro-climatic variability (e.g. Figure 9 in our original manuscript) as compared to the almost complete lack sensitivity to hydro-climatic in low-frequency components and thus older water (e.g. Figure 10), which has also been reported in many other studies (e.g. Hrachowitz et al., 2013, 2015; Soulsby et al., 2015). Overall, this means that while the pattern and dynamics of young water ages may indeed to some degree be affected by heterogeneities within our study basin, it is plausible to assume that they have only minor impact on the estimation of older water ages. *We will add a more detailed discussion on this in the revised version of the manuscript.*

*Spatial representation of hydro-climatic and tracer input heterogeneity in the study*

Notwithstanding the above and to limit adverse effects of a coarser data resolution, we here invested considerable effort into spatial adjustments of hydro-meteorological input data as well as tracer data, according to the best available information in our distributed model implementation. While the major spatial differences in precipitation are accounted for by the identification and use of four individual precipitation zones, major spatial differences in temperature (and thus also in EP) are accounted for by the additional stratification into 100m elevation zones as described in Sections 3.2.1 and 4.2.2. Similarly and more importantly, the tracer input signals were spatially adjusted, as described in Sections 3.2.2 and 3.2.3 as well as in the Supplement, following the method recently developed by Allen et al. (2018, 2019). This method identified strong relationships between multiple catchment characteristics and seasonal stable isotope signals in precipitation. These relationships thus allow a robust estimation of the spatial differences in stable isotope input, both globally (Allen et al., 2019) and perhaps more importantly, also

regionally, as demonstrated in Allen et al. (2018) who quantified spatial stable isotope input for Switzerland, which is just across the border from our study basin in Southern Germany. A comparable approach was applied for precipitation tritium concentrations, which in any case do not exhibit major spatial differences (e.g. Schmidt et al., 2020). The same applies also to water stable isotopes in precipitation for monthly sampling resolution as indicated by the similarity to isotopes for stations close by, i.e. Karlsruhe (Stumpp et al. 2014).

*Ability of the model to represent the response and spatial heterogeneity therein*

To reduce the potential of misrepresentations of the system and its heterogeneities by the model we have deliberately chosen to expose the model to a rigorous calibration and post-calibration evaluation procedure that goes far beyond what is done in the vast majority of studies in scientific hydrology. The use of *eight different performance indicators*, that describe the models' ability to simultaneously reproduce distinct signatures and thus distinct aspects of the system response, allowed to identify and discard solutions that in traditional model calibration/evaluation procedures, based on one or two performance metrics, would have been falsely accepted as feasible. This leads to a robust representation of the system, as can be seen by the models' ability to relatively well and simultaneously reproduce these multiple signatures – both, in the calibration as well as and more importantly in the post-calibration evaluation ("validation") periods as illustrated by Figures 3-5 and Table 4 in the original manuscript and also illustrated here below in Figure FR1, for the example of stream flow Q in Scenario 12.

[Figure]

**Figure FR1.** Model performance of all pareto-optimal solutions accepted as feasible against to reproduce stream flow Q in model calibration vs. model evaluation periods based on the mean squared error ($MSE_Q$). The dark dot indicates the most balanced pareto-optimal solution. The fact that all solutions plot very close to the 1:1 line suggests that the model does reproduce Q in the model post-calibration evaluation period ("validation") almost as good as in the calibration period. This is a strong indicator of the model being a plausible representation of the system response.

However and in addition to the strict model evaluation procedure in our original manuscript, we have taken this concern of the reviewer very serious and decided to *confront the model with additional observations* to further test its ability to meaningfully represent spatial differences in the response. To do so, we have now also evaluated the model outputs against streamflow observations in three sub-catchments (C1: Kirchentellinsfurt, C2: Calw, and C3: Kocherstetten) within the Neckar basin, whereby each one of them largely represents the response from one of the precipitation zones (Figure FR2 here below).

[Figure]

Figure FR2: (a) Sub-catchments C1 – C3 within the Neckar basin used to evaluate the model performance, (b) model performance in the Neckar basin vs. sub-catchment C1, (c) Neckar vs. C2 and (d) Neckar vs. C3, based on Scenario 10. The dots indicate all Pareto-optimal solutions in the multi-objective model performance space. The shades from dark to light indicate the overall model performance based on the Euclidean Distance $D_E$, with the darker solutions representing the overall better solutions (i.e. smaller $D_E$)

It can be seen, that the model calibrated on stream flow of the entire Neckar basin can reproduce stream flow in the 3 sub-catchments similarly well, with C2 and C3 even better reproduced with many of the solutions than the calibrated Neckar stream flow. These results suggest that the model does indeed pick up the major differences in response types due to hydro-climatic heterogeneities throughout the Neckar basin. Together with the spatial adjustments of the tracer inputs as described above, this is further evidence that the model provides an adequate representation of the major features of the hydrological response even at the larger scale of the Neckar basin and therefore also a meaningful spatial representation of the tracer circulation. We will add these additional model tests to the manuscript to better demonstrate the suitability of the model for our study.

Overall, we can and do not claim that our models generate the best possible TTD estimates. Rather, our intention in this analysis is to show the consistency between TTD estimates derived from stable isotopes and tritium, i.e. that both contain enough and comparable information which can be exploited to estimate water ages. In other words, even if TTD estimates of both tracers are subject to uncertainties, the fact that they provide similar TTD estimates when used in the same model type is evidence for a similar information content, supporting the notion that stable isotopes have indeed the potential to see older water, if used in conjunction with suitable modelling approaches. This is explicitly discussed in the text (p.19, l.600ff in the original manuscript).

*__(3) Reviewer Comment:__*

*Secondly, there is a remarkably great difference in model complexities between the individual TT modelling approaches. On the one hand, simple CO models with only one compartment, no temporal/seasonal variation and two pre-defined shape parameters for the TTs have been used, while on the other hand, the SAS model consists of three hydrological response units with multiple storage volumes each, has 11 calibration parameters and is also tested in a spatially distributed implementation. As the authors are*

*clearly aware of, time-variant concepts of CO models (see Hrachowitz et al., 2010; and references cited therein) as well as multi-compartment models representing fast and slow flow routes have been used; using especially the latter is a common approach in CO modelling. Moreover, the SAS model with its comparably large number of parameters is calibrated simultaneously to discharge and at least one of the two tracers, while the CO models are calibrated to only one tracer. I am thus wondering to what extent results from these TT models can be compared at all. I understand that the objective of this paper is not to dismiss a specific model type, but rather to analyse the flexibility of stable water isotopes as TT model tracers. However, this requires to use model setups and data similar to those used in the papers that have demonstrated the truncation of TT distributions by calibration to stable water isotopes. To address this concern, one could think of (i) focussing on a smaller (or even headwater) catchment with preferably daily tracer data, (ii) using established CO models such as the more complex ones in Stewart et al. (2010), and (iii) using measured and modelled P, ET, storage and Q data as input for SAS modelling (potentially with non-random sampling) with one or a maximum of two SAS function compartments, as commonly done in more recent SAS modelling studies (e.g., Benettin et al., 2017; Harman, 2015; Nguyen et al., 2021).*

**Reply:**

We agree with the reviewer that the model approaches are different and we also agree that comparisons need to be consistent and systematic to be meaningful.

However, we also want to point out here – as correctly mentioned by the reviewer – that the objective of our analysis is to analyse the potential of stable isotopes to see older water and not a full-fledged comparison of different model approaches. This is explicitly stated in the research hypothesis "[…] that $^{18}$O as tracer generally and systematically cannot detect tails in water age distributions and that this truncation leads to systematically younger water age estimates than the use of $^{3}$H" (p.5, l.151-152)

Please note that therefore what is actually compared here are models of the *same type* (and same complexity) run with stable isotopes and subsequently with tritium. The comparison is not made between models of different types and/or complexities. In other words, we compare water age estimates obtained from e.g. a CO model with exponential TTD run with $^{18}$O with those obtained from the same model but run with $^{3}$H. In contrast, we do not compare water ages from that CO model with ages estimated from another model, e.g. IM-SAS. This is also emphasized by the last four columns of table 5.

To further clarify, we have estimated water ages based on CO models to check if we would find differences in water ages between $^{18}$O- and $^{3}$H-based model runs in the study basin, using the *same* types of *lumped*, *time-invariant* models that Stewart et al. (2010) based their argument on. The fact that we found significant differences between these estimates, would, without further analysis, further support the observation of Stewart et al. (2010) that $^{18}$O *generally* truncates water ages.

Our intention is *not* to show that CO models are generally not capable to estimate older ages. Perhaps, time-variant implementations can do that very well, but exploring this was not the objective of our study. Also the combined use of $^{18}$O and $^{3}$H in CO models has previously been shown to be useful to estimate older ages. But this is outside the scope of our study. Instead, as clearly stated in the research hypothesis, we test if $^{18}$O can generally be considered to be useless for the determination of ages older than ~4 years. Our results then further suggest, that, if used in combination with IM-SAS models, the hypothesis needs to be rejected, as these models produce similar water ages with $^{18}$O and $^{3}$H that are much older than 4 years. Given that the results of Stewart et al. (2010) as well as our own CO scenarios are based on lumped,

time-invariant CO model implementations, our results eventually also allow the observation that the perceived failure of $^{18}O$ to see older ages is not a general limitation of that tracer, but rather a consequence of its use in *lumped*, *time-invariant* CO models.

However, we agree with the reviewer that we have not tested the more complex CO model implementations from Stewart et al. (2010) in our original manuscript. We therefore took up this advice of the reviewer and did additional model runs, with full calibrations (and evaluations) of a wider range of common time-invariant implementations of CO models, also including more complex ones. Our analysis now includes in addition to exponential (EM) and gamma (GM) models also two parallel reservoir (2EM; scenarios X1-2), three parallel reservoir (3EM; scenarios X3-4) and exponential piston flow (EPM, scenarios X5-6) implementations. The TTD estimates from these additional model implementations are consistent with those in the original analysis: for all tested lumped, time-invariant CO models, the TTDs derived from $^{18}O$ indicated with MTTs ~ 1-2 yrs significantly younger water than those derived from $^{3}H$, which suggest MTTs ~10 yrs throughout (see Table TR1 and Figure FR3 below). This further strengthens our previous results, suggesting that $^{18}O$ when used in lumped, time-invariant CO models underestimates water ages, as suggested by Stewart et al. (2010).

**Table TR1.** Metrics of stream flow TTDs derived from the 10 model scenarios with the different associated calibration strategies based on different CO models, where $C_{\delta^{18}O}$ indicates calibration to $\delta^{18}O$, $C_{^3H}$ calibration to $^{3}H$. The TTD metrics represent the best fits of the respective time-invariant TTD. The water fractions are shown as the fractions of below a specific age T. The columns with absolute difference $\Delta$ illustrate the differences in TTDs from the same models calibrated to $\delta^{18}O$ and $^{3}H$, respectively. The subscripts indicate the scenarios that are compared (e.g., $\Delta_{3,4}$ compares scenarios 3 and 4).

| Scenario | | 3 | 4 | 5 | 6 | X1 | X2 | X3 | X4 | X5 | X6 | $\Delta_{3,4}$ | $\Delta_{5,6}$ | $\Delta_{X1,X2}$ | $\Delta_{X3,X4}$ | $\Delta_{X5,X6}$ |
|---|---|---|---|---|---|---|---|---|---|---|---|---|---|---|---|---|
| Model | | CO-EM | | CO-GM | | CO-2EM | | CO-3EM | | CO-EPM | | Absolute difference | | | | |
| Calibration strategy → TTD metrics ↓ | | $C_{\delta^{18}O}$ | $C_{^3H}$ | $C_{\delta^{18}O}$ | $C_{^3H}$ | $C_{\delta^{18}O}$ | $C_{^3H}$ | $C_{\delta^{18}O}$ | $C_{^3H}$ | $C_{\delta^{18}O}$ | $C_{^3H}$ | $\Delta TT_{\delta^{18}O-^3H}$ $\Delta F(T<x)_{\delta^{18}O-^3H}$ | | | | |
| | Mean (yr) | 1.4 | 10.4 | 2.4 | 9.7 | 1.9 | 9.5 | 2.1 | 9.4 | 1.8 | 10 | -9.0 | -7.3 | -7.6 | -7.3 | -8.2 |
| Percentiles (yr) | 10th | 0.1 | 1.1 | <0.1 | 0.3 | <0.1 | <0.1 | <0.1 | 0.9 | 1.0 | 1.1 | -1.0 | -0.2 | 0.0 | -0.8 | -0.1 |
| | 25th | 0.4 | 3.0 | 0.2 | 1.3 | 0.2 | 0.3 | 0.2 | 2.8 | 1.1 | 2.9 | -2.6 | -1.1 | -0.1 | -2.6 | -1.8 |
| | 50th (median) | 1.0 | 7.2 | 1.0 | 5.0 | 1.1 | 3.6 | 1.3 | 7.3 | 1.5 | 7 | -6.2 | -4.0 | -2.5 | -6.0 | -5.5 |
| | 75th | 1.9 | 14.4 | 3.2 | 13.1 | 2.7 | 13.8 | 3.1 | 15.0 | 2.2 | 13.9 | -12.5 | -9.9 | -11.1 | -11.9 | -11.7 |
| | 90th | 3.2 | 26.3 | 6.8 | 25.4 | 4.8 | 27.3 | 5.6 | 25.6 | 3.0 | 23.1 | -23.1 | -18.6 | -22.5 | -20.0 | -20.1 |
| Water fractions (%) | F(T<3 m)* | 16 | 2 | 28 | 10 | 26 | 25 | 25 | 3 | 0 | 2 | 14 | 18 | 1 | 22 | -2 |
| | F(T<6 m) | 30 | 5 | 38 | 14 | 34 | 34 | 32 | 6 | 0 | 5 | 25 | 24 | 0 | 26 | -5 |
| | F(T<1 yr) | 51 | 9 | 50 | 21 | 47 | 40 | 44 | 10 | 13 | 9 | 42 | 29 | 7 | 34 | 4 |
| | F(T<3 yr) | 88 | 25 | 74 | 39 | 78 | 48 | 74 | 26 | 90 | 26 | 63 | 35 | 30 | 48 | 64 |
| | F(T<5 yr) | 97 | 38 | 85 | 50 | 91 | 55 | 88 | 38 | 99 | 39 | 59 | 35 | 36 | 50 | 60 |
| | F(T<10 yr) | 100 | 62 | 95 | 68 | 99 | 68 | 98 | 60 | 100 | 63 | 38 | 27 | 31 | 38 | 37 |
| | F(T<20 yr) | 100 | 85 | 100 | 85 | 100 | 84 | 100 | 84 | 100 | 86 | 15 | 15 | 16 | 16 | 14 |

[Figure]

**Figure FR3.** Stream flow TTDs derived from the 10 model scenarios with the different associated calibration strategies based on different CO models. The TTDs represent the best fits of the respective time-invariant TTD. Green shades represent the TTDs inferred from $\delta^{18}O$ based on different CO models (from lighter to darker for scenarios 3, 5, X1, X3 and X5) in (a) and (b); the purple shades represent TTDs inferred from $^3H$ based on different CO models (from lighter to darker for scenario 4, 6, X2, X4 and X6); the black dots in (b) indicate the mean transit time for each model scenario.

In addition, and as requested by the reviewer, we have also included a "pure" SAS implementation (scenarios X7-9) with one compartment as described in Benettin et al. (2017), using observed Q to account for storage variations (as opposed to modelled Q in the IM-SAS implementations in scenarios 7-12) and one power-law shaped SAS function to route tracers through the system. Also, the results from this model implementation supports our original interpretation: the SAS model, similar to all other IM-SAS implementations (scenarios 7-12), provides similar TTDs for $^{18}O$ and $^3H$. Both estimates are with MTT ~ 11 yrs also broadly consistent with the higher MTTs obtained from the other IM-SAS implementations (see Figure FR4 and Table TR2 here below).

Overall, all results and TTD estimates from additional model implementations are highly consistent with our previous results and considerably strengthen our conclusions to reject the hypothesis that stable isotopes underestimate water ages. We will add all additional model scenarios in the revised manuscript.

**Table TR2.** Metrics of stream flow TTDs derived from the 9 model scenarios with the different associated calibration strategies based on different SAS models, where $C_{\delta^{18}O}$ indicates calibration to $\delta^{18}O$, $C_{^3H}$ calibration to $^3H$, while $C_{\delta^{18}O,Q}$, $C_{^3H,Q}$ and $C_{\delta^{18}O,^3H,Q}$ indicate multi-objective, i.e. simultaneous calibration to combinations of $\delta^{18}O$, $^3H$ and stream flow. The TTD metrics represent the mean and standard deviations of all daily streamflow TTDs during the modelling period 01/10/2001 – 31/12/2016 are given. The mean transit time was estimated by fitting Gamma distributions to the volume-weighted mean TTDs of each individual scenario. The water fractions are shown as the fractions of below a specific age T. The columns with absolute difference Δ illustrate the differences in TTDs from the same models calibrated to $\delta^{18}O$ and $^3H$, respectively. The subscripts indicate the scenarios that are compared (e.g., $\Delta_{7,8}$ compares scenarios 7 and 8). *Note that the fraction of water younger than 3 months is comparable to the fraction of young water as suggested by Kirchner (2016).

| Scenario | | 7 | 8 | 9 | 10 | 11 | 12 | X7 | X8 | X9 | $\Delta_{7,8}$ | $\Delta_{10,11}$ | $\Delta_{X7,X8}$ |
|---|---|---|---|---|---|---|---|---|---|---|---|---|---|
| Model | | IM-SAS-L | | | IM-SAS-D | | | P-SAS | | | Absolute difference | | |
| Calibration strategy → TTD metrics ↓ | | $C_{\delta^{18}O,Q}$ | $C_{^3H,Q}$ | $C_{\delta^{18}O,^3H,Q}$ | $C_{\delta^{18}O,Q}$ | $C_{^3H,Q}$ | $C_{\delta^{18}O,^3H,Q}$ | $C_{\delta^{18}O}$ | $C_{^3H}$ | $C_{\delta^{18}O,^3H}$ | $\Delta TT_{\delta^{18}O,^3H}$ $\Delta F(T<x)_{\delta^{18}O,^3H}$ | | |
| | Mean (yr) | 17.4 | 11.9 | 11.2 | 15.6 | 13.2 | 12.8 | 11.4 | 11.0 | 11.0 | 5.5 | 2.4 | 0.4 |
| Percentiles (yr) | 10th | 0.5±0.7 | 0.5±0.8 | 0.4±0.6 | 0.3±0.5 | 0.3±0.5 | 0.3±0.4 | 0.04±0.03 | 0.02±0.02 | 0.02±0.02 | 0.0 | 0.0 | 0.02 |
| | 25th | 2.1±2.1 | 1.9±2.1 | 1.5±1.8 | 2.1±1.7 | 1.5±1.7 | 1.4±1.5 | 0.4±0.1 | 0.2±0.1 | 0.2±0.1 | 0.2 | 0.6 | 0.2 |
| | 50th (median) | 9.0±3.3 | 6.5±4.8 | 5.7±4.3 | 8.6±2.6 | 6.7±3.7 | 6.6±3.5 | 3.2±0.2 | 2.4±0.2 | 2.5±0.2 | 2.5 | 1.9 | 0.7 |
| | 75th | 22.2±3.3 | 17.6±6.5 | 16.3±6.2 | 20.8±2.8 | 18.8±4.6 | 17.8±4.2 | 13.7±0.3 | 12.5±0.4 | 12.5±0.3 | 4.6 | 2.0 | 1.2 |
| | 90th | 31.3±4.3 | 29.2±5.0 | 28.6±5.1 | 31.1±4.2 | 30.4±4.3 | 29.9±4.2 | 33.4±0.4 | 33.4±0.4 | 32.7±0.2 | 2.1 | 0.7 | 0.0 |
| Water fractions (%) | F(T<3 m)* | 18±12 | 23±19 | 21±15 | 16±10 | 22±13 | 23±15 | 22±3 | 26±3 | 26±2 | -5 | -6 | -5 |
| | F(T<6 m) | 21±13 | 29±22 | 30±19 | 20±11 | 27±16 | 27±16 | 27±2 | 32±2 | 32±2 | -8 | -7 | -5 |
| | F(T<1 yr) | 24±13 | 32±22 | 35±21 | 22±11 | 30±16 | 29±15 | 34±2 | 39±2 | 39±1 | -8 | -8 | -5 |
| | F(T<3 yr) | 31±11 | 39±20 | 42±19 | 30±10 | 37±14 | 37±14 | 49±1 | 53±1 | 52±1 | -8 | -7 | -4 |
| | F(T<5 yr) | 38±10 | 46±18 | 49±17 | 38±9 | 44±13 | 44±12 | 57±1 | 60±1 | 60±1 | -8 | -6 | -3 |
| | F(T<10 yr) | 52±8 | 59±13 | 62±12 | 53±7 | 58±10 | 58±9 | 69±1 | 71±1 | 71±1 | -7 | -5 | -2 |
| | F(T<20 yr) | 71±5 | 77±7 | 79±7 | 74±4 | 76±5 | 77±5 | 82±0 | 83±0 | 83±0 | -6 | -2 | -1 |

[Figure]

**Figure FR4.** Stream flow TTDs derived from the 9 model scenarios with the different associated calibration strategies based on different SAS models (i.e., scenarios7-9 based on model IM-SAS-L, scenarios 10-12 based on model IM-SAS-D, scenariosX7-X9 based on model P-SAS which is same as that described in Benettin et al. (2017)). The TTDs represent the volume weighted average daily TTDs during the modelling period 01/10/2001 – 31/12/2016 are given. Green shades represent the TTDs inferred from $\delta^{18}O$ based on different SAS models (from lighter to darker for scenario 7,10, X7) in (a) and (b); the purple shades represent TTDs inferred from $^3H$ based on different models (from lighter to darker for scenario 8, 11, X8), the brown lines represent TTDs inferred from combined $\delta^{18}O$ and $^3H$ based on different models (brown shades from lighter to darker for scenario 9, 12, X9); the black dots in (b) indicate the mean transit time for each model scenario. Note that the mean transit time was estimated by fitting Gamma distributions to the volume-weighted mean TTDs of each individual scenario.

**(4) Reviewer Comment:**

*Thirdly, the fact that spatial aggregation introduces bias in CO model-based MTTs, as stated also by the authors, raises the question to what extent comparison of MTT estimates is meaningful. I understand that the authors would like to test the validity of stable water isotopes in TT modelling particularly of older water ages, and that MTT has been a metric commonly reported for CO models. Nonetheless, according to Kirchner (2016 – reference already in manuscript), sine-wave fitting to seasonal isotope data does give robust estimates of the young water fraction Fyw. Hence, it might be more meaningful to compare Fyw estimates by the different TT model approaches, or, even better, to add this as further TT metric in the comparison.*

**Reply:**

We agree, that MTT estimates from stable isotopes may be less robust than previously assumed *if* they are estimated using CO-type of models and *if* there is a large contrast in MTTs from sub-parts of the system (which we do not know in reality), as demonstrated by Kirchner (2016). This, however, can at this point not (yet) be generalized as it does not imply that MTT estimates obtained from different model approaches and/or systems with little internal contrast in MTTs suffer similar uncertainties.

But we also completely agree with the reviewer that the exclusive comparison of MTT has the potential to conceal interesting pattern. In that sense there seems to be a misunderstanding: our analysis was never limited to MTTs. Instead, throughout the experiment and the reporting of the results, we always analyse the *full range of TTDs*, i.e. percentiles and fractions of water of different ages. This can be seen in Table 5, as well as Figures 7 – 10 in the original manuscript but also in Figures FR3-4 and Tables TR1-2 here above. As water ages throughout all percentiles show *similar pattern* between the individual scenarios, we used the MTT for communicative purposes in the text (note that the use of any other percentile would have resulted in equivalent descriptions) as this has traditionally been the most commonly used metric. For the purpose of our analysis we believe that the emphasis on MTT in the text instead of using multiple metrics improves the readability of the manuscript. In addition, we think that MTT is more suitable here than the fraction of young water, because the core of the analysis is older water instead of young water. In any case, the young water fractions $F_{yw}$ are of course also part of the analysis in the original manuscript (Table 5, Figures 7 – 10) but also here above (Tables TR1-2, Figures FR3-4). Please note that we used a different symbol to represent it – F(T<3m) (see p.17, l.536) – to remain consistent with the notation of other metrics throughout the manuscript. We will clarify this in the text.

**(5) Reviewer Comment:**

*Finally, I would highly appreciate if the authors could increase traceability of their results and provide the underlying tracer data as well as model codes. Traceability is one of the main criteria for HESS nowadays and given that the authors address such a fundamental claim in tracer hydrology and TT modelling, I find it necessary for the entire TT community to benefit from this study not only via the paper, but also in terms of data and code accessibility.*

**Reply:**

We agree, and we will upload the model code to an open access repository. Most tracer data are available via open access databases as explicitly highlighted in text and the Data availability section. The water stable isotopes in stream samples will be available soon, together with other stream data from Germany, as those data are currently prepared for publication in a data paper. Still, the data from the Neckar can be shared upon request.

*Minor Comments*

*(6) Reviewer Comment:*

*Lines 35—37: if this refers to the findings by Kirchner (2016), one could be more precise by specifying that the MTT (as commonly reported metric) derived from CO models is affected by spatial aggregation errors.*

**Reply:**

Agreed. We will adjust that in the revised manuscript.

*(7) Reviewer Comment:*

*Line 59: in what sense is there more coherence?*

**Reply:**

There is more coherence in the sense that tracer circulation is explicitly linked to and described by the movement of water (i.e. storage and release), which is the actual agent of physical transport in terrestrial hydrological systems.

*(8) Reviewer Comment:*

*Line 70: does Cl- have a clear seasonal cycle? I assume both weathering and anthropogenic effects (e.g., application of road salt) govern its concentrations. Another possible distinction would be radioactive vs. conservative tracers.*

**Reply:**

The chloride ion has a pronounced seasonal cycle, in particular in coastal and maritime influenced climates. It has been successfully applied as age tracer in many previous studies (e.g. Kirchner et al., 2001, 2010; Page et al., 2007; Shaw et al., 2008; Hrachowitz et al., 2009; Soulsby et al., 2010; McMillan et al., 2012; Benettin et al., 2015; Harman, 2015; Wilusz et al., 2017; Cain et al., 2019; Kaandorp et al., 2021; Meira Neto et al., 2022). Anthropogenic effects, such as road gritting, can indeed influence the chloride concentrations. That is why the above studies are limited to catchments with minor human influence.

*(9) Reviewer Comment:*

*Lines 80—98: the focus on the amplitude ratio for the "traditional" TT approaches is fine for simple one-compartment gamma (and thus also exponential) models, but is this also relevant for multiple-compartment CO models and other pre-defined TT shapes such as the dispersion model? This suggests that CO models are exclusively based on the amplitude ratio and shift in seasonal isotope ratios.*

**Reply:**

We are not entirely sure what the reviewer wants to express here. The concept of seasonal tracers as means to estimate stream water ages is rooted in the attenuation of seasonal tracer precipitation amplitudes in the stream water. This is independent of the model application. _Any_ model that aims to represent the movement of such a seasonal tracer through a catchment will have to reproduce these observed attenuation between precipitation stream tracer amplitudes, i.e. the amplitude ratio.

*(10) Reviewer Comment:*

*Lines 84—85: "practically" and "feasibly" twice?*

**Reply:**

Indeed. We will correct that.

*(11) Reviewer Comment:*

*Lines 97: to what extent could a spatial aggregation bias also affect spatially lumped (one-compartment) SAS models?*

**Reply:**

This is unknown and to some extent also investigated here, as explicitly mentioned in the original manuscript (e.g. p.5, l.147ff; p.21, l.636ff; p.22, l.698ff).

*(12) Reviewer Comment:*

*Lines 197: you used the CORINE dataset from 2018. To what extent has land use remained stable since 2001?*

**Reply:**

There was no significant change between the here defined land use classes over the 2001-2018 period, as shown in Table TR3 below.

**Table TR3:** Landuse in the Neckar basin between 1990 and 2018 based on CORINE landcover data.

| Landcover percentage | 1990 | 2000 | 2006 | 2012 | 2018 |
|---|---|---|---|---|---|
| Forest (%) | 35 | 35 | 35 | 36 | 36 |
| Grass/Crop (%) | 53 | 53 | 52 | 50 | 50 |
| Urban (%) | 11 | 12 | 13 | 14 | 14 |
| Water (%) | 1 | ~0 | ~0 | ~0 | ~0 |

**(13) Reviewer Comment:**

L*ine 374: we do not necessarily see passive storage volumes in the most recent SAS model studies.*

**Reply:**

This seems to be a misunderstanding. Indeed, studies based on the "pure" SAS approach that do not model Q, typically define a mixing/sampling storage $S_{tot}$, although the symbols and terminology vary between individual papers (e.g. Benettin et al., 2017). This $S_{tot}$ represents the total storage available for mixing/sampling in a component and is thereby fully equivalent with our $S_{S,tot}$. The difference is that we have to distinguish a hydraulically active part $S_S$ of that storage that represents the hydraulic head above the river bed to generate Q in our model as visualized in e.g. Zuber (1986, Figure 1 – "dynamic" and "minimum" volume) or Hrachowitz et al. (2016; Figure 2), so that $S_{S,tot}=S_S+S_{S,p}$. As "pure" SAS models do not generate Q they also do not need this distinction. Besides that, two definitions of storage are completely identical.

**(14) Reviewer Comment:**

*Lines 398—414: I am wondering to what extent we can trust the spatially distributed implementation, given that there is only one calibration gauge at the outlet of the entire catchment. This also relates to my general comment about the considerable size and few data for the study basin.*

**Reply:**

This is indeed an important comment. To further test the IM-SAS implementations for their ability to reflect the spatial differences in the study basin, we have now evaluated the models' ability to reproduce observed stream flow in several sub-catchments within the Neckar river basin. As described in detail in reply to Comment (2) above and as can be seen in Figure FR2, the results suggest that the model provides a rather robust representation of the hydrological response and its spatial variability throughout the Neckar basin. We will add this analysis to the revised version of the manuscript.

**(15) Reviewer Comment:**

*Line 411: could you specify what the distributed moisture accounting approach is?*

**Reply:**

This type of model implementation, elsewhere also referred to as "semi-lumped" as in detail described by Ajami et al. (2004), runs a model with spatially distributed forcing data but using the same model parameters. For example, here, each precipitation zone receives different precipitation, but the model parameters are the same in all four precipitation zones. This approach has in past been shown to be very effective for improving the representation of spatially variable response dynamics while limiting the amount of necessary model parameters (e.g. Fenicia et al., 2008; Euser et al., 2015).

*(16) Reviewer Comment:*

*Lines 420—421: why have the authors not applied a multi-objective calibration to the CO models?*

**Reply:**

We are not sure what the reviewer intends to express here. The CO models in our study exclusively model the tracer circulation in the basin. They generate only one single output variable, i.e. the tracer concentration in the stream. We therefore cannot perform the same multi-objective calibration as for the IM-SAS models that besides tracer concentrations also reproduce streamflow Q. If the reviewer had a simultaneous calibration of $^{18}O$ and $^{3}H$ in mind, we would like to emphasize that the objective of this paper is to test if the _exclusive_ use of $^{18}O$ underestimates water ages. A simultaneous calibration to both tracers in CO models will not add any additional information to answer this question. Please also note that the simultaneous calibration to $^{18}O$ and $^{3}H$ in the IM-SAS models was only done to test if/how it affects parameters that control water fluxes in the model. Major differences in model parameters between the different calibration approaches would have been an indication for differences of how the individual models route water and tracers through the system and thus a source of potential uncertainty in the interpretation.

*(17) Reviewer Comment:*

*Line 424: this is interesting but I think, as stated in my general comments, that TTs should be obtained from a SAS model with storage, input and output fluxes defined a priori (as if they were "real" data), rather than computing TTs from simultaneous calibration against flow and tracers. I think that this would be a more straightforward methodology given the scope of TT modelling and tracers. As presented here, we do not know to what extent simulated TTs are affected by equifinality in the hydrological model parameters.*

**Reply:**

Please see above: as replied to Comment (3) we have now added such a model implementation (scenario X7-8; Figure FR4 and Table TR2). The results lead to the same conclusions as the IM-SAS model implementations: $^{18}O$ and $^{3}H$ lead to similar TTDs, and there is no indication for $^{18}O$ truncating water ages. This further strengthens our original conclusions. We will add this model implementation to the revised manuscript.

***(18) Reviewer Comment:***

*Lines 553—555: not a complete sentence*

**Reply:**

We well correct this.

***(19) Reviewer Comment:***

*Line 571: not only, but also…?*

**Reply:**

We will correct this.

***(20) Reviewer Comment:***

*Lines 577—578: I think you could easily implement the multi-objective calibration for CO models as well.*

**Reply:**

Indeed. It would be easily to implement that, but as explained in response to Comment (16) it does not add any additional information to test the research hypothesis.

***(21) Reviewer Comment:***

*Lines 619—620: so here one could at least test how time-variant/seasonal CO models perform*

**Reply:**

This would indeed be an interesting analysis. However, it is outside the scope of this study as explained in response to Comment (3) above.

***(22) Reviewer Comment:***

*Lines 642—644: could this not be an indication of the fact that there are too many degrees of freedom and the model succeeds to fit the tracer data, regardless of whether it is spatially lumped or semi-distributed?*

**Reply:**

As shown in Figure FR1 above, there is little indication of model overfitting that could results from "too many degrees of freedom". One explanation of the observed similarity between the lumped and distributed models could be that much of the climatic and topographic heterogeneity within the catchment is filtered out in the response (see also reply to Comment (2) above), so that a lumped representation may be sufficient to pick up the major features of the hydrological response in the study basin.

**(23) Reviewer Comment:**

*Lines 656—657: see, e.g., Nguyen et al. (2022) who found substantial differences in SAS-based transport models between spatially lumped and semi-distributed setup.*

**Reply:**

We will refer to that study as an example of a setting where spatial differences seem to be more relevant.

**References:**

Ajami, N. K., Gupta, H., Wagener, T., & Sorooshian, S. (2004). Calibration of a semi-distributed hydrologic model for streamflow estimation along a river system. Journal of hydrology, 298(1-4), 112-135.

Aubert, A. H., Kirchner, J. W., Gascuel-Odoux, C., Faucheux, M., Gruau, G., & Mérot, P. (2014). Fractal water quality fluctuations spanning the periodic table in an intensively farmed watershed. Environmental Science & Technology, 48(2), 930-937.

Benettin, P., Kirchner, J. W., Rinaldo, A., & Botter, G. (2015). Modeling chloride transport using travel time distributions at Plynlimon, Wales. Water Resources Research, 51(5), 3259-3276.

Benettin, P., Soulsby, C., Birkel, C., Tetzlaff, D., Botter, G., & Rinaldo, A. (2017). Using SAS functions and high‐resolution isotope data to unravel travel time distributions in headwater catchments. Water Resources Research, 53(3), 1864-1878.

Cain, M. R., Ward, A. S., & Hrachowitz, M. (2019). Ecohydrologic separation alters interpreted hydrologic stores and fluxes in a headwater mountain catchment. Hydrological Processes, 33(20), 2658-2675.

Euser, T., Hrachowitz, M., Winsemius, H. C., & Savenije, H. H. (2015). The effect of forcing and landscape distribution on performance and consistency of model structures. Hydrological Processes, 29(17), 3727-3743.

Fenicia, F., Savenije, H. H., Matgen, P., & Pfister, L. (2008). Understanding catchment behavior through stepwise model concept improvement. Water Resources Research, 44(1).

Godsey, S. E., Aas, W., Clair, T. A., De Wit, H. A., Fernandez, I. J., Kahl, J. S., ... & Kirchner, J. W. (2010). Generality of fractal 1/f scaling in catchment tracer time series, and its implications for catchment travel time distributions. Hydrological Processes, 24(12), 1660-1671.

Hrachowitz, M., Soulsby, C., Tetzlaff, D., Dawson, J. J. C., & Malcolm, I. A. (2009). Regionalization of transit time estimates in montane catchments by integrating landscape controls. Water Resources Research, 45(5).

Hrachowitz, M., Savenije, H., Bogaard, T. A., Tetzlaff, D., & Soulsby, C. (2013). What can flux tracking teach us about water age distribution patterns and their temporal dynamics?. Hydrology and Earth System Sciences, 17(2), 533-564.

Hrachowitz, M., Fovet, O., Ruiz, L., & Savenije, H. H. (2015). Transit time distributions, legacy contamination and variability in biogeochemical 1/fα scaling: how are hydrological response dynamics linked to water quality at the catchment scale?. Hydrological Processes, 29(25), 5241-5256.

Kaandorp, V. P., Broers, H. P., Van Der Velde, Y., Rozemeijer, J., & De Louw, P. G. (2021). Time lags of nitrate, chloride, and tritium in streams assessed by dynamic groundwater flow tracking in a lowland landscape. Hydrology and Earth System Sciences, 25(6), 3691-3711.

Kirchner, J. W., Feng, X., & Neal, C. (2000). Fractal stream chemistry and its implications for contaminant transport in catchments. Nature, 403(6769), 524-527.

Kirchner, J. W., Tetzlaff, D., & Soulsby, C. (2010). Comparing chloride and water isotopes as hydrological tracers in two Scottish catchments. Hydrological Processes, 24(12), 1631-1645.

Kirchner, J. W., & Neal, C. (2013). Universal fractal scaling in stream chemistry and its implications for solute transport and water quality trend detection. Proceedings of the National Academy of Sciences, 110(30), 12213-12218.

McMillan, H., Tetzlaff, D., Clark, M., & Soulsby, C. (2012). Do time‐variable tracers aid the evaluation of hydrological model structure? A multimodel approach. Water Resources Research, 48(5).

Meira Neto, A. A., Kim, M., & Troch, P. A. (2022). Physical Interpretation of Time‐Varying StorAge Selection Functions in a Bench‐Scale Hillslope Experiment via Geophysical Imaging of Ages of Water. Water Resources Research, 58(4), e2021WR030950.

Page, T., Beven, K. J., Freer, J., & Neal, C. (2007). Modelling the chloride signal at Plynlimon, Wales, using a modified dynamic TOPMODEL incorporating conservative chemical mixing (with uncertainty). Hydrological Processes: An International Journal, 21(3), 292-307.

Shaw, S. B., Harpold, A. A., Taylor, J. C., & Walter, M. T. (2008). Investigating a high resolution, stream chloride time series from the Biscuit Brook catchment, Catskills, NY. Journal of Hydrology, 348(3-4), 245-256.

Soulsby, C., Tetzlaff, D., & Hrachowitz, M. (2010). Are transit times useful process‐based tools for flow prediction and classification in ungauged basins in montane regions?. Hydrological Processes, 24(12), 1685-1696.

Soulsby, C., Birkel, C., Geris, J., Dick, J., Tunaley, C., & Tetzlaff, D. (2015). Stream water age distributions controlled by storage dynamics and nonlinear hydrologic connectivity: Modeling with high‐resolution isotope data. Water Resources Research, 51(9), 7759-7776.

Wilusz, D. C., Harman, C. J., & Ball, W. P. (2017). Sensitivity of catchment transit times to rainfall variability under present and future climates. Water Resources Research, 53(12), 10231-10256.

---

## Author Comment (AC3)

We are glad to see that our work raises interest, and we would like to thank Julien Farlin (JF) for sharing his perspective and thoughts on our manuscript in a detailed list of comments, which help to further strengthen the analysis and to sharpen the key message. We, however, also note that JF's main concerns are based on several misunderstandings, unfortunate misrepresentations or misinterpretations of our work, and statements that are factually incorrect or not relevant to the study. To reply, we will here below provide clarifications to the major points made by JF.

**(1) The research hypothesis cannot and should not be tested**

JF begins his comments with arguing that the finding of Stewart et al. (2010; for brevity hereafter ST2010) of underestimating water ages with stable isotopes compared to tritium needs to be understood as a mere warning and that it cannot be generalized. While we will not speculate on how ST2010 *meant* their paper, we keep to what is actually *written* in that paper. In fact, throughout ST2010, the inability of stable isotopes to detect older water is described and further contrasted with tritium in statements that are general, rather assertive and effectively without any qualifications that may suggest that this is not a general phenomenon.

For example:

"[…] water resident in the catchment for longer than about 4 years is not expected to show detectible variation in $^{18}O$ […] and therefore is effectively invisible to the method."

"[…] use of $^{3}H$ for estimation of residence time in watersheds, in order to reveal the real age and origin of streamwater." (as opposed to the perceived "not" real age when using stable isotopes)

"The current largely sole focus on streamwater residence time deduced from $^{18}O$ studies has truncated our view of streamwater residence time […]"

Together with the earlier results of DeWalle et al. (1997), who suggested that stable isotopes can only detect water with mean transit times MTT < ~ 4 years, the notion that stable isotopes therefore generally cannot see older water has since then become somewhat of an informal consensus in hydrology. This is illustrated by explicit statements to this effect in many papers over the last decades (e.g. McDonnell and McGuire, 2006; Seeger and Weiler, 2014; Stewart et al., 2017, 2021) but also by the consistently low water ages estimated when exclusively using stable isotopes in time-invariant CO models reported by many studies (e.g. McDonnell and McGuire et al., 2006; Hrachowitz et al., 2009; Godsey et al., 2010; ST2010) as compared to higher water ages obtained in other studies that used tritium as tracer (ST2010 and references therein). Thus, irrespective of how ST2010 *meant* their study, we believe it is important to further scrutinize the hypothesis that stable isotopes cannot detect older water with different methods and from a different perspective than previous studies did (i.e. ST2010; Rodriguez et al., 2021): a rejection of that hypothesis would have the obvious benefit of opening many more possibilities for the meaningful use of the more widely available stable isotope data (in contrast to tritium) as tracers in many systems that may be characterized by the presence of older water.

JF then continues to argue that it is a "fundamental mistake" to test (and eventually reject) the above hypothesis. As a first reason for that JF suggests that for a specific catchment it is simpler to use tritium and stable isotopes together to estimate water ages. It is difficult to follow the logic of the argument as

to why it would speak against testing the above hypothesis. Apart from that and perhaps more importantly, the suggested combined use of both tracers is surely valuable but only possible in theory. In reality, there is only a very small number of locations world-wide where data for both tracers is actually available as also highlighted by JF ("But as I have found out recently, such datasets are extremely rare") and were tritium is not influenced by controlled tritium release from nuclear power plants. As a second reason JF puts forward the need to analyse many different catchments. This reason seems to suggest that we intend to show that the use of stable isotopes and tritium generally, if not always, results in similar age estimates. Or in other words, it suggests that we test the hypothesis that "*[…] $^{18}O$ as tracer generally and systematically can detect tails in water age distributions and that there is no truncation that would leads to systematically younger water age estimates than the use of $^{3}H$.*" This is a hypothesis rooted in logical positivism ("verifiability"). As such it is impossible to test in real world conditions with available data/knowledge as indeed all (or at least many) catchments would need to be analysed if the fallacy of defective induction (i.e. faulty generalization) would want to be avoided.

As such the above is a fundamental misrepresentation of our work. Instead and as explicitly formulated in our research hypothesis we test the hypothesis that "*[…] $^{18}O$ as tracer generally and systematically cannot detect tails in water age distributions and that this truncation leads to systematically younger water age estimates than the use of $^{3}H$.*" By doing this we adopt the concept of falsifiability, which is, since at least Popper (1934), a standard technique of scientific method: we formulate a hypothesis that is actually falsifiable by an empirical test based on the available data. Falsifiability is based on the *modus tollens*: if a hypothesis A is correct then what needs to be observed is B = true. However, if what is observed in reality B = false, A has to be rejected. This also entails that it is sufficient to find one single instance of B = false (here: one case in which the exclusive respective use of $^{18}O$ and $^{3}H$ give similar water ages) to reject the hypothesis A and therefore avoids the need to test all other instances (here: catchments). JF therefore criticizes our analysis for something it was never intended for, which is an example for the use of a classic Straw Man argument.

Finally, JF argues that in some catchments water age estimates inferred from $^{18}O$ and $^{3}H$ may be more similar than in others. We completely agree, and we have never contested that. Instead, we have even dedicated a substantial part of Section 6 in the original manuscript to discuss implications of that in detail (P.20, l.621ff).

**(2) The data set is unsuitable**

JF goes on to point out that the study river basin is large and perhaps too large for a meaningful analysis based on the available data. We agree that it remains a defining challenge in hydrology to fully account for heterogeneities in larger systems. Notwithstanding the above and to limit adverse effects of a coarser data resolution, we here invested considerable effort into spatial adjustments of hydro-meteorological input data as well as tracer data, according to the best available information. While the major spatial differences in precipitation input were characterized based on a network of 16 precipitation gauges in the region and robustly clustered into four individual precipitation zones, both, $^{18}O$ and $^{3}H$ composition in precipitation only exhibit limited spatial differences across the greater region as shown by Stumpp et al. (2014) for $^{18}O$ and Schmidt et al. (2020) for $^{3}H$. In spite of that, we nevertheless further accounted for their respective spatial heterogeneities following the robust method developed by Allen et al. (2018, 2020).

It is also noted by JF that our model was only constrained by observations at the main outlet of the Neckar study basin and not by further observations within the basin. This is indeed an excellent suggestion we have not considered in the original analysis. We have therefore now heeded the advice and decided to confront the model with additional observations to further test its ability to meaningfully represent spatial differences in the response. To do so, we have now also confronted with streamflow observations in three sub-catchments (C1: Kirchentellinsfurt, C2: Calw, and C3: Kocherstetten) within the Neckar basin, whereby each one of them largely represents the response from one of the precipitation zones defined for this study (Figure FR1 here below).

[Figure]

**Figure FR1**. (a) Sub-catchments C1 – C3 within the Neckar basin used to evaluate the model performance, (b) model performance in the Neckar basin vs. sub-catchment C1, (c) Neckar vs. C2 and (d) Neckar vs. C3, based on Scenario 10. The dots indicate all Pareto-optimal solutions in the multi-objective model performance space. The shades from dark to light indicate the overall model performance based on the Euclidean Distance $D_E$, with the darker solutions representing the overall better solutions (i.e. smaller $D_E$)

It can be seen, that the model calibrated on stream flow of the entire Neckar basin can reproduce stream flow at the 3 additional gauges within the basin similarly well, with C2 and C3 even showing a better performance to reproduce streamflow with many of the solutions than the calibrated Neckar stream flow. These results suggest that the model does indeed pick up the major differences in responses due to hydro-climatic heterogeneities throughout the Neckar basin. Together with the spatial adjustments of the tracer inputs as described above, this is further evidence that the model and the underlying spatial differences in input provide an adequate representation of the major features of the hydrological response even at the larger scale of the Neckar basin. The resolution of the spatial differences in the hydrological response also controls spatial differences in tracer circulation and the resulting contrasts in TTDs between the precipitation zones as shown in Figure FR2, with MTTs ranging between ~9 yrs and ~21 yrs for the individual precipitation zones. We will add this additional model evaluation to the manuscript to better demonstrate the suitability of the data used in the model in our study.

[Figure]

**Figure FR2.** The contrasts in TTDs between the precipitation zones from three scenarios 10-12 based on the IM-SAS-D model. The TTDs represent the volume weighted average daily TTDs for the modelling period 01/10/2001 – 31/12/2016. The TTDs in (a) and (b) are inferred from scenario10 ($C_{\delta^{18}O,Q}$) and green shades represent the TTDs inferred from four different precipitation zones (i.e. P1,P2,P3,P4); The black dots in (b) indicate the MTT values (14.5 yr, 21.9 yr, 15.3 yr, and 10.2 yr for P1-P4 respectively) from scenario10 ($C_{\delta^{18}O,Q}$). The TTDs in (c) and (d) are inferred from scenario11 ($C^3_{H,Q}$) and purple shades represent the TTDs inferred from four different precipitation zones (i.e. P1,P2,P3,P4) ; The black dots in (d) indicate the MTT values (12.3 yr, 21.4 yr, 13.0 yr, and 7.9 yr for P1-P4 respectively) from scenario11 ($C^3_{H,Q}$). (e) and (f) are inferred from scenario12 ($C_{\delta^{18}O,^3H,Q}$) and brown shades represent the TTDs inferred from four different precipitation zones (i.e. P1,P2,P3,P4); The black dots in (f) indicate the MTT values (12.0 yr, 17.7 yr, 12.6 yr, and 8.6 yr for P1-P4 respectively) from scenario12 ($C_{\delta^{18}O,^3H,Q}$). Note that the long term-annual precipitation in four zones: P2<P3<P1<P4.

**(3) The model is overparameterized and not sufficiently well tested**

In a next point, JF raises the concern that the IM-SAS models implementations in our study are "probably largely overparameterized", without substantiating this claim with further evidence. We firstly want to clarify that the term "overparameterized" is used for models that have more parameters than necessary. In the presence of noise in data (e.g. observational uncertainties) and thus in any application in river basin hydrology, it is *a priori* not possible to objectively determine if a model is overparameterized. However, overparameterization can, but *does not necessarily,* lead to model overfitting, i.e. a situation in which the model cannot reproduce the system response to similar level with previously unseen data than with data it was trained with. Models are typically tested for overfitting based on split-sample tests. The results of these tests indicate *little, if any, overfitting*. This can be seen in Figure FR3 here below for Scenarios 10-12, where the vast majority of all pareto-optimal solutions plot on or close to the 1:1 line. Similarly, the proxy basin split-sample test with the three additional gauges within the Neckar basin further supports this observation (see above Figure FR2). In the original manuscript the results of these split-sample tests are reported and summarized by the calibration and validation performance metrics in Table 4 and Supplementary Table S5.

[Figure]

[Figure]

[Figure]

**Figure FR3**. Model split-sample tests using model performance metrics (Mean Squared Error MSE) in calibration (2000-2009; cal.) and validation periods (2010-2016; val.) at the outlet of the Neckar Basin for the (a) Scenario 10, (b) Scenario 11 and (c) Scenario 12. The dots indicate all Pareto-optimal solutions in the multi-objective model performance space. The colour shades from dark to light indicate the associated Euclidean Distances $D_E$, with the darkest shades representing the lowest $D_E$ and thus the best available solution.

JF then further expresses his opinion that the IM-SAS model implementations in our analysis do not sufficiently well reproduce the observations and that the models do not "work properly". As, in particular in hydrology, there is neither a fully objective way nor a general community consensus to judge if a specific model is fit for purpose or not, this issue can be discussed back and forth. In the end it remains a subjective expression of opinion.

Although direct comparisons of different studies are difficult, we nevertheless would at least like to provide some perspective here as to what can reasonably be expected from state-of-the-art hydrological models. In a recent paper, Kratzert et al. (2019) collated and compared the results of several previous modelling studies that had previously implemented multiple other, frequently used standard hydrological models in >500 contrasting catchments across the US. Amongst others, these models included *VIC* (13 parameters; Liang et la., 1996; Newman et al., 2017), *SAC-SMA* (18 parameters; Burnash at al., 1973; Newman et al., 2017) and *mhM* (>50 global parameters, but simultaneously calibrated to multiple basins; Samaniego et al., 2010; Mizukami et al., 2019). These models were *exclusively calibrated to streamflow* over comparable study periods of ~10-15 years. Comparing the performance of our solution that best reproduces streamflow, which is equivalent to $NSE_Q$=0.71 in the calibration and 0.65 in the test periods,

respectively, with the performances of these other standard models, suggests that our model performs *better* than the 75% (*VIC*), 66% (*SAC-SMA*) and 46% (*mhM*) of the implementations of these models (Figure FR4). While this suggests that our IM-SAS-D implementation has certainly room for improvement, it makes it at the same time difficult to credibly argue that the model does not "work properly".

[Figure]

**Figure FR4**. Performance of best available solution to reproduce Q in the model validation period (2010-2016) for IM-SAS-D (red line), compared to the distributions of model validation period performances of the *VIC*, *SAC-SMA* and *mHM* models in >500 catchments in the US (after Kratzert et al., 2019).

It is noted that JF flatly dismisses our IM-SAS model implementations seemingly exclusively based on his claim that the model overestimates and even "systematically exaggerates" peaks (see JF's comment to Figure 5). This is factually incorrect. As shown in Figures 5b-d (red line) in the original manuscript, the model instead tends to underestimate some of highest peaks – a common phenomenon in hydrological models and most frequently related to stream flow data uncertainties under high flow conditions. In contrast, the model overall captures the temporal dynamics and timing of the response (Figure 5b-c), the flow magnitudes under non-peak flow conditions (Figure 5d), as well as the shape of the recession and thus the memory of the system (Figure 5f).

JF similarly argues that our IM-SAS implementations do a poor job in representing the observed tritium signals and, in particular, its seasonal variations. We agree, that these dynamics are not very well represented, as explicitly described in the original manuscript (P.15, l.470). However, and again, it is worth to consider what current state-of-the-art model can be expected to achieve: our modelled tritium dynamics and tritium levels are not worse and for some cases even *clearly better* than the ones in the few existing previous studies (e.g. Koeniger et al., 2005; Duvert et al., 2016; Visser et al., 2019; Rodriguez et al., 2021), and much better than the ones modelled with CO models in the study basin (Table 4, Figure 4 and Supplementary Figures S7,9 in the original manuscript as well as Table TR1 and Figures FR5-FR9 for here below).

**Table TR1.** Performance metrics for $MSE^3_H$ of the model implementations (CO models and IM-SAS models) and the associated calibration strategies for the 2001 – 2009 calibration period (cal.) and the 2010 – 2016 model evaluation period (val.). For brevity only the values for the most balanced solution, i.e., lowest $D_E$ (Eq. 14) for scenarios 8-12 are shown here.

| Scenario | | 4 | 6 | X2 | X4 | X6 | 8 | 9 | 11 | 12 |
|---|---|---|---|---|---|---|---|---|---|---|
| Model | | CO-EM | CO-GM | CO-2EM | CO-3EM | CO-EPM | IM-SAS-L | IM-SAS-L | IM-SAS-D | IM-SAS-D |
| Calibration strategy → Performance metric ↓ | | $C^3_H$ | $C^3_H$ | $C^3_H$ | $C^3_H$ | $C^3_H$ | $C^3_{H,Q}$ | $C_\delta{}^{18}{}_O,^3{}_{H,Q}$ | $C^3_{H,Q}$ | $C_\delta{}^{18}{}_O,^3{}_{H,Q}$ |
| $MSE\ _{3_H}$ | cal. | 5.903 | 5.791 | 5.171 | 5.170 | 5.926 | 2.972 | 2.823 | 2.920 | 2.981 |
| | val. | 5.155 | 4.597 | 3.964 | 4.000 | 5.115 | 2.389 | 2.285 | 2.357 | 2.450 |

[Figure]

**Figure FR5.** Time series of stream $^3H$ reproduced by CO-EM model, i.e., calibration strategy $C^3_H$ (scenario 4), for the model calibration and evaluation periods. (a) Observed $^3H$ signals in precipitation (light blue-purple dots; size of dots indicates associated precipitation volume) and in streamflow (pink dots) as well as the modelled $^3H$ stream signal (light purple dots), (b) zoom-in of observed and modelled $^3H$ signals for the 01/01/2007 – 31/12/2012 period for scenarios 4.

[Figure]

**Figure FR6.** Time series of stream $^3H$ reproduced by CO-GM model, i.e., calibration strategy $C^3_H$ (scenario 6), for the model calibration and evaluation periods. (a) Observed $^3H$ signals in precipitation (light blue-purple dots; size of dots indicates associated precipitation volume) and in streamflow (pink dots) as well as the modelled $^3H$ stream signal (light purple dots), (b) zoom-in of observed and modelled $^3H$ signals for the 01/01/2007 – 31/12/2012 period for scenarios 6.

[Figure]

Figure FR7. Time series of stream $^3$H reproduced by CO-2EM model, i.e., calibration strategy C$^3$H (scenario X2), for the model calibration and evaluation periods. (a) Observed $^3$H signals in precipitation (light blue-purple dots; size of dots indicates associated precipitation volume) and in streamflow (pink dots) as well as the modelled $^3$H stream signal (light purple dots), (b) zoom-in of observed and modelled $^3$H signals for the 01/01/2007 − 31/12/2012 period for scenarios X2.

[Figure]

**Figure FR8.** Time series of stream $^3$H reproduced by CO-3EM model, i.e., calibration strategy C$^3$H (scenario X4), for the model calibration and evaluation periods. (a) Observed $^3$H signals in precipitation (light blue-purple dots; size of dots indicates associated precipitation volume) and in streamflow (pink dots) as well as the modelled $^3$H stream signal (light purple dots), (b) zoom-in of observed and modelled $^3$H signals for the 01/01/2007 − 31/12/2012 period for scenarios X4.

[Figure]

**Figure FR9.** Time series of stream ³H reproduced by CO-EPM model, i.e., calibration strategy C³_H (scenario X6), for the model calibration and evaluation periods. (a) Observed ³H signals in precipitation (light blue-purple dots; size of dots indicates associated precipitation volume) and in streamflow (pink dots) as well as the modelled ³H stream signal (light purple dots), (b) zoom-in of observed and modelled ³H signals for the 01/01/2007 – 31/12/2012 period for scenarios X6.

**(4) Comparison of models**

In a fourth comment, JF states that the comparison between the different model types to estimate water ages is an important aspect of this manuscript. JF further voices his concerns that the way the CO models were implemented makes comparison to IM-SAS models questionable. It is true that in the original manuscript we have kept the implementation of CO models to a minimum. The reason for that was, in contrast to JF's impression, that the comparison between CO and IM-SAS model types is in fact *not* an important aspect of this manuscript. The intention of our work is *not* to show that CO models can or cannot estimate older ages. Perhaps, time-variant implementations can do that very well. That would be excellent news indeed but exploring this was not the objective of our study. Similarly, the combined use of ¹⁸O and ³H in CO models has previously been shown in many studies, including ST2010, to be useful to estimate older ages. Exploring both above issues in some more detail could be interesting studies in themselves but would not contribute additional information that is relevant to test our research hypotheses that is explicitly and clearly formulated as: "*[…] ¹⁸O as tracer generally and systematically cannot detect tails in water age distributions and that this truncation leads to systematically younger water age estimates than the use of ³H.*"

Strictly spoken, our research hypothesis could here have been tested by merely comparing water ages generated from e.g. scenarios 7 and 8 or, alternatively, scenarios 10 and 11 (IM-SAS, constrained by $^{18}$O and $^3$H, respectively). With the intention to provide a wider context we have also added the remaining 10 SW, CO and IM-SAS scenarios. The SW and CO scenarios were *exclusively* included to check if using model approaches *comparable* to those (i.e. lumped, *time-invariant*) which previous work has based its conclusions on (i.e. that the exclusive use of $^{18}$O in these models causes a truncation of TTDs at rather young water ages; DeWalle et al., 1997; ST2010), would lead to equivalent conclusions in the Neckar study basin. To provide a more complete picture, we have now extended the analysis to a full calibration of the two CO models in the original manuscript (i.e. EM, GM) and added three additional, frequently used CO models: two parallel reservoirs (2EM; scenarios X1-2), three parallel reservoirs (3EM; scenarios X3-4) and exponential piston flow (EPM, scenarios X5-6). The TTD estimates from these more complete and additional model implementations are consistent with those in the original analysis: for all tested *lumped, time-invariant* CO models, the TTDs derived from $^{18}$O indicated with MTTs ~ 1-2 yrs significantly younger water than those derived from $^3$H, which suggest MTTs ~10 yrs throughout (see Table TR2 below). This further strengthens our previous results, suggesting that $^{18}$O when used in *lumped, time-invariant* CO models cannot distinguish water older than 4 yrs (DeWalle et al., 1997) and thus underestimates water ages, as suggested by ST2010 (Table TR2 and Figure FR10). The complete results together with the relevant Figures will be added to the revised manuscript.

**Table TR2.** Metrics of stream flow TTDs derived from the 10 model scenarios with the different associated calibration strategies based on different CO models, where $C_\delta{}^{18}O$ indicates calibration to $\delta^{18}O$, $C^3{}_H$ calibration to $^3$H. The TTD metrics represent the best fits of the respective time-invariant TTD. The water fractions are shown as the fractions of below a specific age T. The columns with absolute difference Δ illustrate the differences in TTDs from the same models calibrated to $\delta^{18}O$ and $^3$H, respectively. The subscripts indicate the scenarios that are compared (e.g., $\Delta_{3,4}$ compares scenarios 3 and 4).

| Scenario | | 3 | 4 | 5 | 6 | X1 | X2 | X3 | X4 | X5 | X6 | $\Delta_{3,4}$ | $\Delta_{5,6}$ | $\Delta_{X1,X2}$ | $\Delta_{X3,X4}$ | $\Delta_{X5,X6}$ |
|---|---|---|---|---|---|---|---|---|---|---|---|---|---|---|---|---|
| Model | | CO-EM | | CO-GM | | CO-2EM | | CO-3EM | | CO-EPM | | Absolute difference | | | | |
| Calibration strategy → TTD metrics ↓ | | $C_\delta{}^{18}O$ | $C^3{}_H$ | $C_\delta{}^{18}O$ | $C^3{}_H$ | $C_\delta{}^{18}O$ | $C^3{}_H$ | $C_\delta{}^{18}O$ | $C^3{}_H$ | $C_\delta{}^{18}O$ | $C^3{}_H$ | $\Delta TT_\delta{}^{18}O^{-3}H$ $\Delta F(T<x)_\delta{}^{18}O^{-3}H$ | | | | |
| Percentiles (yr) | Mean (yr) | 1.4 | 10.4 | 2.4 | 9.7 | 1.9 | 9.5 | 2.1 | 9.4 | 1.8 | 10 | -9.0 | -7.3 | -7.6 | -7.3 | -8.2 |
| | 10th | 0.1 | 1.1 | <0.1 | 0.3 | <0.1 | <0.1 | <0.1 | 0.9 | 1.0 | 1.1 | -1.0 | -0.2 | 0.0 | -0.8 | -0.1 |
| | 25th | 0.4 | 3.0 | 0.2 | 1.3 | 0.2 | 0.3 | 0.2 | 2.8 | 1.1 | 2.9 | -2.6 | -1.1 | -0.1 | -2.6 | -1.8 |
| | 50th (median) | 1.0 | 7.2 | 1.0 | 5.0 | 1.1 | 3.6 | 1.3 | 7.3 | 1.5 | 7 | -6.2 | -4.0 | -2.5 | -6.0 | -5.5 |
| | 75th | 1.9 | 14.4 | 3.2 | 13.1 | 2.7 | 13.8 | 3.1 | 15.0 | 2.2 | 13.9 | -12.5 | -9.9 | -11.1 | -11.9 | -11.7 |
| | 90th | 3.2 | 26.3 | 6.8 | 25.4 | 4.8 | 27.3 | 5.6 | 25.6 | 3.0 | 23.1 | -23.1 | -18.6 | -22.5 | -20.0 | -20.1 |
| Water fractions (%) | F(T<3 m)* | 16 | 2 | 28 | 10 | 26 | 25 | 25 | 3 | 0 | 2 | 14 | 18 | 1 | 22 | -2 |
| | F(T<6 m) | 30 | 5 | 38 | 14 | 34 | 34 | 32 | 6 | 0 | 5 | 25 | 24 | 0 | 26 | -5 |
| | F(T<1 yr) | 51 | 9 | 50 | 21 | 47 | 40 | 44 | 10 | 13 | 9 | 42 | 29 | 7 | 34 | 4 |
| | F(T<3 yr) | 88 | 25 | 74 | 39 | 78 | 48 | 74 | 26 | 90 | 26 | 63 | 35 | 30 | 48 | 64 |
| | F(T<5 yr) | 97 | 38 | 85 | 50 | 91 | 55 | 88 | 38 | 99 | 39 | 59 | 35 | 36 | 50 | 60 |
| | F(T<10 yr) | 100 | 62 | 95 | 68 | 99 | 68 | 98 | 60 | 100 | 63 | 38 | 27 | 31 | 38 | 37 |
| | F(T<20 yr) | 100 | 85 | 100 | 85 | 100 | 84 | 100 | 84 | 100 | 86 | 15 | 15 | 16 | 16 | 14 |

[Figure]

**Figure FR10.** Stream flow TTDs derived from the 10 CO model scenarios with the different associated calibration strategies. The TTDs represent the best fits of the respective time-invariant TTD. Green shades represent the TTDs inferred from δ[18]O based on different CO models (from lighter to darker for scenarios 3, 5, X1, X3 and X5) in (a) and (b); the purple shades represent TTDs inferred from [3]H based on different CO models (from lighter to darker for scenario 4, 6, X2, X4 and X6); the black dots in (b) indicate the mean transit time for each model scenario.

Note, even with the additional CO model implementations, what is actually relevant to compare here are models of the *same type* (and same complexity) run with stable isotopes and subsequently with tritium. The relevant comparison is *not* made between models of different types and/or complexities (although semantically it is often not easy to clearly separate the two in the text). In other words, we compare water age estimates obtained from e.g. a CO model with exponential TTD run with [18]O with those obtained from the same model but run with [3]H. In contrast, the comparison of water ages from that CO model with ages estimated from another model, e.g. IM-SAS, has little relevance for testing the research hypothesis. This is also emphasized by the last five columns of Table 5 in the original manuscript and Table TR2 above.

To further clarify, we have estimated water ages based on CO models to check if we would find differences in water ages between [18]O- and [3]H-based model runs in the study basin, using the *same* types of *lumped, time-invariant* models (and thus not accounting for transient conditions) that ST2010 based their argument on. The fact that we found significant differences between these estimates, would, without further analysis, further support the observation of ST2010 that [18]O *generally* truncates water ages.

Our results then further suggest, that, if used in combination with IM-SAS models, our research hypothesis needs to be rejected, as these models produce similar water ages with [18]O and [3]H that are much older than 4 years. Given that the results of ST2010 as well as our own CO scenarios are based on *lumped, time-invariant* CO model implementations, our results eventually also allow the observation that the perceived failure of [18]O to see older water is not a general limitation of that tracer, but rather a consequence of its use in *lumped, time-invariant* CO models in the Neckar basin. Please note that observing this in the results and reporting it in the manuscript is fundamentally different from, as JF expresses it: "*claiming that the SAS is superior by pretending that something is not possible with the lumped parameter models*".

In addition to the above and to further address the concern of "overparameterization", we have also included a "pure" SAS model (scenarios X7-9; 2 calibration parameters) with one compartment as described in Benettin et al. (2017), using observed Q to account for storage variations (as opposed to modelled Q in the IM-SAS implementations in scenarios 7-12) and one power-law shaped SAS function to route tracers through the system. Also, the results from this model implementation strongly support our original interpretation: the SAS model, similar to all other IM-SAS implementations (scenarios 7-12), provides similar TTDs for $^{18}$O and $^3$H. Both estimates are with MTT ~ 11 yrs also broadly consistent with the higher MTTs obtained from the other IM-SAS implementations (see Figure FR11 and Table TR3 here below).

Overall, all results and TTD estimates from additional model implementations are highly consistent with our previous results and considerably strengthen our conclusions to reject the hypothesis that stable isotopes underestimate water ages. We will add all additional model scenarios in the revised manuscript.

**Table TR3.** Metrics of stream flow TTDs derived from the 9 model scenarios with the different associated calibration strategies based on different SAS models, where $C_{\delta^{18}O}$ indicates calibration to $\delta^{18}$O, $C^3_H$ calibration to $^3$H, while $C_{\delta^{18}O,Q}$, $C^3_{H,Q}$ and $C_{\delta^{18}O,^3H,Q}$ indicate multi-objective, i.e. simultaneous calibration to combinations of $\delta^{18}$O, $^3$H and stream flow. The TTD metrics represent the mean and standard deviations of all daily streamflow TTDs during the modelling period 01/10/2001 – 31/12/2016. The mean transit time was estimated by fitting Gamma distributions to the volume-weighted mean TTDs of each individual scenario. The water fractions are shown as the fractions of below a specific age T. The columns with absolute difference Δ illustrate the differences in TTDs from the same models calibrated to δ18O and 3H, respectively. The subscripts indicate the scenarios that are compared (e.g., Δ7,8 compares scenarios 7 and 8). *Note that the fraction of water younger than 3 months is comparable to the fraction of young water as suggested by Kirchner (2016).

| Scenario | 7 | 8 | 9 | 10 | 11 | 12 | X7 | X8 | X9 | $\Delta_{7,8}$ | $\Delta_{10,11}$ | $\Delta_{X7,X8}$ |
|---|---|---|---|---|---|---|---|---|---|---|---|---|
| Model | IM-SAS-L | | | IM-SAS-D | | | P-SAS | | | Absolute difference | | |
| Calibration strategy → TTD metrics ↓ | $C_{\delta^{18}O,Q}$ | $C^3_{H,Q}$ | $C_{\delta^{18}O,^3H,Q}$ | $C_{\delta^{18}O,Q}$ | $C^3_{H,Q}$ | $C_{\delta^{18}O,^3H,Q}$ | $C_{\delta^{18}O}$ | $C^3_H$ | $C_{\delta^{18}O,^3H}$ | $\Delta TT_{\delta^{18}O,^3H}$ $\Delta F(T<x)_{\delta^{18}O,^3H}$ | | |
| Mean (yr) | 17.4 | 11.9 | 11.2 | 15.6 | 13.2 | 12.8 | 11.4 | 11.0 | 11.0 | 5.5 | 2.4 | 0.4 |
| Percentiles (yr) 10th | 0.5±0.7 | 0.5±0.8 | 0.4±0.6 | 0.3±0.5 | 0.3±0.5 | 0.3±0.4 | 0.04±0.03 | 0.02±0.02 | 0.02±0.02 | 0.0 | 0.0 | 0.02 |
| 25th | 2.1±2.1 | 1.9±2.1 | 1.5±1.8 | 2.1±1.7 | 1.5±1.7 | 1.4±1.5 | 0.4±0.1 | 0.2±0.1 | 0.2±0.1 | 0.2 | 0.6 | 0.2 |
| 50th (median) | 9.0±3.3 | 6.5±4.8 | 5.7±4.3 | 8.6±2.6 | 6.7±3.7 | 6.6±3.5 | 3.2±0.2 | 2.4±0.2 | 2.5±0.2 | 2.5 | 1.9 | 0.7 |
| 75th | 22.2±3.3 | 17.6±6.5 | 16.3±6.2 | 20.8±2.8 | 18.8±4.6 | 17.8±4.2 | 13.7±0.3 | 12.5±0.4 | 12.5±0.3 | 4.6 | 2.0 | 1.2 |
| 90th | 31.3±4.3 | 29.2±5.0 | 28.6±5.1 | 31.1±4.2 | 30.4±4.3 | 29.9±4.2 | 33.4±0.4 | 33.4±0.4 | 32.7±0.2 | 2.1 | 0.7 | 0.0 |
| Water fractions (%) F(T<3 m)* | 18±12 | 23±19 | 21±15 | 16±10 | 22±13 | 23±15 | 22±3 | 26±3 | 26±2 | -5 | -6 | -5 |
| F(T<6 m) | 21±13 | 29±22 | 30±19 | 20±11 | 27±16 | 27±16 | 27±2 | 32±2 | 32±2 | -8 | -7 | -5 |
| F(T<1 yr) | 24±13 | 32±22 | 35±21 | 22±11 | 30±16 | 29±15 | 34±2 | 39±2 | 39±1 | -8 | -8 | -5 |
| F(T<3 yr) | 31±11 | 39±20 | 42±19 | 30±10 | 37±14 | 37±14 | 49±1 | 53±1 | 52±1 | -8 | -7 | -4 |
| F(T<5 yr) | 38±10 | 46±18 | 49±17 | 38±9 | 44±13 | 44±12 | 57±1 | 60±1 | 60±1 | -8 | -6 | -3 |
| F(T<10 yr) | 52±8 | 59±13 | 62±12 | 53±7 | 58±10 | 58±9 | 69±1 | 71±1 | 71±1 | -7 | -5 | -2 |
| F(T<20 yr) | 71±5 | 77±7 | 79±7 | 74±4 | 76±5 | 77±5 | 82±0 | 83±0 | 83±0 | -6 | -2 | -1 |

[Figure]

**Figure FR11.** Stream flow TTDs derived from the 9 model scenarios with the different associated calibration strategies based on different SAS models (i.e., scenarios7-9 based on model IM-SAS-L, scenarios 10-12 based on model IM-SAS-D, scenariosX7-X9 based on model P-SAS which is same as that described in Benettin et al. (2017)). The TTDs represent the volume weighted average daily TTDs during the modelling period 01/10/2001 – 31/12/2016. Green shades represent the TTDs inferred from $\delta^{18}O$ based on different SAS models (from lighter to darker for scenario 7,10, X7) in (a) and (b); the purple shades represent TTDs inferred from $^3H$ based on different models (from lighter to darker for scenario 8, 11, X8), the brown lines represent TTDs inferred from combined $\delta^{18}O$ and $^3H$ based on different models (brown shades from lighter to darker for scenario 9, 12, X9); the black dots in (b) indicate the mean transit time for each model scenario.

**Specific Comments**

**(5) Comment:**

*L49: Basically, the problem lies in how to inject a tracer „instantaneously" over the entire watershed, and not so much in the availability of„adequate observation technology", since all one needs to do after the injection is to sample the output for long enough to reach near complete recovery. As a side note, the first catchment scale tracer experiment I know of is that by Rodhe et al. [6].*

Reply:

We agree. This is what was meant.

**(6) Comment:**

*L50: The phrasing is too vague. Transit time distribution is EITHER inferred from input and output measurements ([7], [8]), OR assumed in order to calculate from input and output measurements useful catchment caracteristics (i.e. the mean transit time and the storage volume).*

Reply:

Even if the shape of TTD is assumed, its parameters need to be inferred from observed tracer input-output relationships.

**(7) Comment:**

*L52: Citations for the sine-wave method are missing, for instance, Maloszewski et al. [9].*

Reply:

The citation is given at the end of the line. Should read as Maloszewski et al. (1983).

**(8) Comment:**

*L53: The lumped-parameter models may have been introduced for groundwater environments (see Eriksson [10]), but Piotr Maloszewski and Willibald Stichler in particular have used them early for surface water studies (see [11] and [12] for instance, but there are many more).*

Reply:

Certainly. We are aware of that and do not disagree.

**(9) Comment:**

*L56: The fact that the model representing the TTD must be chosen a priori has nothing to do with the steady-state approximation. These are two different things. A model must be chosen a priori in transient mode as well. And the authors could mention that the choice may be a priori, but is not arbitrary at all, and that model choice has to be guided by the boundary conditions and the sampling scheme. Additionally, SAS models are also based on an a priori choice for the selection functions.*

Reply:

True. A model must be chosen a priori in transient mode as well. However, we believe that the notion that "*[…] the choice […] is not arbitrary at all […]*" is too optimistic for what is done in many studies in reality, where the choice is not well or not at all justified. We agree, that SAS models are also based on *a priori* choice of the sampling distribution, which is explicitly stated in L.61.

*L56: „While this assumption […]“. This sentence is much too vague and inaccurate. Firstly, Zuber [4] has clearly suggested in his paper presenting a transient approach for the lumped parameter models that as long as the total storage accessible to tracer is large compared to the transient storage (what the authors refer to as „the temporal variability in the hydrometeorological drivers"), then the steady-state approximation would yield nearly the same result as the transient fit. This hypothesis was then illustrated for a surface water case study of the Lange Bramke catchment [5], where this turned out to be indeed the case. This will of course depend on the local hydrogeological setting, but should be considered. Secondly, the variability in precipitation input is completely taken into account in lumped-parameter modelling, since the time steps of the input can be defined freely. So if daily data is available, nothing speaks against making calculations at that definition (whether this is such a good idea is another issue altogether). Thirdly, spatial heterogeneities in flow paths can absolutely be modelled using lumped parameter models by coupling them in parallel or in series, as was done routinely by Maloszewski and colleagues (starting with [13]). It is true that these potential heterogeneities are lumped together in a single measured output, but in that regard, the SAS face exactly the same limitation, namely that of extracting information from relating a single input to a single output. Fourthly, what do the authors mean by „misinterpretation" ? Typically, the results of lumped-parameter modelling is a mean transit time of tracer and a storage volume which should be compared to the hydrogeological information available concerning porosity. It is not the model results that are misinterpreted, but rather, model results can be wrong if an inapropriate model has been chosen, for instance.*

Reply:

We agree. Of course, the smaller fraction of transient volumes to the total volume the more similar the transient and steady-state estimates. In its essence this is already encapsulated in the general definition of mean turnover times (e.g. Eriksson, 1958; Bolin and Rodhe, 1973; Nir, 1973).

We disagree on the second point. While variability in precipitation tracer composition is indeed taken into account, volumes are considered only as total precipitation in the vast majority of studies, with only very limited efforts to account for the effects of temporal variability in evaporation, which on average accounts for ~60% of the water balance in the Neckar basin, or snow melt and thus "effective precipitation". The same is true for the representation of lateral spatial variability (i.e. parallel models), which is only done in a small number of studies. A statement to this effect is given in L.55.

Water ages modelled with CO approaches can be and have in the past been frequently misinterpreted as plausible estimates, which in many cases cannot be assumed anymore following the results of ST2010, Kirchner (2016) or Stewart et al. (2017), who demonstrated that these estimates can be subject to major uncertainties, depending on the tracer used and the heterogeneities in the system.

*(11) Comment:*

*L59: Given the constant string of publications, in particular by Maloszewski and colleagues, over thirty years, exploring systematically the possibilities and limitations of such models, I think one cannot seriously argue that they lack a coherent framework.*

Reply:

We do not argue that they "lack a coherent framework". Instead, it is stated that SAS is *more* coherent, as it is fundamentally based on explicitly accounting for storage variations in the water age balance (e.g. Benettin et al., 2022).

***(12) Comment:***

*L61: „without the need". This is phrased as if the SAS approach could do away with a priori model choice. But then, in the next sentence, one learns the exact opposite. The SAS, just like the lumped parameter models, have at their core a series of functions necessary to relate input and output, so in that regard, they are the same, and trying to present the one approach as „freer" from a priori choices as the other is incorrect.*

Reply:

This sentence exactly expresses what it is meant to express. In fact there is no need to *a priori* define a TTD in the SAS approach. The TTDs emerge from the storage and release dynamics. To do so, instead a sampling function is *a priori* defined. This is a neutral description of the differences between the two approaches.

***(13) Comment:***

*L61: „change in water storage are considered". So are they using the transient approach proposed by Zuber*

Reply:

Since both methods represent the physical transport of water and thus of water isotopes, they are of course related and mathematically even almost equivalent, with the difference that Zuber's transient description, based on Niemi (1977) and functionally similar to Nir (1973), does not explicitly *track* the individual water volumes (and thus tracer concentrations) of varying age in an explicit treatment of the "water age balance" (e.g. Benettin et al., 2022) but instead directly applies the convolution operation. Notwithstanding the above, the SAS-function approach is conceptually rather rooted in the – again, mathematically equivalent – development of the hydro-chemical routing in the Birkenes (Lundquist, 1977) and HBV models (Bergström, 1973) that go back to the early 1970s. These models can explicitly *track* water volumes in parcels of varying age and explicitly *sample/mix* the outflow from these storage volumes as illustrated in particular by Figure 1 in Bergström et al. (1985) and similarly formulated in many other studies beginning from that time (e.g. Christophersen and Wright, 1981; Christophersen et al., 1982; Seip et al., 1985; de Groisbois et al., 1988; Hooper et al., 1988; Barnes and Bonell, 1996).

***(14) Comment:***

*L64-65: The explicit tracking is different from unsteady state, and should not be confused with it. By setting a constant storage, SAS can be used in steady state mode*

Reply:

We completely agree. Of course the storage can be set constant. However, doing this would thwart the fundamental idea behind SAS.

**_(15) Comment:_**

_L75: „The second type […]". This needs qualification. Dating can be done in two different ways using tritium. Either one takes advantage of the tritium peak resulting from the atmospheric bomb testing of the 1950s and 1960s, or now, in the post-peak era, from the shift between the mean annual input and output due to decay losses in the subsurface. Please note that other radioactive tracers used for dating such as krypton 85 display a steadily increasing trend since the 1960s, and as such, it is not so much the decay than the rate of increase that is used for dating. The same holds true for non-radioactive tracers such as the chlorofluorocarbons._

Reply:

We agree. However, descriptions of the different ways to use tritium or other tracers are of minor relevance for this study. Still, we will make sure that we are talking specifically about tritium here ("…water age can be estimated with tritium based…"

**_(16) Comment:_**

_L85: The entire paragraph seems a bit out of place in an introduction. Why so many details concerning the upper limit of the sine-wave method?_

Reply:

Because the estimates of DeWalle et al. (1997), based on SW, were the first to claim that $^{18}O$ can only detected ages < ~4 yrs and as SW models are merely simplifications of CO models, the DeWalle et al. (1997) analysis is in a way the basis of ST2010.

**_(17) Comment:_**

_L96: How is that back-of-the-envelope-calculation done ?_

Reply:

It was done as described in the text: using the tracer amplitudes used and reported by DeWalle et al. (1997) and replacing the exponential model with a gamma model ($\alpha=0.5$), as for example described by Kirchner (2016).

**(18) Comment:**

*L97: The sensitivity of the sine-wave methods have nothing to do with potential aggregation biases, these are just two different issues*

Reply:

We agree. Of course these are two different factors. We do not suggest anything else. However, both can lead to uncertainties in TTD estimates.

**(19) Comment:**

*L125: Is three years of measurements for a tracer that varies on an annual basis so bad? This is three replicate. The handful of tritium measurements was enough for dating in the 80s when the decrease over time was still steep.*

Reply:

We are not sure what to make of this comment. Sure, three years of data is a good start. But longer timeseries allow a much more robust estimation of older water ages as otherwise the tails of TTDs are never confronted with data.

**(20) Comment:**

*L127: What do the authors mean by „precluded" ? The exponential model describes a continuous distribution of transit times from zero (for flow lines close to the outlet) to infinity (for flow lines near the watershed divide). How does that preclude longer transit times ? And since in the studies cited the same models have been calibrated for both tracers, the underlying distribution of transit times is also the same*

Reply:

Good point. This should indeed read as "heavy tails", i.e. not exponentially bounded tails. This will be re-formulated to "Many of these studies relied on lumped parameter convolution integral approaches with time-invariant TTDs whose pre-defined functional form when applied with seasonally variable tracers was limited to shapes (e.g. exponential) that already a priori precluded the representation of heavy tails and thus old ages."

**(21) Comment:**

*L128: „in a spatially lumped way". Yes, but for the SAS, one also uses a „lumped" input. And Maloszewski et al. [9] for instance modelled two separate reservoirs as well as quickflow „with a turnover time up to hours or days", so not quite lumped. And how probable „aggregation problems" are might depend quite significantly on the size of the watershed, and how smart the isotopic sampling was done*

Reply:

No, it is factually incorrect that SAS generally uses a lumped input. The IM-SAS-D implementation in our study uses a spatially distributed input: each precipitation zone is characterized by an individually different precipitation and tracer input at each time step. That is the whole point of the discretization into precipitation zones, as also explained in detail in the reply to Comment (2) above.

Adverse effects of aggregation were shown to arise from systems that exhibit strong internal contrasts in water ages (Kirchner, 2016). This may be related to the size of a system or not. We do not know, as it will depend on the specific system of interest.

**(22) Comment:**

*L135: Looking at the graphs showing modelling results in Rodriguez et al., I find it striking how bad the fit is. Sure, most measurements are within the confidence intervals, but this is masking the fact that the best solution misses most of the individual data points. Given this, how much credit should one give to the comparison of mean transit times done by Rodriguez et al. ?*

Reply:

We find it inappropriate to comment on the subjectively rated "bad" results of another study. In our opinion, the results of Rodriguez et al. (2021) indeed show some room for improvement but their model still captures the overall levels and major fluctuations of both $^2$H and $^3$H to a level of what can be expected of state-of-the-art models. This includes, for example, the clear dips in stream $^2$H for November 2016 or the relatively high $^3$H levels in spring 2017 in their study catchment. Overall, their model results are not much better or much worse than the results of many other similar studies, in particular those that use $^3$H – see also reply to Comment (3) above.

**(23) Comment:**

*L139: I agree with Stewart et al.. Given the constant average value of tritium over the seven years of measurements in the Weierbach catchment, one has to conclude that the tritium peak has already been flushed out, which indicates mean tracer transit times of a few years at most, i.e. a negligible flux from flow lines with transit times longer than that*

Reply:

Sure, this is a plausible possibility. We never contested that in the original manuscript and even explicitly stated it in L.139.

**(24) Comment:**

*L146: What do the authors mean by „integrated" ? That both the tracer and water fluxes are modelled ? If one is interested in studying tracer storage and release dynamics, why try at the same time to reproduce measured discharge as well instead of using it as constraint ? Adding a hydrological model to the model*

*describing tracer transport is bound to complicate the parameter estimation procedure and increase the overall „uncertainty" by increasing the number of parameters needed fitting. And lumped parameter models are also „processbased", since the transit time distribution should be chosen to reflect the hydrogeological situation, and in the case of variable flow, the tracer fluxes explicitly depend on storage volume, which controls discharge out of the system.*

Reply:

The term "integrated" means what it says: sampling of water with different ages following a sampling function integrated into a model that explicitly represents the temporal fluctuation of water storage volumes and fluxes. Given that water isotopes are quite obviously part of water molecules, it would be surprising *not* to attempt to find formulations that can describe both, the physical movement of the individual molecule (or parcels of molecules) at specific flow velocities and at the same time the actual water volumes that reach the stream (or any other state or flux in the system), following the propagation of a pressure wave at specific celerities (e.g. McDonnell and Beven, 2014). If this increases the overall uncertainty is in reality *impossible* to know, as we do not know what the real system response is. What it does instead is that it relaxes the very strong assumptions in CO models and thus, by extension, indeed increases the uncertainty *admitted* in the model by diluting the frequently perceptive sense of accuracy in CO models – see also reply to Comment (22). In addition, it depends on the perspective. Many papers have shown that the use of tracers in integrated models is indeed very valuable to *reduce* uncertainty (e.g. Fenicia et al., 2008; Birkel et al., 2010, 2015; Birkel and Soulsby, 2015; Kuppel et al., 2018; Piovano et al., 2019; Rodriguez et al., 2021; Stadnyk and Holmes, 2023).

**(25) Comment:**

*L147: Since lumped parameter models can also be used in variable flow situations, why did the authors not do it for a fair comparison ? It is a bit like comparing two racing cars, but with one of them forced to stay in first gear for the entire race.*

Reply:

Because the objective of our study is not to explore the capability of variable flow CO models nor to do any full-fledged model comparison at all. Instead, our study analyses if $^{18}$O does generally not see older water ages. This argument was previously made – from different perspectives – by DeWalle et al. (1997) and ST2010. Both studies based their assessment on the results of previous studies that mostly (if not all of them) used time-invariant, steady-state CO model implementations. Adding a variable flow CO model would not add any additional information to our study. See also the detailed reply to Comment (4) above.

**(26) Comment:**

*L150: I do not think that Stewart et al. meant that the bias in estimated mean transit time is "systematic". Rather, they warned that this might be the case more often than not, and that one should be aware of this, and if possible use both tracers simultaneously. Or to put it in a different light, if the actual transit time distribution does not deviate too much from the theoretical model, both estimates should be about the same. So maybe this is making much ado about nothing, and wanting to prove more than can actually*

*be proven. Also, how do you generalize the acceptance or the rejection of this hypothesis for one catchment to all possible catchments ?*

Reply:

We will not speculate on what ST2010 *meant*, we keep to what is *written* in their paper and many other papers since then and in which little – if any – qualification as "warning" can be found. See detailed replies to Comment (1) above.

**(27) Comment:**

*L153: Choosing an extremely large watershed, displaying an elevation difference of nearly a thousand metres and a precipitation difference of 900 mm per year, with an isotopic signal potentially influenced by snow fractionation, may not be the best choice considering the limitations the authors have described before.*

Reply:

See detailed replies to Comments (2) and (3) above.

**(28) Comment:**

*L174: Since the output was only available at the downstream end of the Neckar, near its confluent with the Rhine, only the input was roughly spatially distributed. For such a large watershed, I think this is a serious limitation of the data set, as the output lumps together so many different subwatersheds with different characteristics and hydrological responses. This seems contradictory to the warning higher in the text about "aggregation problems".*

Reply:

We agree. Of course this is a limitation of the study, which is also explicitly discussed in the original manuscript. However, the spatial discretization of input, i.e. precipitation and tracer concentration, together with a model calibration and testing procedure that forces the model to simultaneously reproduce eight different signatures of the system response and now additionally the flow at three river gauges within the study basin goes far beyond what is done in the vast majority of modelling studies. The robustness of the model and its ability to also well reproduce streamflow at internal gauges, then also allows some insights into the role of heterogeneity and potential spatial differences in TTDs. Please also see detailed replies to Comments (2) and (3) above.

**(29) Comment:**

*L210: I understand the desire to take spatial variability of the input into account, but using kriging adds more parameters and more a priori decisions to the modelling.*

Reply:

Yes.

**(30) Comment:**

*L253: It is a pity that using lumped parameter models in transient mode was not considered in the step-wise approach adopted here.*

Reply:

Please see replies Comments (4) and (25) above

**(31) Comment:**

*L273: Another common lumped parameter model is the dispersion model. Given the size of the watershed and the large macrodispersion to be expected, using it too might have been useful.*

Reply:

The general shapes of the dispersion model resembles the shapes of gamma models with shape parameter $\alpha > 1$. As we have now included a full calibration of the gamma model (see above reply to Comment (4)), which showed no indication of any values of that come close to that, we have not further explored this option.

**(32) Comment:**

*L281: Why is "a priori" italicized here, but not even mentioned on line 365, where the authors chose a priori a uniform distribution for the SAS functions ? To be clear, one or more functions describing the storage of the tracer within the watershed are needed for LPM and SAS approaches, and they have to be chosen a priori. But the choice for the LPM is NOT arbitrary, as the transit time distributions can be derived from mass balance and groundwater hydraulics, be it the exponential, gamma or dispersion model. In that regard, the SAS approach is less process based, not more, as to my knowledge, there still is no physically-based justification for choosing uniform rather than gamma functions or anything else to describe how tracer is released from storage. Effectiveness (against which hard constraint ?) is too vague a reason, and numerical convenience as mentioned on line 371 is even a bad one*

Reply:

The expression "*a priori*" is italicized throughout the manuscript as it is a Latin expression and thus a foreign language expression which are italicized in standard publishing styles.

The fact that sampling functions need to be *a priori* chosen is explicitly mentioned in L.61. The utilization of "use" in L.365 of course also implies that they are *a priori* chosen (how else could they be used then?).

We agree, that no general formulation has been found for the shape of sampling functions, but this is subject to ongoing research (e.g. Harman and Kim, 2014; Kim et al., 2022). It is of course also not unproblematic to claim that parametric TTDs, such as exponential or gamma distributions, can describe real world systems in which TTDs *necessarily* have to be jagged and highly irregular (e.g. Benettin et al., 2015).

The "numerical convenience" is not a "bad" reason. It is a reality and a hard constraint on what can feasibly be done, given the memory requirements of explicitly computing TTDs over long time periods, which can and do easily exceed the memory of good standard computers (~32GB).

**(33) Comment:**

*L290: Since the authors kept alpha at 0.5, it is strictly speaking not a calibration parameter.*

Reply:

The gamma models are now also fully calibrated.

**(34) Comment:**

*L303: I find the description of the hydrological model too superficial for an element that is essential for calculating variable tracer fluxes*

Reply:

This is quite a standard hydrological model and all essential elements describing the functioning of the model including all relevant model equations necessary to build the model are provided in the original manuscript. For more details on the model development, a list of multiple references is given in the manuscript.

**(35) Comment:**

*L320: For a watershed of the size of the Neckar, not all water entering the channel on day "t" will exit on the same day, so channel routing becomes necessary as well. Was this implemented here ? Judging from figure 2, it does not seem to be.*

Reply:

Given an average river flow length of ~150km in the basin (with the longest tributary at ~ 250 km) and typical river flow velocities of ~ 2m/s, which can reach > 4m/s under high flow conditions (https://pudi.lubw.de/detailseite/-/publication/40824, in German), the average flow time is ~ 20hrs (or ~10hrs under high flow conditions). This further implies that the water entering the channel on a day will have largely left the basin on that day or on the day after, at latest. Channel routing was therefore considered to be of minor relevance here, and it was in a deliberate decision in the spirit of model parsimony not added to the model.

**(36) Comment:**

*L374: This is seen from a modeller's perspective, but could also be explained physically, as Zuber has done in his 1986 paper.*

Reply:

We are aware of that (see also Hrachowitz et al., 2016, 2021) and will add it here.

**(37) Comment:**

*L375: This is Zuber's [4] "minimum volume". Please cite his paper.*

Reply:

It is of course *also* Zuber's (1986) "minimum volume" but others used and documented it already before that, e.g. "Bmin" of e.g. Christophersen and Wright (1981) and potentially even others before that.

**(38) Comment:**

*L383 : The description of the sine-wave model is 11 lines long, that of the lumped parameter models 15 lines long, and that for the SAS model 88 lines long, which reflects well the difference in complexity. I wonder whether the data available warrants such a complex approach requiring so many fitting parameters*

Reply:

This is not about difference in complexity as in detailed explained in the reply to comment (4). Assuming that JF equates complexity to the number of parameters, there is no difference, as the actual SAS approach only requires 2 parameters, i.e. a shape parameter for the sampling function and a parameter defining the maximum storage. This section describes how the SAS approach is used here.

**(39) Comment:**

*L397: The implementation of the spatially distributed model requires many assumptions and additional parameters (8 compared to the "lumped" SAS model, which already has 11), all of which are solely constrained by a single measured output for discharge and two tracers at the outlet of the entire watershed with a total surface area of 13,000 square kilometres. Is this reasonable ?*

Reply:

The results clearly suggest that it is reasonable, as explained in detail in the replies to Comments (2)-(4) above. In addition, we note that CO models require just as many assumptions. The difference in these CO models, in particular for the time-invariant formulations, is that the assumptions are even stronger: the use of these models entails a strong prior belief and the use of narrow  model priors of *all other processes* in the system except for the parameter of the TTD. In other words, the prior distributions of the

parameters of *all other process* and influencing factors, such as variable flow or evaporation, collapse to Dirac Delta functions which reflect the belief of the modeller that all these other processes occur exactly in the way the modeller assumes them to occur. The modeller therefore assumes either to have a more complete knowledge than (s)he actually has and/or assumes that all other processes in the hydrological system are actually negligible. As a consequence, such models are much more vulnerable to Type II errors (e.g. Dekking et al., 2005), i.e. false negatives and thus rejecting a good model when it should have been accepted (Beven, 2010).

**(40) Comment:**

*L416: Why choose daily time steps, since the tracer data is available on a monthly basis and the stream gauge is situated at the outlet of a 13,000 square kilometres watershed ? Coarser time steps might also reduce the problems of overestimation of the discharge shown on figure 5.*

Reply:

Because the availability of daily precipitation and streamflow data allows to do that, which in turn allows to represent – to some degree – the shorter time-scale fluctuations in the system. In spite of the size of the basin, the model can resolve the daily resolution rather well as described above in the replies to Comments (2) and (3). Note that it is factually incorrect that the model has problems with "overestimation of the discharge".

**(41) Comment:**

*L421: Rainfall-runoff modelling is a whole branch of hydrology in itself, and here, the authors have coupled it with a tracer storage and release routine. Isn't this adding up difficulties instead of reducing them ? And should not the authors be more critical of modelling results obtained with relatively little data with which to constrain the numerous model parameters?*

Reply:

By forcing the model to simultaneously reproduce *eight individual system response signatures* (including 2 tracers; e.g. Stadnyk and Holmes, 2023) and, now additionally, the stream flow in *three sub-catchments* within the Necker, we have exposed the model to a rigorous calibration and evaluation/test procedure that goes *far beyond* the vast majority of modelling studies, which are typically based on one or two performance metrics. It is hard to conceive of even stricter calibration and evaluation procedures. Please see detailed replies to Comments (3) and (4) above. Although there is no trivial way to meaningfully compare it, we would nevertheless put this into further perspective: the 18 model parameters of the IM-SAS-D implementation are constrained by 8 performance metrics (+3 sub-catchments). This gives, even for our most complex model, a parameter to performance metric ratio of 18/8=2.25. This comes, for example, close to the ratio of 2/1=2 for a CO-GM model (2 parameters), constrained by 1 performance metric and is even *lower* than the ratio 3/1=3 of a CO-2EM (2 exponential distributions) model constrained by the $^{18}O$ stream tracer concentration alone.

**(42) Comment:**

*L455: Why relegate the graphs showing the fits of the "base line models" in the supplementary material ? This does not help the reader to make a judgment for himself concerning the quality of the respective fits*

Reply:

Because these models are of secondary relevance to address the research hypothesis as in detail explained in reply to Comment (4) above. In any case, the performance metrics necessary to assess the model performances are fully reported in Table 4 in the main text of the original manuscript for convenience of the reader and she can assess the supplementary material for details.

**(43) Comment:**

*L460: That seasonal fluctuations are not reproduced without adjusting the fluxes to storage variation is not surprising. But for tritium dating, this is of no importance, because the passing of the tritium peak and the tritium decrease over time is what is used for fitting. See Zuber et al. [5] for a discussion of this.*

Reply:

We completely agree that for the models it is more relevant to reproduce the general tritium levels, while seasonal fluctuations are less important. We do not suggest otherwise in the text.

**(44) Comment:**

*L467: Obviously, an 11-parameter model will in many cases yield a better fit than a 1 parameter model. But avoiding overparameterization is also important in a sound scientific approach.*

Reply:

This is in fact much less obvious than claimed in the comment above. To avoid model overfitting there are several options that are extensively described in literature and that include (a) to reduce number of parameters or (b) to increase the number of performance criteria (e.g. Gupta et al., 1998, 2008; Efstratiadis and Koutsoyiannis, 2010; Hrachowitz et al., 2014). We here chose to follow (b). Please see the detailed replies to Comments (2), (3) and (41) above.

**(45) Comment:**

*L471: It is not surprising that seasonal fluctuations are better reproduced by a model that takes seasonal variations in storage into consideration, compared to a model that does not. The same behaviour could most probably be obtained by using the lumped parameter models with a variable flow formulation*

Reply:

This is indeed very encouraging, as it suggests that the models do, to some extent, what they are expected to do.

*(46) Comment:*

*L475: Same question as above. A hydrological model with 9 free parameters should reproduce well any stream hydrograph, if only one stream gauge is considered, but an important question is whether the data is sufficient to constrain model parameters in a way that is meaningful and not parameter tweaking.*

Reply:

Same answer as above. Please see the detailed replies to Comments (2), (3), (41) and (44).

*(47) Comment:*

*L480: A couple of comparative graphics might do better than this long and rather tedious analysis of the respective model performances*

Reply:

In principle, we agree that visualizations do convey message often more clearly than text/tables. However, as the manuscript has already many Figures and a full illustration of all aspects would require quite some more Figures, we decided to only show selected results in Figure 5 and Figures S10-14 in the Supplementary Material and to instead report the full performance metrics in Table 4.

*(48) Comment:*

*L485: It is not really surprising for a watershed of this size that the departure of the transit time distribution from an exponential model is large enough to lead to a discrepancy between estimated mean transit times. The authors could have taken up Stewart et al.'s [3] and Farlin and Maloszewski's [14] suggestion and used a double exponential to take this potential departure into account. Alternatively, varying the alpha parameter of the gamma model might have allowed a combined good fit to both oxygen-18 and tritium by increasing the weight of the very short transit times. A graph showing the fit is essential in the main text, rather than relegated in the supplemental information.*

Reply:

We agree, it is not surprising. We have now added three more CO models (including double exponential) and did a full calibration for CO-GM as explained above in the replies to Comment (4).

*(49) Comment:*

*L486: Transit time distributions are not explicitly defined in the SAS approach, but since the selection functions are, transit time distributions are still implicitly defined.*

Reply:

The TTDs may be partially, but surely not fully defined, as they do not only depend on the sampling function but, critically also, on the variable storage volumes.

**(50) Comment:**

*L488: The importance of storage volume for the mean transit times was indeed shown, but by Maloszewski and Zuber in 1983 [13] and Zuber in 1986 [4].*

Reply:

These papers have definitely shown the role of storage, but the general understanding goes much further back (e.g. Bolin and Rodhe, 1973; Nir, 1973; Christophersen and Wright, 1981). In any case, none of these did so in the explicit context of SAS models. We thus keep the reference Harman (2015) as is.

**(51) Comment:**

*L490: Zuber [4] was I think the first to clarify the importance of what the authors call "passive storage volume" in isotope hydrology*

Reply:

Depends on what is meant by "clarify". The idea was definitely used earlier than that, at least going back to Christophersen and Wright (1981) and their use of parameter "Bmin" in the Birkenes model.

**(52) Comment:**

*L513: The fraction of younger water used to be applied loosely relatively to "older" water. But I suppose the authors refer here to Kirchner's young water fraction, in which case, they might want to cite his paper, and correct the definition to between 2 and 3 months. Incidentally, the notation "F(T<3 m)" has not been defined previously (one has to guess the "m" stands for "months", for instance).*

Reply:

It is, as described in the original manuscript on L.513, the fractions of water younger than 3 months, i.e. F(T<3m), which is loosely comparable to Kirchner's definition, which is cited in the caption of Table 5. We will define "m" as months in the text.

**(53) Comment:**

*L515: Nothing conclusive can be gained from this comparative analysis, as the setup of the lumped parameter modelling was artificially kept to a bare minimum, ignoring more complex possibilities such as variable flow rates or combining models (here for instance two exponential, or allowing the alpha parameter to vary, not to mention running the convolution with a variable storage volume).*

Reply:

As clarified in replies to Comment (4) and several others, CO models were here exclusively used to get an idea if similar results to those in the studies cited in ST2010 would be found. An exhaustive model comparison was never the objective of our analysis, as clearly formulated in the research hypothesis.

**(54) Comment:**

*L543: The equation relating mean transit time and storage volume can be found in Maloszewski and Zuber [13]. The phrasing is slightly misleading, as it implies that storage estimation is only possible with the SAS, which is not correct*

Reply:

It can even be found in much earlier papers (e.g. Bolin and Rodhe, 1973; Nir, 1973). Apart from that, this is a neutral statement of what IM-SAS models do. It is not stated here that "only" SAS models can do that.

**(55) Comment:**

*L566: But the authors have failed to follow up on Stewart et al.'s [3] suggestion to use a double exponential in combination with both tracers*

Reply:

We have not "failed" to do so. Instead, we have made the deliberate decision not to use both tracers to calibrate the CO models as this would quite obviously not contribute anything to answering the question if $^{18}$O, when used *alone*, underestimates water ages. See detailed replies to Comment (4).

**(56) Comment:**

*L576: This line of reasoning seems very biased to me. The point is that in order to simulate both tracer and water fluxes, the SAS need 11 to 19 parameters, all of which must be constrained solely by three time series (two tracers and discharge), all measured only at the outlet of a huge watershed. And one could very well (i) calibrate a lumped parameter model simultaneously for both tracers, as this only depends on the optimization procedure chosen, and (ii) estimate from the discharge measurements the additional parameter needed to add variable fluxes to the convolution. With the lumped parameter approach, this would be three to four parameters, depending on model choice.*

Reply:

Please see the detailed replies to Comments (2), (3), (4), (41), (44), (46), (55) and several others. The IM-SAS models are well constrained and tested (8 performance metrics + now 3 sub-catchments) and suggestions (i) and (ii) do not contribute to testing the research hypothesis.

**(57) Comment:**

*L604: Before concluding that lumped parameters "are incapable of extracting meaningful information" from stable isotope measurements, the authors should first use lumped parameter models to their full potential.*

Reply:

We have now added further time-invariant CO models individually calibrated to $^{18}$O (and $^3$H), which are at the basis of previous arguments that $^{18}$O cannot see older water (De Walle et al., 1997; ST2010). Any other type of CO model implementation does not contribute to test our research hypothesis (see replies to Comment (4) and others). To avoid misunderstandings, we add *time-invariant* to the lumped convolution integral models.

**(58) Comment:**

*L605: In the scientific method, "anecdotal evidence" may be useful initially to recognize a problem, but has no place in the argumentation that should follow the first hunch.*

Reply:

While it is not used in any formal way to assess the results of our study, there is no reason not to mention or not to take the results of these studies serious here. Please also note that the conclusions of ST2010 were to a large part based on such anecdotal evidence (see Tables II and III therein).

**(59) Comment:**

*L616: The basis for the authors' argument is provided by using lumped parameter models inappropriately, and hence, cannot stand as solid evidence*

Reply:

We used the same types of time-invariant SW/CO model implementations, individually calibrated with $^{18}$O (and $^3$H) as in DeWalle et al. (1997) and the studies reported in ST2010. These are the papers that much of the mainstream consensus that $^{18}$O cannot see older water is based on. We therefore use these CO in a perfectly appropriate way to test our research hypothesis. Any, more detailed implementations, e.g. transient modes and, in particular, calibration to both tracers, will very likely find older water. But again this is not the point of our analysis, which is explicitly defined in our research hypothesis and it will also not contribute to test the hypothesis. See also replies to Comments (4) and many others above. To avoid further misinterpretation, we will include a clear statement in the discussion about time-variant models and calibration to both tracers at the same time.

**(60) Comment:**

*L619: Maybe, but then why haven't the authors made use of the possibilities offered by lumped parameter models to consider transient flow and hydrological information ? The authors have arrived at the conclusion that the hypothesis can be rejected only by ignoring most possibilities offered by lumped parameter models*

Reply:

The analysis and discussion is all about the apparent underestimation of water ages that emerge from *time-invariant* CO models that are *exclusively* calibrated to $^{18}$O as reported by DeWalle et al. (1997) and ST2010. We nowhere contest that implementations of transient flow or simultaneous calibration to $^{18}$O and $^{3}$H cannot find older water ages. The analysis is not about demonstrating that such implementations do or do not work. It is exclusively about testing if $^{18}$O as tracer alone can, if used in a suitable model (whatever that model is – could just as well be a transient CO model), see older water. If we (plausibly) assume that a transient CO model could see older water – excellent. That would be even more evidence to reject our hypothesis that $^{18}$O cannot see older waters. If such a model could not see older water – also good. The hypothesis could nevertheless be rejected based on the results of IM-SAS.

***(61) Comment:***

*L637: Actually, what James Kirchner meant was that estimating the mean transit time using the damping of the amplitude of a seasonal tracer measured at the outlet of a watershed where subwatersheds display dramatically different mean transit times can be completely erroneous, because the relationship between mean transit time and damping is not linear, whereas tracer mixing is. I see no reason why the SAS should not be just as prone to this kind of error, since the method also adopts a simple input-output approach. Splitting up the catchment into sub-regions does not change this if only done for the input. And given the size of the cathment, this problem might even be extreme. Or do the authors expect on the opposite that the size of the basin smoothes out subcatchment differences ? This is worthy of a much more thorough consideration in the discussion, and the authors should at least give solid qualitative reasons for neglecting aggregation problems.*

Reply:

We are fully aware of that. As IM-SAS-D is forced by spatially different input, it also produces differences in the TTDs generated by the four precipitation units as can be seen in Figure FR2 above (in fact, even in those produced by the individual HRUs at individual elevation zones, although these have not been explicitly tracked, due to insufficient computer memory). If the combination of these TTDs would have resulted in significantly older water ages than the TTDs generated by the lumped IM-SAS-L implementation, this would have been very strong evidence for the relevance of aggregation in real world systems. No major differences were found (see Table 5, Figure 7), which can be interpreted in different ways, but which in any case leaves the question of the role of spatial heterogeneity in TTDs/MTTs unresolved with the available data, as described in detail in Section 6.2.

***(62) Comment:***

*L675: All conclusions reached in this paragraph are based on (i) a simplistic implementation of lumped parameter models that is far from the state of the art and (ii) the reliance on a overparameterized SAS model that fails to reproduce both tracer and discharge dynamics. All this should be redone from the ground up.*

Reply:

We have extensively refuted these assertions in our replies above.

***(63) Comment:***

*L1030: For both lumped parameter models used, the exponential and the gamma functions, the authors calibrated one parameter, and consequently ended up with a single best fit. Isn't that something like an advantage in a way ? Using more parameters that could be independently determined used to be a no go in hydrology up to the turn of the century. Also, the gamma model has actually two free parameters, not one, so keeping the alpha parameter constant at 0.5 is an a priori decision that seems strange after the authors' warning against a priori decisions concerning lumped parameter models further up in the text. And the authors have not considered the winter to summer infiltration ratio, which often shifts the mean annual isotope values towards the winter average [16]. Concerning the number of parameters, the SAS models used have between 11 and 19 parameters, compared to the one parameter for the lumped parameter models (two for the gamma, plus one if considering the winter to summer recharge ratio, plus one if making unsteady state calculations, which should have been done to exploit fully the possibilities of lumped parameter models in variable flow systems and allow a fair comparison with the SAS results). Given the data set used for parameter estimation is the same and consists only in measured inputs and outputs to two different tracers, are not the results of the lumped parameter models, being much more parsimonious, also much less uncertain ? Not trying to reproduce discharge, but only focusing on the isotopes, could help reduce the number of fitting parameters of the SAS models.*

Reply:

It is not clear in how far one single best fit would constitute an advantage. Instead, it gives a perceptive sense of accuracy by imposing very narrow prior assumptions on the model, suggesting much more confidence on what we know about the system than we have in reality and making models more vulnerable to Type II errors. Please see also the replies to other, related comments above.

Note that the remaining comments on the Figures are mostly repetitions of the comments above and have been addressed by our replies above. We therefore will not respond to them in detail here.

We hope that the detailed clarifications above and modifications to our manuscript clear up any misinterpretations and avoid misrepresentations of our work, and hope the extensive discussion here benefits the research community.

Best regards,

Markus Hrachowitz, Siyuan Wang, Gerrit Schoups, Christine Stumpp

**References** (includes only the ones not cited in our original manuscript)

Barnes, C. J., & Bonell, M. (1996). Application of unit hydrograph techniques to solute transport in catchments. Hydrological Processes, 10(6), 793–802.

Bergström, S., and Forsman, A. (1973) Development of a conceptual deterministic rainfall runoff model, Nordic Hydrology, Vol. 4, No. 3.

Bergström, S., Carlsson, B., Sandberg, G., & Maxe, L. (1985). Integrated modelling of runoff, alkalinity, and pH on a daily basis. Hydrology Research, 16(2), 89-104.

Beven, K. J. (2010). Preferential flows and travel time distributions: defining adequate hypothesis tests for hydrological process models. Hydrological Processes, 24(12), 1537-1547.

Birkel, C., & Soulsby, C. (2015). Advancing tracer‐aided rainfall–runoff modelling: A review of progress, problems and unrealised potential. Hydrological Processes, 29(25), 5227-5240.

Birkel, C., Soulsby, C., & Tetzlaff, D. (2015). Conceptual modelling to assess how the interplay of hydrological connectivity, catchment storage and tracer dynamics controls nonstationary water age estimates. Hydrological Processes, 29(13), 2956-2969.

Bolin, B., & Rodhe, H. (1973). A note on the concepts of age distribution and transit time in natural reservoirs. Tellus, 25(1), 58–62.

Burnash, R. J., Ferral, R. L., and McGuire, R. A. (1973). A generalized streamflow simulation system, conceptual modeling for digital computers, Joint Federal and State River Forecast Center, U.S. National Weather Service, and California Department of Water Resources Tech. Rep., 204 pp..

Christophersen, N., & Wright, R. F. (1981). Sulfate budget and a model for sulfate concentrations in stream water at Birkenes, a small forested catchment in southernmost Norway. Water Resources Research, 17(2), 377-389.

Christophersen, N., Seip, H. M., & Wright, R. F. (1982). A model for streamwater chemistry at Birkenes, Norway. Water Resources Research, 18(4), 977-996.

De Grosbois, E., Hooper, R. P., & Christophersen, N. (1988). A multisignal automatic calibration methodology for hydrochemical models: a case study of the Birkenes model. Water Resources Research, 24(8), 1299-1307.

Dekking, F. M., Kraaikamp, C., Lopuhaä, H. P., & Meester, L. E. (2005). A Modern Introduction to Probability and Statistics: Understanding why and how (Vol. 488). London: Springer.

Efstratiadis, A., & Koutsoyiannis, D. (2010). One decade of multi-objective calibration approaches in hydrological modelling: a review. Hydrological Sciences Journal–Journal Des Sciences Hydrologiques, 55(1), 58-78.

Fenicia, F., McDonnell, J. J., & Savenije, H. H. (2008). Learning from model improvement: On the contribution of complementary data to process understanding. Water Resources Research, 44(6).

Gupta, H. V., Sorooshian, S., & Yapo, P. O. (1998). Toward improved calibration of hydrologic models: Multiple and noncommensurable measures of information. Water Resources Research, 34(4), 751-763.

Gupta, H. V., Wagener, T., & Liu, Y. (2008). Reconciling theory with observations: elements of a diagnostic approach to model evaluation. Hydrological Processes: An International Journal, 22(18), 3802-3813.

Harman, C. J., & Kim, M. (2014). An efficient tracer test for time‐variable transit time distributions in periodic hydrodynamic systems. Geophysical Research Letters, 41(5), 1567-1575.

Hooper, R. P., Stone, A., Christophersen, N., de Grosbois, E., & Seip, H. M. (1988). Assessing the Birkenes model of stream acidification using a multisignal calibration methodology. Water Resources Research, 24(8), 1308-1316.

Hrachowitz, M., Fovet, O., Ruiz, L., Euser, T., Gharari, S., Nijzink, R., ... & Gascuel‐Odoux, C. (2014). Process consistency in models: The importance of system signatures, expert knowledge, and process complexity. Water resources research, 50(9), 7445-7469.

Kim, M., Volkmann, T. H., Wang, Y., Meira Neto, A. A., Matos, K., Harman, C. J., & Troch, P. A. (2022). Direct observation of hillslope scale StorAge selection functions in experimental hydrologic systems: Geomorphologic structure and preferential discharge of old water. Water Resources Research, 58(3), e2020WR028959.

Koeniger, P., Wittmann, S., Leibundgut, C., & Krause, W. J. (2005). Tritium balance modelling in a macroscale catchment. Hydrological Processes: An International Journal, 19(17), 3313-3320.

Kratzert, F., Klotz, D., Shalev, G., Klambauer, G., Hochreiter, S., & Nearing, G. (2019). Towards learning universal, regional, and local hydrological behaviors via machine learning applied to large-sample datasets. Hydrology and Earth System Sciences, 23(12), 5089-5110.

Kuppel, S., Tetzlaff, D., Maneta, M. P., & Soulsby, C. (2018). EcH 2 O-iso 1.0: Water isotopes and age tracking in a process-based, distributed ecohydrological model. Geoscientific Model Development, 11(7), 3045-3069.

Liang, X., Lettenmaier, D. P., Wood, E. F., & Burges, S. J. (1994). A simple hydrologically based model of land surface water and energy fluxes for general circulation models. Journal of Geophysical Research: Atmospheres, 99(D7), 14415-14428.

Lundquist, D., Hydrochemical modelling of drainage basins, SNSF Project (in Norwegian), Intern. Rep. 31/77, 27 pp., Agric. Res. Counc. of Norway, Aas, 1977

Mizukami, N., Rakovec, O., Newman, A. J., Clark, M. P., Wood, A. W., Gupta, H. V., & Kumar, R. (2019). On the choice of calibration metrics for "high-flow" estimation using hydrologic models. Hydrology and Earth System Sciences, 23(6), 2601-2614.

Newman, A. J., Mizukami, N., Clark, M. P., Wood, A. W., Nijssen, B., & Nearing, G. (2017). Benchmarking of a physically based hydrologic model. Journal of Hydrometeorology, 18(8), 2215-2225.

Niemi, A. J. (1977). Residence time distributions of variable flow processes. The International Journal of Applied Radiation and Isotopes, 28(10-11), 855-860.

Nir, A. (1973). Tracer relations in mixed lakes in non-steady state. Journal of Hydrology, 19(1), 33-41.

Piovano, T. I., Tetzlaff, D., Carey, S. K., Shatilla, N. J., Smith, A., & Soulsby, C. (2019). Spatially distributed tracer-aided runoff modelling and dynamics of storage and water ages in a permafrost-influenced catchment. Hydrology and Earth System Sciences, 23(6), 2507-2523.

Popper, K.R.: The logic of scientific discovery (1934)

Samaniego, L., Kumar, R., & Attinger, S. (2010). Multiscale parameter regionalization of a grid‐based hydrologic model at the mesoscale. Water Resources Research, 46(5).

Seip, H. M., R. Seip, P. J. Dillon, and E. de Grosbois, Model of sulfate concentration in a small stream in the Harp Lake catchment, Ontario, Can. J. Fish. Aquat. Sci., 42, 927-937, 1985

Stadnyk, T. A., & Holmes, T. L. (2023). Large Scale Hydrologic and Tracer Aided Modelling: A Review. Journal of Hydrology, 129177.

Stewart, M. K., Morgenstern, U., Gusyev, M. A., & Małoszewski, P. (2017). Aggregation effects on tritium-based mean transit times and young water fractions in spatially heterogeneous catchments and groundwater systems. Hydrology and Earth System Sciences, 21(9), 4615-4627.

---

## Author Response (AR2)

The following revised manuscript is based on Reviewer#2 comments with highlight shades.

**Response to Editor:**

*Both the Referees agree about the great improvement of the revised manuscript, which now substantially meets the standards required for publication in HESS.*
*However, as pointed out by Referee #2, the manuscript would benefit if some of the results of the additional model simulations, carried out in response to some of the comments received during the peer review process, could find more room in the discussion of the results.*
*Hence, I ask you this final effort before considering the paper suitable for publication.*

**Reply:**

We thank the Editor for the positive assessment of our work.

We have addressed the remaining comments of Reviewer #2 in the revisions attached. In particular we have added the results of the P-SAS model to Figures 3, 4 and 10 as requested by the reviewer. In addition, we emphasize P-SAS throughout the Results and Discussion Sections.

**Response to Reviewer #2:**

*(1) Reviewer Comment:*

*Thank you for your detailed replies to all of my comments and for including additional simulations. I think that the manuscript is now much more balanced regarding the use of different modelling concepts.*

**Reply:**

We thank the reviewer for the positive evaluation of our manuscript and thank her for the additional comments.

*(2) Reviewer Comment:*

*As the P-SAS implementation has been added to the revised manuscript, most of the figures describing simulation results in the manuscript show IM-SAS results only (Figs. 3-7; 10-12) and there is a large emphasis on these model scenarios in the text (e.g., lines 516-537; 539-566; 590-598). Given both types of SAS models give comparably good and matching results for d18O and 3H (with the absolute difference in TTs being smaller for P-SAS), would it be possible to show the results of both model types or replace some of the IM-SAS figures by P-SAS figures?*

**Reply:**

We have now added the results of P-SAS to Figure 3, Figure 4 and to Figure 10. In addition, we emphasize P-SAS

throughout the Results and Discussion Sections as highlighted in yellow in the revised manuscript (lines 514, 544, 547, 549, 551, 576, 579, 586, 608, 634, 637, 640, 643, 664, 667, 686, 688, 755, 759, 760)

**Minor Comments**

**(3) Reviewer Comment:**

*- lines 463–468: mention calibration approach for P-SAS as well.*

**Reply:**

Ok.

**(4) Reviewer Comment:**

*- lines 640–641: "to a lesser degree" sounds to me as if P-SAS were not be able to represent all signatures that accurately. I assume that what you mean here is that it cannot represent all kinds of hydrological signatures as it does not simulate streamflow.*

**Reply:**

That is correct.

**(5) Reviewer Comment:**

*- lines 642–643: related to the previous comment - CO models could also be calibrated to two different tracers (as mentioned by the authors in the reply). Hence, P-SAS and CO are in that sense more similar than P-SAS and IM-SAS.*

**Reply:**

Agreed.

[revised manuscript text omitted]